# Graph Coloring for Multi-Task Learning

## Abstract

When different objectives conflict with each other in multi-task learning, gradients begin to interfere and slow convergence, thereby potentially reducing the final model's performance. To address this, we introduce SON-GOKU, a scheduler that computes gradient interference, constructs an interference graph, and then applies greedy graph-coloring to partition tasks into groups that align well with each other. At each training step, only one group (color class) of tasks are activated, and the grouping partition is constantly recomputed as task relationships evolve throughout training. By ensuring that each mini-batch contains only tasks that pull the model in the same direction, our method improves the effectiveness of any underlying multi-task learning optimizer without additional tuning. Since tasks within these groups will update in compatible directions, multi-task learning will improve model performance rather than impede it. Empirical results on six different datasets show that this interference-aware graph-coloring approach consistently outperforms baselines and state-of-the-art multi-task optimizers. We provide extensive theory showing why grouping and sequential updates improve multi-task learning, with guarantees on descent, convergence, and the ability to accurately identify what tasks conflict or align.

## 1 Introduction

Multi-task learning (MTL) trains a single model to solve several tasks simultaneously, sharing knowledge across them to learn more effectively (Caruana, 1997). This allows models to generalize better and converge faster. However, a key issue known as negative transfer arises when tasks don't align very well with each other (Sener & Koltun, 2018; Shi et al., 2023). When two tasks push the shared network in different directions their gradients clash, slowing or even reversing learning. Prior work addresses this issue primarily via (1) gradient manipulation, which reshapes task gradients to reduce conflicts, and (2) loss reweighting, which rescales task objectives to balance their influence. While effective in some specific settings, these strategies typically treat conflict locally at the level of shared-parameter updates and often overlook the evolving global structure of interactions among tasks throughout training.

Some recent works focus on partitioning tasks into subsets (groups) and updating those groups separately. These approaches have been found to improve accuracy and training stability by forming groups with high measured affinity and then updating one group at a time (Fifty et al., 2021; Jeong & Yoon, 2025). Grouping can outperform gradient manipulation and loss reweighting when tasks form clusters with aligned gradients, because each update then reduces direct clashes in the shared layers, lowers gradient variance within the step, and lets compatible tasks reinforce one another while conflicting tasks wait for their turn.

However, grouping methods often face a few key limitations: (1) many rely on dense pairwise affinities that grow noisy and costly as the number of tasks rises (Fifty et al., 2021; Standley et al., 2020; Sherif et al., 2024); (2) others predetermine or rarely update groups, so they drift as task relations change (Wang et al., 2024; Ruder, 2017); and (3) several use local heuristics that fail to enforce global compatibility or to specify how groups should rotate over time (Zhang & Yang, 2018; Malhotra et al., 2022).

---

Anonymized code implementation is available at this URL: `https://anonymous.4open.science/r/SON-GOKU-ICLR-95AB`

We present **SON-GOKU** (**S**cheduling via **O**ptimal I**N**terference-aware **G**raph-C**O**loring for Tas**K** Grouping in M**U**ltitask Learning). We measure gradient interference, build a graph of tasks from those measurements, greedily color the graph to form non-conflicting compatible task groups, and update one color group per step during training. This design addresses the earlier issues. We estimate the interference graph from lightweight minibatch statistics and keep it sparse, which avoids noisy dense matrices and scales to many tasks. We recolor the graph at regular intervals so the groups track changing relations during training. Greedy graph coloring ensures we update only compatible tasks in each step, and the color order gives a simple way to cycle through the groups. Our proposed scheduler does not have to work in isolation, it can function on top of existing loss-reweighting and gradient-manipulation MTL approaches.

In our theoretical analysis (Section 5) we show that, under standard conditions, SON-GOKU tends to group tasks whose gradients are, on average, aligned within each group, with high probability. We further show that, over a refresh window, sequentially updating these low-conflict groups yields *at least* as much expected descent as a single mixed update, and *strictly more* when between-group interference is sufficiently negative. We also prove that SON-GOKU preserves descent and reaches the usual non-convex SGD rate under mild assumptions, with only a small factor that depends on the within-group conflict level. In Appendix D we discuss the scheduler's amortized time complexity and the tradeoffs it offers between speed and performance. We discuss ways in which practitioners can reduce its time complexity under certain conditions.

Empirical results from experiments demonstrate that SON-GOKU consistently improves outcomes compared to other MTL approaches, especially when SON-GOKU is coupled with existing approaches. Our contributions are as follows:

- We propose SON-GOKU, an interference-aware scheduler that measures cross-task gradient conflict, builds a conflict graph, colors it to form compatible groups, and activates one group per step. It can be used on top of standard MTL optimizers.

- We provide theoretical analysis that offers guarantees on SON-GOKU's grouping, convergence, scheduling behavior, and more.

- Across six datasets, SON-GOKU improves over strong baselines and pairs well with methods like PCGrad, AdaTask, and GradNorm, delivering consistent gains.

- We perform an ablation study showing that dynamic recoloring and history-averaged conflict estimates are key contributors to performance.

## 2 RELATED WORK

Prior work has identified the phenomenon of gradient interference in multi-task learning and explored several strategies to mitigate it. We group these such strategies into four families: (1) *Tuned Loss Weighting*, (2) *Adaptive Loss Weighting*, (3) *Gradient-Level Conflict Mitigation*, and (4) *Empirical Task Grouping*. SON-GOKU falls into family (4).

Many MTL methods (especially earlier ones) adjust task influence by learning or adapting loss weights. Examples include uncertainty-based scaling (Kendall et al., 2018), rate-based schemes such as DWA (Liu et al., 2019), and fast bilevel formulations like FAMO (Liu et al., 2023). FAMO in particular is notable for its $\mathcal{O}(1)$ per-step time complexity. These approaches keep all tasks active each step while modulating relative magnitudes. A completely different approach, which emerged in 2018 with MGDA (Sener & Koltun, 2018), focuses on updating shared-parameter *update directions* to mitigate interference. Methods like PCGrad (Yu et al., 2020), CAGrad (Liu et al., 2021), and MGDA (Sener & Koltun, 2018) modify the geometry of the shared update to reduce cross-task conflicts while still updating all tasks each step. A smaller body of work forms subsets of tasks to update together, using offline affinity estimation or training-dynamics signals (Fifty et al., 2021; Standley et al., 2020; Wang et al., 2024; Sherif et al., 2024). See Appendix Q for additional analysis of non-conflict task grouping. Most recently, Selective Task Group Updates proposes online grouping with sequential updates, reporting that update order can influence task-specific learning (Jeong & Yoon, 2025). SON-GOKU differs in mechanism from existing approaches (Section 4). It complements loss reweighting and gradient surgery, and we provide explicit guarantees on descent, convergence, and graph partition recovery. An expanded discussion and commentary of related work is provided in Appendix M.

## 3 PROBLEM SETUP

We formalize multi-task learning (MTL) (Caruana, 1997) as optimizing a shared network while activating only a subset of tasks at each step. Each task contributes a loss whose gradients may align or conflict. We quantify conflict using (the negative of) cosine similarity, embed tasks in a conflict graph, and later use that graph to derive a schedule (see Appendix P for a unique, modular, formulation and results with alternative measures of affinity). This section fixes notation and states the optimization goal that the proposed approach addresses.

### 3.1 DATA AND NOTATION

Let $\mathcal{T} = \{T_1, \ldots, T_K\}$ be the set of tasks. The model has shared parameters $\theta \in \mathbb{R}^d$ and task-specific parameters $\phi_k \in \mathbb{R}^{d_k}$ for $T_k$. Each task draws examples $(x, y_k)$ from a distribution $\mathcal{D}_k$ and defines a per-example loss $\ell_k(\theta, \phi_k; x, y_k)$. Its population loss is

$$L_k(\theta, \phi_k) := \mathbb{E}_{(x,y_k) \sim \mathcal{D}_k}\big[\ell_k(\theta, \phi_k; x, y_k)\big]. \tag{1}$$

We minimize the standard weighted MTL objective

$$F(\theta, \phi_1, \ldots, \phi_K) = \sum_{k=1}^{K} w_k \, L_k(\theta, \phi_k), \tag{2}$$

with nonnegative task weights $w_k$ (default $w_k = 1$). Note that, for simplicity in later sections, we absorb $w_k$ into the per-task gradient estimates. This is permissible since positive scalings do not change cosine signs or the induced conflict graph.

At step $t$, for any task $k$ that is active we compute stochastic gradients on a mini-batch $\mathcal{B}_k^{(t)} \subset \mathcal{D}_k$:

$$g_k^{(t)} := \nabla_\theta L_k(\theta_t, \phi_{k,t}; \mathcal{B}_k^{(t)}), \qquad h_k^{(t)} := \nabla_{\phi_k} L_k(\theta_t, \phi_{k,t}; \mathcal{B}_k^{(t)}). \tag{3}$$

In our proposed method, we form exponential moving averages (EMA) of per-task gradients within a refresh window to stabilize cosine estimates so that they do not become stale (Sec. 4).

#### 3.1.1 INTERFERENCE COEFFICIENT

We quantify pairwise interaction with the interference coefficient

$$\rho_{ij} = -\frac{\langle \tilde{g}_i, \tilde{g}_j \rangle}{\|\tilde{g}_i\| \, \|\tilde{g}_j\|}, \tag{4}$$

where $\tilde{g}_i$ and $\tilde{g}_j$ are the EMA-smoothed gradients at refresh. Positive $\rho_{ij}$ indicates conflict (negative cosine). $\rho_{ij} \leq 0$ indicates alignment or neutrality.

#### 3.1.2 CONFLICT GRAPH

Fix a tolerance $\tau \in (0, 1)$. The conflict graph is

$$G_\tau = (\mathcal{T}, E_\tau), \qquad E_\tau = \big\{(i, j) : \rho_{ij} > \tau\big\}. \tag{5}$$

Vertices are tasks. An edge between a pair means to not update that pair together. We will utilize $G_\tau$ for coloring and scheduling in Section 4

### 3.2 GOAL

At training step $t$ we choose an active set $S_t \subseteq \mathcal{T}$ and update only those tasks:

$$\theta_{t+1} = \theta_t - \eta_t \sum_{k \in S_t} g_k^{(t)}, \qquad \phi_{k,t+1} = \begin{cases} \phi_{k,t} - \eta_t h_k^{(t)}, & k \in S_t, \\ \phi_{k,t}, & k \notin S_t. \end{cases} \tag{6}$$

The problem the scheduler addresses is to design the sequence $\{S_t\}_{t=1}^T$ so that: (1) every task is visited regularly; and (2) conflicting tasks seldom appear together. We instantiate this via greedy graph coloring in Section 4 and analyze the guarantees in Section 5.

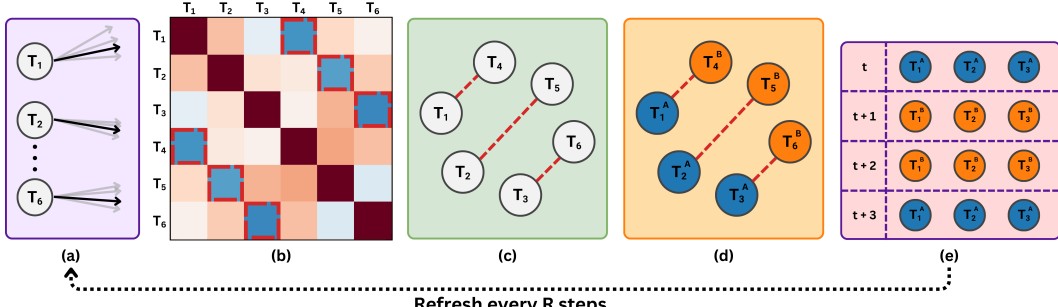

**Refresh every R steps**

Figure 1: Interference-aware scheduling pipeline: (a) For each task $T_i$ (circles $T_1 \ldots T_6$), we smooth recent per-step gradients with an Exponential Moving Average (EMA); (b) From these EMA vectors we compute the pairwise cosine matrix. In the figure, cells outlined with red dashes mark pairs with cosine $< -\tau$. These are flagged as conflicts; (c) We build the conflict graph whose nodes are tasks $T_i$ and whose red dashed edges connect exactly those pairs identified in (b); (d) We apply greedy graph coloring so that no conflict edge lies within a color, producing low-conflict groups. In the example shown, we have two groups: A as blue and B as orange; (e) During training we activate one group per step. After every $R$ steps (here, $R = 4$) we 'refresh' and run the pipeline again from step A, where we update the EMAs with the latest gradients.

## 4 PROPOSED APPROACH

We design an interference-aware scheduler that partitions tasks into low-conflict groups and activates exactly one group per optimization step. The procedure consists of four stages: (1) estimating pairwise interference, (2) building and coloring the conflict graph, (3) generating a periodic schedule, and (4) updating that schedule as training evolves. An overview of the scheduler is provided as Algorithm 1 in Appendix A. A visualization of SON-GOKU is provided in Figure 1 alongside a simple summary in the Figure caption.

### 4.1 ESTIMATING GRADIENT INTERFERENCE

We absorb task weights into per-task losses, so $g_k^{(t)}$ is the gradient of the weighted loss $w_k L_k$. Cosine calculations and graph construction are not impacted by applying positive scaling.

At step $t$ and for every task $T_k$ appearing in the current mini-batch we compute a task-specific stochastic gradient

$$g_k^{(t)} = \nabla_\theta \mathcal{L}_k\big(\theta_t, \phi_{k,t}; \mathcal{B}_k^{(t)}\big), \tag{7}$$

using an independent sub-batch $\mathcal{B}_k^{(t)} \subset \mathcal{D}_k$. We then update an exponential moving average

$$\tilde{g}_k^{(t)} = \beta\, \tilde{g}_k^{(t-1)} + (1-\beta)\, g_k^{(t)}, \qquad \beta \in [0,1), \tag{8}$$

which stabilizes cosine estimates while requiring only two buffers per task (current and previous). To estimate such cosines in practice, we employ low dimensional sketches of the EMA for each task, so the additional memory usage scales well (Ghashami et al., 2016a). Whenever we refresh the schedule (every $R$ steps) we form the pairwise interference matrix.

$$\rho_{ij}^{(t)} = -\frac{\langle \tilde{g}_i^{(t)}, \tilde{g}_j^{(t)} \rangle}{\|\tilde{g}_i^{(t)}\|\, \|\tilde{g}_j^{(t)}\|}, \qquad i, j \in \{1, \ldots, K\}. \tag{9}$$

Computing all $K(K-1)/2$ cosines via the Gram matrix in the $r$-dimensional sketch space costs $O(Kdr + K^2 r)$, namely $O(Kdr)$ to form the sketch $M_f$ and $O(K^2 r)$ for the Gram product, where

$r \ll d$ is the sketch width (see Appendix D.5.1). We also write $h_k^{(t)} = \nabla_{\phi_k} \mathcal{L}_k(\theta_t, \phi_{k,t}; \mathcal{B}_k^{(t)})$ for the gradient with respect to the task-specific parameters $\phi_k$.

## 4.2 CONFLICT GRAPH CONSTRUCTION

Given a tolerance $\tau \in (0, 1)$, the conflict graph at update round $r$ is

$$G_\tau^{(r)} = (V, E_\tau^{(r)}), \quad V = \{1, \ldots, K\} E_\tau^{(r)} = \{(i, j) \ : \ \rho_{ij}^{(t_r)} > \tau\}. \tag{10}$$

To clarify, tasks are indexed by integers $1 \ldots K$ in Equation 10. Edges connect tasks whose averaged gradients have cosine similarity less than $-\tau$. Intuitively, larger $\tau$ yields a sparser conflict graph, typically fewer colors (larger per-step groups), and more frequent updates per task. Smaller $\tau$ results in a denser graph, more colors (smaller per-step groups), and less frequent updates per task. This construction reflects optimization-time interference. $G_\tau^{(r)}$ is symmetric and undirected, derived from current gradient geometry to decide which tasks should not be updated together.

## 4.3 PARTITIONING VIA GREEDY GRAPH COLORING

We apply the Welsh-Powell largest-first greedy heuristic (Welsh & Powell, 1967) to color $G_\tau^{(r)}$ and obtain color classes $C_1^{(r)}, \ldots, C_{m_r}^{(r)}$. Classical graph-theory results (West, 2000; Diestel, 2017) guarantee the heuristic uses no more than $\Delta + 1$ colors, where $\Delta$ is the maximum vertex degree. In practice $\Delta$ is small because many task pairs do not interfere, yielding concise schedules.

## 4.4 SCHEDULE GENERATION AND EXECUTION

We create a periodic schedule of length $m_r$:

$$S_t = C_{(t \bmod m_r)+1}^{(r)}, \qquad t_r \le t < t_{r+1} = t_r + R. \tag{11}$$

Each training step activates exactly one color class; over one period every task in that class receives a gradient update, while conflicting tasks (edges in $E_\tau^{(r)}$) are guaranteed not to co-occur.

### 4.4.1 MINIMUM UPDATE FREQUENCY

If the greedy coloring yields a singleton class for a rarely updated task, we increase its update frequency by duplicating it only into steps whose active color has no conflict edge to that task.

### 4.4.2 WARM-UP AND ANNEALING

We start with $\tau = 1$ (no edges, full simultaneous training) for the first $T_{\mathrm{warm}}$ steps, then logarithmically anneal $\tau$ to a target value $\tau^*$. This mitigates noisy gradient signals early in training. Similarly, we can set the refresh period with a smaller $R$ to adapt to changing gradients and increase it as training stabilizes (Appendix O).

## 4.5 TIME COMPLEXITY AND SPACE COMPLEXITY

Using the sketched implementation described in Appendix D, a single refresh of the SON-GOKU scheduler has time complexity $O(Kdr + K^2r)$, where $r \ll d$ is the sketch width. However, unlike many MTL approaches, our scheduler concentrates its extra work in occasional refreshes. This time complexity therefore becomes $O\left(\frac{Kr(d+K)}{R}\right)$ amortized per training step where $R$ is the refresh period (the number of training steps between conflict-graph rebuilds). For the small, fixed $r$ used in our experiments, this overhead still grows roughly quadratically in $K$ but is independent of $d$ up to the $O(Kdr)$ sketching term and shrinks linearly with the refresh period $R$. Similarly, SON-GOKU's persistent space complexity of $O(K^2)$ scales with $K$ but not $d$, the number of model parameter dimensions, allowing it to maintain low memory usage even with large backbone models. We provide

a full analysis of the time complexity in Appendix D and discuss approaches to reducing time complexity under certain conditions in Appendix D.5. See also Appendix R for scaling behavior with larger backbones.

## 5 THEORETICAL ANALYSIS

We discuss some of the main guarantees behind SON-GOKU. For a very brief overview: (1) Updating groups of tasks whose gradients are mostly low-conflict (no internal edges) reduces the objective on average and still achieves the usual $1/\sqrt{T}$ convergence rate; (2) Over a refresh window, scheduling several group updates can beat one mixed update that uses all tasks at once; and (3) With a small number of recent gradient measurements per task (via EMA) and a margin separating conflicts, the estimated conflict graph matches the ideal one, giving a short schedule where every task is updated at least once every $\Delta + 1$ steps ($\Delta$ is the maximum number of conflicts for any task). We provide expanded assumptions, definitions, proofs, reasoning, analysis, etc. in Appendix 5.4–I (see also Appendix N, P–R).

### 5.1 DESCENT PRESERVATION WITHIN A LOW-CONFLICT GROUP

If the active set $S_t$ at step $t$ is $\tau$-compatible, then the combined update is a descent direction with a quantitative lower bound:

$$\Big\| \sum_{k \in S_t} g_{k,t} \Big\|^2 \ \geq \ \Big(1 - \tau\left(|S_t| - 1\right)\Big) \sum_{k \in S_t} \|g_{k,t}\|^2 \tag{12}$$

Thus the step cannot flip to ascent whenever $\tau(|S_t| - 1) < 1$. This is proved by expanding the polarization identity and controlling cross terms under the $\tau$-compatibility condition (see Appendix E). Essentially, this means that SON-GOKU's per-step updates are safe when groups are low conflict. The aggregate direction keeps pointing downhill and the cancellation is quantitatively limited by $\tau$ and group size.

### 5.2 NONCONVEX CONVERGENCE AT THE STANDARD RATE UP TO A SMALL FACTOR

Under standard smoothness and noise conditions (see Appendix I) and with steps $\eta = c/\sqrt{T}$, SON-GOKU achieves the usual nonconvex SGD rate, with a mild $(1 + \tau)$ factor that reflects within-group conflict:

$$\min_{t < T} \mathbb{E} \|\nabla F(\theta_t)\|^2 \ \leq \ \frac{2(F_0 - F^\star)}{c\sqrt{T}} (1 + \tau) \ + \ \frac{cL\sigma^2}{\sqrt{T}} \tag{13}$$

When $\tau = 0$, the constant matches the classical bound (Bottou et al., 2018; Ghadimi & Lan, 2013); as $\tau \to 1$, it at most doubles, matching the intuition that conflict can cancel up to half of the progress. This demonstrates that scheduling does not degrade asymptotic progress. SON-GOKU preserves the $1/\sqrt{T}$ decay of the gradient norm while controlling the constant through the compatibility threshold $\tau$. In other words, we keep the standard rate of SGD and trade a small constant for reduced interference.

### 5.3 WHEN SCHEDULED GROUPS OUTPERFORM A SINGLE MIXED UPDATE

We compare two ways to use the same gradients gathered at a refresh: a scheduled sequence of per-group steps (i.e., the scheduler used in SON-GOKU) versus a single aggregated step. Using a telescoping $L$-smooth bound and evaluating both trajectories at a common linearization (i.e., expanding $F$ at the refresh start $\theta_{t_r}$ and applying the same first-order model with the same step size) the scheduled bound is never worse and is strictly better when cross-group interaction terms are sufficiently negative (so mixed updates would cancel progress).

Essentially, when different groups' gradients pull in opposing directions (so adding them together would cancel progress) the scheduler has an advantage. In that case, taking the updates one group at a time is provably better. Our theory guarantees a larger drop in the objective during that refresh than the one-shot step, even though both use the same step size and the same gradients. Under the PL condition, the scheduled path maintains the usual contraction factor and gains a nonnegative extra decrease term over the window.

## 5.4 Exact Recovery of the Population Conflict Graph and Task Partition

We show that, after observing gradients for only a modest number of steps, the scheduler can exactly reconstruct the true conflict relations among tasks by averaging recent gradients (EMA), computing pairwise cosines, thresholding at $-\tau$, and coloring the resulting graph. Under a separation margin $\gamma$ around the threshold (tasks are meaningfully different), bounded noise, and bounded drift within each refresh window, the conflict graph estimated from finite data agrees, with high probability, with the ideal population conflict graph $G^\star \tau$ (defined from the pairwise cosines of the true mean gradients $\{\mu_i\}_{i=1}^{K}$ at the start of the refresh window). Equivalently, when the uniform cosine estimation error is below $\gamma$, we have $\widehat{G}_\tau = G_\tau^\star$ and the resulting grouping recovers the ground-truth task partition. This explains why the scheduler's group structure is trustworthy and ties the required number of recent gradient measurements per task to interpretable quantities such as noise level, margin, and the number of tasks. For example, an effective sample size of $n_{\text{eff}} \gtrsim \frac{\sigma^2}{m_0^2 \gamma^2} \log(K/\delta)$ suffices in our analysis.

## 5.5 Scheduling Properties with Few Groups and Bounded Staleness

Welsh-Powell greedy coloring uses at most $\Delta + 1$ colors on a graph whose maximum degree is $\Delta$ (Bonamy et al., 2018). Running the colors in a fixed cycle means each task is updated at least once every $m \leq \Delta + 1$ steps. Equivalently, no task waits more than $\Delta$ steps between updates (bounded staleness).

This means that the schedule length is controlled by the worst conflict degree $\Delta$ rather than by the total number of tasks $K$. This results in two important benefits: (1) a minimum update-frequency guarantee, since every task receives an update at least once per cycle of length $\leq \Delta + 1$; and (2) compatibility with standard bounded-delay conditions used in analyses of asynchronous SGD (e.g., Niu et al. 2011; Lian et al. 2015), with delay parameter at most $\Delta$. When $\Delta \ll K$, we achieve both low interference (few conflicts per step) and low staleness (short update gaps).

## 6 Experimental Setup

### 6.1 Datasets

We evaluate across six benchmarks spanning vision, multimodal, and time-series. For each dataset we specify a small set of primary tasks and add positive and negative auxiliaries to stress interference. Architectures are standard backbones (e.g., ResNet-18 for image tasks, CNN/BiLSTM for time-series) with task-specific heads. Full dataset and task definitions, auxiliary construction, and architecture details (including preprocessing and head designs) appear in Appendix J and Table 4. We provide additional experiments with varying backbones in Appendix R.

### 6.2 Baseline and State-of-the-Art Comparisons

We compare against loss-weighting (Uniform, GradNorm, AdaTask), multi-objective (MGDA, Nash-MTL, FairGrad), projection/surgery (PCGrad, CAGrad), and fast adaptive weighting (FAMO). We provide short method notes in Appendix K and discuss these approaches in Section 2.

### 6.3 Scheduler Extension Models

In addition to standalone models, we also evaluate combinations of the scheduler with existing approaches.

1. *SON-GOKU + AdaTask.* Combines our interference-aware task selection with AdaTask's dynamic loss weighting, applying adaptive weights only to scheduler-selected tasks.

2. *SON-GOKU + GradNorm Warm Start.* Initializes training with GradNorm for stable gradient magnitudes, then transitions to our scheduler after 3 epochs.

3. *SON-GOKU + PCGrad.* Applied PCGrad's gradient projection specifically to tasks selected by our scheduler, providing fine-grained conflict resolution within $\tau$-compatible groups.

Table 1: Performance of Evaluated Approaches Across Datasets. *DM* represents Density-Matched ablation variants

| Model | Accuracy (%) ↑ | | | F&B | | HEALTH | | NYUv2 | | |
|---|---|---|---|---|---|---|---|---|---|---|
| | CIFAR-10 | AV-MNIST | MM-IMDb | Acc. (%) ↑ | MAE ↓ | Acc. (%) ↑ | MAE ↓ | Angle Error ↓ | Seg. MIOU ↑ | Depth RMSE ↓ |
| Uniform | 55 ±2.2 | 63 ±1.5 | 56 ±2.8 | 45 ±2.4 | 0.57 ±0.030 | 52 ±2.0 | 0.54 ±0.024 | 21.6 ±0.27 | 0.059 ±0.003 | 0.73 ±0.018 |
| GradNorm | 61 ±1.6 | 65 ±1.1 | 58 ±2.0 | 47 ±2.3 | 0.57 ±0.020 | 53 ±2.1 | 0.52 ±0.019 | 21.4 ±0.23 | 0.054 ±0.004 | 0.65 ±0.016 |
| MGDA | 59 ±2.9 | 62 ±1.7 | 56 ±3.3 | 44 ±3.0 | 0.57 ±0.036 | 53 ±2.5 | 0.53 ±0.030 | 21.8 ±0.33 | 0.063 ±0.005 | 0.75 ±0.024 |
| PCGrad | 61 ±1.9 | 65 ±1.3 | 58 ±2.3 | 50 ±2.1 | 0.55 ±0.024 | 58 ±2.0 | 0.48 ±0.021 | 20.9 ±0.24 | 0.070 ±0.004 | 0.69 ±0.013 |
| CAGrad | 59 ±2.0 | 62 ±1.1 | 57 ±2.5 | 46 ±2.5 | 0.58 ±0.031 | 53 ±1.9 | 0.52 ±0.024 | 21.9 ±0.29 | 0.065 ±0.004 | 0.73 ±0.018 |
| AdaTask | 63 ±1.5 | 67 ±0.9 | 59 ±1.9 | 47 ±1.9 | 0.59 ±0.026 | 55 ±2.2 | 0.52 ±0.024 | 20.3 ±0.23 | 0.069 ±0.004 | 0.65 ±0.015 |
| FAMO | 64 ±1.2 | 70 ±1.0 | 61 ±1.6 | 52 ±2.0 | 0.53 ±0.021 | 60 ±1.8 | 0.49 ±0.018 | 19.9 ±0.19 | 0.074 ±0.003 | 0.63 ±0.012 |
| FairGrad | 62 ±1.8 | 66 ±1.3 | 59 ±2.5 | 52 ±2.5 | 0.54 ±0.026 | 60 ±2.0 | 0.47 ±0.022 | 20.7 ±0.27 | 0.072 ±0.004 | 0.67 ±0.015 |
| Nash-MTL | 63 ±1.9 | 66 ±1.2 | 60 ±2.1 | 52 ±2.3 | 0.54 ±0.024 | 60 ±2.3 | 0.47 ±0.023 | 20.6 ±0.24 | 0.073 ±0.004 | 0.67 ±0.013 |
| Static One-Shot | 61 ±2.0 | 66 ±1.1 | 58 ±2.6 | 48 ±2.3 | 0.56 ±0.027 | 54 ±2.1 | 0.51 ±0.025 | 20.5 ±0.25 | 0.071 ±0.004 | 0.65 ±0.016 |
| Single-Step | 40 ±4.2 | 59 ±2.4 | 20 ±5.4 | 42 ±3.9 | 0.60 ±0.041 | 47 ±3.5 | 0.55 ±0.034 | 26.4 ±0.55 | 0.042 ±0.006 | 0.81 ±0.029 |
| SON-GOKU (Threshold, DM) | 63 | 68 | 59 | 49 | 0.55 | 56 | 0.51 | 20.6 | 0.071 | 0.61 |
| SON-GOKU (kNN-Symm.) | 60 | 65 | 55 | 46 | 0.57 | 52 | 0.53 | 22.1 | 0.066 | 0.70 |
| SON-GOKU (kNN-Symm., DM) | 61 | 66 | 57 | 47 | 0.56 | 54 | 0.52 | 21.4 | 0.068 | 0.66 |
| SON-GOKU (Signed-only) | 56 | 63 | 52 | 43 | 0.60 | 50 | 0.56 | 24.0 | 0.053 | 0.76 |
| SON-GOKU (Signed-only, DM) | 58 | 64 | 54 | 45 | 0.59 | 52 | 0.54 | 23.0 | 0.056 | 0.73 |
| SON-GOKU (Quantile) | 64 | 68 | 60 | 50 | 0.54 | 57 | 0.50 | 20.3 | 0.072 | 0.60 |
| SON-GOKU (Quantile, DM) | 65 | 69 | 61 | 51 | 0.53 | 58 | 0.50 | 20.0 | 0.072 | **0.59** |
| SON-GOKU + GradNorm | 62 ±1.4 | 69 ±1.0 | 59 ±1.7 | 51 ±1.8 | 0.53 ±0.022 | 59 ±1.7 | 0.49 ±0.018 | 19.6 ±0.19 | 0.073 ±0.003 | 0.64 ±0.011 |
| SON-GOKU + AdaTask | **67** ±1.2 | **71** ±0.9 | **63** ±1.6 | 52 ±1.7 | 0.53 ±0.021 | 59 ±1.8 | 0.48 ±0.017 | 20.1 ±0.20 | 0.068 ±0.004 | 0.67 ±0.013 |
| SON-GOKU + PCGrad | 65 ±1.3 | 70 ±0.9 | 60 ±1.8 | **54** ±2.0 | **0.52** ±0.024 | **62** ±1.6 | **0.45** ±0.020 | 19.7 ±0.18 | **0.076** ±0.003 | 0.62 ±0.010 |
| SON-GOKU | 65 ±1.5 | 69 ±1.0 | 61 ±1.8 | 51 ±1.9 | 0.53 ±0.023 | 58 ±1.7 | 0.50 ±0.018 | 19.8 ±0.20 | 0.073 ±0.004 | **0.59** ±0.012 |

### 6.3.1 SINGLE-STEP CONFLICT ESTIMATION

Here, we set the history length to $H = 1$, so every recoloring step relies on only the most recent mini-batch gradients to estimate interference. Without aggregation over many past steps, the conflict graph should become highly noisy, causing unstable task groupings from one update window to the next. This variant tests the importance of historical conflict statistics in the scheduler.

## 7 RESULTS AND DISCUSSION

Results for all models across every experiment are depicted in Table 1. All metrics are held-out test results under identical training setups and architectures. Across ten metrics on six datasets, our conflict-aware schedulers consistently match or exceed all baseline methods.

### 7.1 OVERALL PERFORMANCE IMPROVEMENTS

Overall, the conflict-aware approaches improve over the uniform baseline by 10%-20% on CIFAR-10 and by 7% on MM-IMDb, indicating that grouping tasks according to measured interference is more effective than treating all tasks equally at every update. On NYUv2, we see similar improvements across all the metrics. These results suggest that the scheduler's graph coloring cleanly separates high-conflict tasks, preserving the projection or LR-balancing advantages (stemming from PCGrad's gradient projection and AdaTask's learning-rate adaptation, respectively) while removing residual interference (see App. S for grouping patterns at training time and more analyses). As we evaluated across diverse tasks and datasets, our results also demonstrate clear improvements in generalization.

### 7.2 ABLATION STUDY ON SCHEDULER DESIGN

We evaluate nine controlled ablations of six types: (i) **Static One-Shot Coloring**, which runs greedy graph coloring once at the start of training and then freezes the groups, testing dependence on *dynamic* recoloring as gradients change; (ii) **Single-Step Conflict Estimation**, which sets the history length to $H = 1$ so each recoloring uses only the most recent mini batch, testing the importance of averaging conflict statistics over time; (iii) **Threshold Graph (baseline)**, which connects tasks $i$ and $j$ when the smoothed cosine $\hat{s}_{ij}(t)$ falls below a global threshold $-\tau(t)$; (iv) **kNN-Symmetric Graph**, which connects each task to its $m$ most conflicting neighbors and then symmetrizes the edges, enforcing roughly fixed degree per task and comparing local degree control against the global threshold rule; (v) **Signed-Only Graph**, which adds an edge only if $\hat{s}_{ij}(t) < 0$, yielding a very sparse graph and ignoring moderate (but potentially harmful) conflicts; and (vi) **Quantile Threshold Graph**, which at each refresh sets $\tau(t)$ so that only the worst $p\%$ of cosine values are treated as conflicting, keeping edge density approximately stable and testing an adaptive cutoff versus a fixed global threshold. We evaluate each graph rule under two settings. In the *fixed* $\tau$ setting, all rules share the same $\tau(t)$

schedule used in the main experiments. In the *density-matched* setting, we adjust the hyperparameters of each rule so that all graphs have approximately the same edge density at each refresh. This isolates the effect of which pairs are marked as conflicting, rather than how many edges are present. We go into much further detail regarding the ablation in Appendix K.3.

These ablations directly test the assumptions behind SON-GOKU. Static One-Shot, which freezes groups, consistently underperforms the full scheduler on most metrics, indicating that task relations change enough during training that dynamic recoloring is needed to maintain $\tau$-compatibility as gradients drift (Sections 5.1–5.2). Single-Step, which uses $H = 1$, is clearly worse across datasets, matching our claim that batch cosines are too noisy. Instead, averaging conflict statistics over short history windows provides the clean information needed for accurate graph recovery (Section 5.4). Among graph constructions, simple threshold and quantile rules (and their density-matched variants) perform similarly well, suggesting that any approach that reliably isolates the worst conflicting pairs is sufficient. In contrast, Signed-Only and kNN-Symmetric, which ignore conflict magnitude or have purely local degree control, degrade performance more noticeably, especially on NYUv2 and the tabular benchmarks. Overall, the best performing configurations are precisely those that match the descent and recovery conditions analyzed in Sections 5.1–5.2 and 5.4.

### 7.3 ADDITIONAL ANALYSIS

#### 7.3.1 OPTIMIZER-TASK ALIGNMENT

Interestingly, we observe that AdaTask-based approaches tend to be the best on classification tasks (CIFAR-10, AV-MNIST, MM-IMDb) while PCGrad-based approaches tend to be the best on tasks that model regression (NYUv2).

We believe that this stems from unique differences in the features of classification and regression-based models. For example, cross-entropy gradients near decision boundaries tend to be bursty and high in variance (Shrivastava et al., 2016; Lin et al., 2017; Hoffer et al., 2017). By scaling each task's step size according to its running gradient norm, AdaTask smooths out these spikes.

On the other hand, we believe that PCGrad under the scheduler performs particularly well on regression and dense-prediction tasks as their tasks tend to generate smooth, large-magnitude gradients whose directions change gradually. PCGrad removes only the small component of the gradient that conflicts across tasks, preserving the main descent direction while reducing interference.

#### 7.3.2 SYNERGY BETWEEN SCHEDULING AND BASELINES

We believe that the superior results found in the combinations of the scheduler and baseline models can be traced to the way scheduling and optimization reinforce one another.

First, greedy graph coloring partitions tasks into $\tau$-compatible groups, segregating tasks with highly divergent gradients. This yields a guaranteed lower bound on descent (Proposition 6), directly improving optimization efficiency.

Within each low-conflict group, the optimizer can do its job under more ideal conditions. PCGrad can remove the remaining minor conflicting components, preserving the majority of the descent direction. AdaTask can adjust each task's learning rate without being impacted by large adversarial gradients.

This $\Delta + 1$ color bound ensures that every task is scheduled at least once per period. This prevents tasks from being essentially starved of updates.

Finally, by computing interference over a window, the scheduler smooths out gradient fluctuations. This prevents the erratic schedule changes that projection-only grouping methods have been shown to face (Yu et al., 2020; Shi et al., 2023; Zhang et al., 2024), thereby better stabilizing convergence.

#### 7.3.3 SON-GOKU'S ABILITY TO CREATE GENERALIZABLE MODELS

While our guarantees in Section 5 and Appendices B–F are stated in optimization terms, they also directly increase gradient coherence and limit destructive interference in ways that are known to favor generalization to unseen data. Section 5.1 shows that the aggregated group gradient remains aligned with descent and that intra-group gradient conflict is explicitly limited by $\tau$ and $|S_t|$. Section 5.3 then

Table 2: Wall-clock time (seconds $\pm$ standard deviation) vs. number of tasks $K$.

| Method (R if applicable) | K=3 | K=6 | K=16 | K=40 |
|---|---|---|---|---|
| Uniform | $0.2656 \pm 0.1201$ | $0.3240 \pm 0.0629$ | $0.3798 \pm 0.1050$ | $0.4054 \pm 0.1190$ |
| GradNorm | $5.4714 \pm 0.7137$ | $5.1201 \pm 0.6112$ | $4.9042 \pm 0.5869$ | $4.7372 \pm 0.9286$ |
| AdaTask | $2.1816 \pm 0.0934$ | $2.1032 \pm 0.1012$ | $2.2853 \pm 0.0718$ | $2.2278 \pm 0.1370$ |
| PCGrad | $3.6212 \pm 0.3517$ | $23.1266 \pm 0.8773$ | $176.7566 \pm 2.8171$ | $1127.1337 \pm 34.2603$ |
| MGDA | $97.1081 \pm 5.4645$ | $121.4371 \pm 9.0923$ | $132.4913 \pm 3.1752$ | $134.0878 \pm 2.2621$ |
| FAMO | $2.0725 \pm 0.2073$ | $1.9980 \pm 0.1998$ | $2.1710 \pm 0.2171$ | $2.1164 \pm 0.2116$ |
| FairGrad | $3.8020 \pm 0.5703$ | $15.2079 \pm 2.2812$ | $108.1450 \pm 16.2218$ | $675.9065 \pm 101.3860$ |
| Nash-MTL | $5.7030 \pm 1.1406$ | $22.8118 \pm 4.5624$ | $162.2176 \pm 32.4435$ | $1013.8598 \pm 202.7720$ |
| SON-GOKU ($R = 32$) | $1.9896 \pm 0.3651$ | $3.3202 \pm 0.5745$ | $6.0897 \pm 0.9425$ | $12.1432 \pm 1.2044$ |
| SON-GOKU + AdaTask ($R = 32$) | $3.7718 \pm 0.9654$ | $5.0511 \pm 0.6531$ | $7.5903 \pm 1.1920$ | $14.5182 \pm 2.0660$ |
| SON-GOKU + GradNorm ($R = 32$) | $7.0202 \pm 1.0711$ | $8.1661 \pm 0.9355$ | $10.7227 \pm 2.2088$ | $16.5760 \pm 1.8418$ |
| SON-GOKU + PCGrad ($R = 32$) | $1.9834 \pm 0.3586$ | $3.4971 \pm 0.3840$ | $6.1395 \pm 0.9425$ | $10.9097 \pm 1.5263$ |

compares two ways to apply the same gradients during a refresh, either a single mixed update or a scheduled sequence of group updates. Together, these analyses imply that each step in SON-GOKU provides more informative signals and less interference, or, equivalently, a higher gradient-to-noise ratio (Sun et al., 2023; Fan et al., 2023; McCandlish et al., 2018). Building on this, Section 5.4 shows that SON-GOKU's estimated conflict graph recovers the population structure with high probability, so the schedule repeatedly updates clusters of related tasks rather than conflicting tasks. By enforcing positive affinity within groups, SON-GOKU is able to train related tasks together. This enables effective sharing of model parameters across different tasks, reducing the complexity of the model and increasing sample efficiency (Caruana, 1997; Argyriou et al., 2007; Vandenhende et al., 2022). With this alongside a high gradient-to-noise signal ratio, SON-GOKU theoretically generalizes across many different datasets, domains, and distributions and can perform well even under non-ideal conditions (e.g., noisy labels, class or task imbalance, distribution shift, etc.) (Michalkiewicz et al., 2023). Our ablation results (Table 1) demonstrate that variants without dynamic recoloring or history averaging perform worse, indicating accurate and low-conflict grouping is essential.

### 7.4 SPEED AND TRADEOFFS

SON-GOKU has a time complexity of $O\big(Kr(d + K)/R\big)$ (Section 4.5) amortized per training step (Section 4.5). Table 2 shows near-linear growth over this range of $K$ at $R$=32, reflecting sparsity in the graphs and batched cosine computation. SON-GOKU's time rises from around 2 seconds ($K = 3$) to 12 seconds ($K = 40$), remaining far below methods that perform heavy conflict handling. For example, PCGrad, FairGrad, and Nash-MTL increase steeply with $K$. In contrast, FAMO and AdaTask are among the fastest and largely flat with $K$, as expected from their constant overhead.

SON-GOKU is also memory efficient, with an only incremental memory footprint that scales with the number of tasks $K$, not the parameter dimension $d$. The scheduler's peak memory during a refresh step is $O(K^2+Kr)$ and the persistent state between refreshes is $O(K^2)$ (see Appendix N for further theoretical and experimental analysis). By contrast, methods that retain $K$ full gradients require $O(Kd)$ additional memory. This implies that, on larger backbones (high $d$), SON-GOKU's memory overhead is modest and grows mainly with the task count $K$, rather than with model size.

These contrasts demonstrate the tradeoffs between speed and fidelity to task interference. Faster methods like FAMO minimize overhead, while methods that model conflicts can improve accuracy. These tradeoffs have to be assessed on a case-by-case basis, based on values that factor into each approach's time complexity and the importance of training speed versus performance.

## 8 CONCLUSION

We introduced SON-GOKU, an interference-aware scheduler that estimates cross-task alignment, builds a sparse conflict graph, and greedily colors it to activate one low-conflict group per step. Formally, we provide rigorous theoretical guarantees that justify the design and effectiveness of the scheduler. Empirically, across six benchmarks, SON-GOKU improves over strong baselines and recent approaches. It complements optimizers like PCGrad and AdaTask, indicating that scheduling and gradient shaping are synergistic. By modeling task interactions with a conflict graph and schedule, SON-GOKU offers a simple, scalable, and theory-backed mechanism for robust multitask training.

## 9 REPRODUCIBILITY STATEMENT

We provide a clean code repository for reproducibility in the supplementary materials, and this is also provided in an online (de-identified) Git repository. The scripts in this repository contain functionality for downloading, loading, and preprocessing all datasets used in training. The code also includes implementations for SON-GOKU and all 10 of its ablations. We provide clear and easy-to-use training scripts with pre-configured parameters, allowing for reproduction of the exact experiments used across all datasets. We provide further details regarding empirical experiments and evaluation in Appendices J–L. To make our theoretical analysis easier to follow and more transparent, we provide highly detailed descriptions of assumptions, propositions, and proofs that could not fit in the main text in Appendices B–I. Furthermore, to make effective real-world deployment easier, we provide practical guidance regarding SON-GOKU in Appendices D, N, and O.

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

## A    FULL ALGORITHM BLOCK FOR PROPOSED APPROACH

---

**Algorithm 1** SON-GOKU Scheduler

---

**Require:** Initial shared params $\theta_0$, heads $\{\phi_k\}_{k=1}^{K}$, EMA buffers $\tilde{g}_k^{(0)} = 0$, total steps $T$, learning-rate
  schedule $\{\eta_t\}$, refresh length $R$, warm-up $T_{\text{warm}}$, target threshold $\tau^*$, minimum coverage $f_{\min}$,
  EMA parameter $\beta$
 1: *Gradients follow the weighted-loss convention (Sec. 4).*
 2: $r \leftarrow 0, t_r \leftarrow 0$                ▷ current refresh round and start index
 3: $\tau \leftarrow 1; m_0 \leftarrow 1; C_1^{(0)} \leftarrow \{1, \ldots, K\}$          ▷ warm-start schedule
 4: **for** $t = 0, \ldots, T - 1$ **do**
 5:   **Warm-up/Anneal:** $\tau \leftarrow \text{ANNEAL}(t)$         ▷ approach in Sec. 4.4
 6:   **Scheduling:** $S_t \leftarrow C_{(t \bmod m_r)+1}^{(r)}$
 7:   **Forward/Backward:**
 8:   **for all** $k \in S_t$ **do**
 9:    compute per-task gradients $g_k^{(t)}$ and $h_k^{(t)}$ (defs: Sec. 4.1)
 10:   **end for**
 11:   **Parameter update (shared):** $\theta_{t+1} \leftarrow \theta_t - \eta_t \sum_{k \in S_t} g_k^{(t)}$
 12:   **Parameter update (task-specific):**
 13:   **for all** $k \in S_t$ **do**
 14:    $\phi_{k,t+1} \leftarrow \phi_{k,t} - \eta_t h_k^{(t)}$
 15:   **end for**
 16:   **EMA:**
 17:   **for all** $k \in S_t$ **do**
 18:    update $\tilde{g}_k^{(t+1)}$ (Eq. 8)
 19:   **end for**
 20:   **if** $(t + 1) \bmod R = 0$ **then**             ▷ refresh
 21:    **EMA refresh:** update all $\tilde{g}_i$ using small mini-batches (Sec. 4.1)
 22:    **Interference matrix:** compute $\rho_{ij}^{(t+1)}$ via Eq. 9
 23:    **Conflict graph:** build $G_\tau^{(r+1)}$ via Eq. 10
 24:    **Greedy coloring:** Welsh–Powell $\rightarrow \{C_1^{(r+1)}, \ldots, C_{m_{r+1}}^{(r+1)}\}$
 25:    **Minimum coverage:** enforce $f_i \geq f_{\min}$ using compatible-slot duplication (Sec. 4.4.1 )
 26:    $r \leftarrow r + 1; t_r \leftarrow t + 1$
 27:   **end if**
 28: **end for**

---

Algorithm block 1 provides an overview of the SON-GOKU scheduler. At a high level, the procedure
consists of four stages: (1) estimating pairwise interference, (2) building and coloring the conflict
graph, (3) generating a periodic schedule, and (4) updating that schedule as training evolves.

## B    EXACT RECOVERY OF POPULATION CONFLICT GRAPH & TASK PARTITION

### B.1    SETTING, DEFINITIONS, AND POPULATION OBJECTS

Let $K \geq 2$ be the number of tasks and $d \geq 1$ the parameter dimension. At designated refresh
iterations, the scheduler:

 (i) computes a per-task exponential moving average (EMA) of stochastic gradients over a probe
   window of $R$ iterations,

 (ii) forms a cosine-similarity matrix from the $K$ EMA vectors,

 (iii) builds a conflict graph by thresholding negative cosines at a fixed level $-\tau$ with $\tau \in (0, 1)$,

 (iv) computes a proper coloring of the conflict graph, and

 (v) schedules one color class per iteration until the next refresh

**Definition B.1.** *At the beginning of a refresh window (i.e., at a fixed iterate $\theta$), let*

$$\mu_i \in \mathbb{R}^d \qquad (i = 1, \ldots, K) \tag{14}$$

*denote the population task gradients (or the window-stationary means). Define the population cosine matrix $C^\star \in [-1, 1]^{K \times K}$ by*

$$C_{ij}^\star = \frac{\langle \mu_i, \mu_j \rangle}{\|\mu_i\| \|\mu_j\|}, \qquad i \neq j, \quad C_{ii}^\star = 1. \tag{15}$$

**Definition B.2.** *Fix $\tau \in (0, 1)$. The population conflict graph $G^\star = (V, E^\star)$ on vertex set $V = \{1, \ldots, K\}$ has an edge $\{i, j\}$ iff $C_{ij}^\star < -\tau$. The true grouping $\mathcal{P}^\star$ is one of:*

(A) *Component Model: the vertex partition given by the connected components of $G^\star$.*

(B) *Multipartite model: a partition $V = \bigsqcup_{r=1}^m P_r$ (with $m \geq !$) such that $G^\star$ is the complete $m$-partite graph induced by $\{P_r\}_{r=1}^m$ (no edges within any $P_r$, all cross-part edges present)*

*When we later speak of group recovery, we mean equality of the empirical partition (defined from data) with $\mathcal{P}^\star$, up to label permutation in case (B).*

## B.2 ASSUMPTIONS

We adopt the following assumptions, which are standard in analyses of stochastic-gradient methods and verifiable in practice (see, e.g., Robbins & Monro 1951; Kushner & Yin 2003; Nemirovski et al. 2009; Bottou et al. 2018; Wainwright 2019; for concentration of geometrically weighted and mixing sequences, see Merlevède et al. 2011; De la Pena et al. 2009).

**Assumption 1** (Separation margin around the threshold). *There exists $\gamma \in (0, 1 - \tau)$ such that for all $i \neq j$:*

$$\begin{cases} C_{ij}^\star \leq -(\tau + \gamma), & \text{if } i \text{ and } j \text{ lie in different groups of } \mathcal{P}^\star, \\ C_{ij}^\star \geq -(\tau - \gamma), & \text{if } i \text{ and } j \text{ lie in the same group of } \mathcal{P}^\star. \end{cases} \tag{16}$$

**Assumption 2** (Probe noise model and EMA). *In the refresh window of length $R$, the per-iteration stochastic task gradients admit the decomposition*

$$g_{i,t} = \mu_i + \xi_{i,t}, \qquad t = 1, \ldots, R, \tag{17}$$

*where $\{\xi_{i,t}\}_{t=1}^R$ are mean-zero, sub-Gaussian with parameter $\sigma^2$, and satisfy a $\phi$-mixing or martingale-difference condition ensuring concentration with geometric weights. The EMA for task $i$ is*

$$\tilde{g}_i = \sum_{t=1}^R w_t \, g_{i,t}, \qquad w_t = \frac{(1 - \beta) \, \beta^{R-t}}{1 - \beta^R}, \qquad \beta \in [0, 1). \tag{18}$$

*Define the effective sample size $n_{\text{eff}}$ by*

$$n_{\text{eff}}^{-1} := \sum_{t=1}^R w_t^2 = \frac{(1 - \beta)^2 (1 - \beta^{2R})}{(1 - \beta^R)^2 (1 - \beta^2)}. \tag{19}$$

*In particular, as $R \to \infty$ (with fixed $\beta \in [0, 1)$), we have $n_{\text{eff}} \to \frac{1+\beta}{1-\beta}$.*

**Assumption 3** (Slow drift within a refresh). *Over the refresh window, the changes in $\mu_i$ are small enough to be absorbed in the concentration bounds below (equivalently, one can regard $\mu_i$ as constant within the window by working at the start-of-window iterate and moving any drift into the noise process).*

**Assumption 4** (Minimum norm and task inclusion). *There exists $m_0 > 0$ such that $\|\mu_i\| \geq m_0$ for all tasks included in the graph. In our implementation, we make it so that tasks with $\|\tilde{g}_i\| < \nu$ (for a small $\nu \ll m_0$) are temporarily excluded from graph construction until stabilized.*

**Assumption 5** (Threshold selection). *The threshold $\tau$ is fixed across refreshes or selected using data independent of the probe window used to form $\{\tilde{g}_i\}$ (e.g., via a separate pilot set). The analysis below treats $\tau$ as deterministic with respect to the probe sample.*

### B.3 Deterministic group recovery from the conflict graph

We begin with basic graph-theoretic facts that we will use once we have established that the empirical conflict graph coincides with its population counterpart.

**Proposition 1** (Chromatic number of a complete multipartite graph). *If $G^\star$ is complete $m$-partite with parts $\{P_r\}_{r=1}^m$, then $\chi(G^\star) = m$.*

*Proof.* Picking one vertex from each part yields a clique of size $m$, hence $\chi(G^\star) \geq m$. Coloring each part with a distinct color is proper, hence $\chi(G^\star) \leq m$. Therefore $\chi(G^\star) = m$. □

**Theorem 1** (Identifiability via optimal coloring under model (B)). *Assume model (B), i.e., $G^\star$ is complete $m$-partite with parts $\{P_r\}_{r=1}^m$. Let $c : V \to \{1, \ldots, m\}$ be a proper coloring of $G^\star$ that uses exactly $\chi(G^\star)$ colors. Then each color class equals some part $P_r$ (up to relabeling).*

*Proof.* In a complete multipartite graph, any two vertices from different parts are adjacent. Thus, no color class can contain vertices from two different parts, so each color class is contained in some $P_r$. By Proposition 1, $\chi(G^\star) = m$, so any optimal coloring uses exactly $m$ colors. Since there are $m$ nonempty parts, none can be split across two colors. Hence, the color classes coincide with $\{P_r\}_{r=1}^m$ up to permutation. □

**Proposition 2** (Identifiability via components under model (A)). *Under model (A), the grouping $\mathcal{P}^\star$ equals the connected components of $G^\star$. Consequently, any procedure that returns the connected components of the empirical graph recovers $\mathcal{P}^\star$ whenever the empirical graph equals $G^\star$.*

### B.4 Uniform control of empirical cosines from EMA gradients

We now quantify the deviation of the empirical cosine matrix $\widehat{C}$ formed from $\{\tilde{g}_i\}$ relative to $C^\star$.

**Lemma 1** (EMA vector concentration in directions of interest). *Assume Assumption 2 and Assumption 3. There exists a constant $c > 0$ depending only on the mixing parameters such that for any fixed unit vector $u \in \mathbb{S}^{d-1}$ and any $\varepsilon > 0$.*

$$\Pr\Big( \big|\langle \tilde{g}_i - \mu_i, \, u \rangle\big| > \varepsilon \Big) \leq 2\exp\Big( -c\, n_{\text{eff}}\, \varepsilon^2/\sigma^2 \Big). \tag{20}$$

*In particular, for any finite set of unit vectors $\{u_j\}_{j=1}^M$, a union bound yields*

$$\Pr\Big( \max_{1 \leq j \leq M} \big|\langle \tilde{g}_i - \mu_i, \, u_j \rangle\big| > \varepsilon \Big) \leq 2M\exp\Big( -c\, n_{\text{eff}}\, \varepsilon^2/\sigma^2 \Big). \tag{21}$$

*Proof.* The scalar process $\{\langle \xi_{i,t}, u \rangle\}_{t=1}^R$ is sub-Gaussian with variance proxy $\sigma^2$ and satisfies the same mixing condition. Exponential-weighted averages of such sequences obey Hoeffding-Azuma/Berstein-type tail bounds with variance proxy $\sigma^2 \sum_t w_t^2 = \sigma^2/n_{\text{eff}}$. The stated inequality follows. □

**Lemma 2** (Cosine stability under perturbations). *Assume Assumption 4 and let $\epsilon > 0$. If for a pair $(i, j)$ we have*

$$\Big|\langle \tilde{g}_i - \mu_i, \, \tfrac{\mu_j}{\|\mu_j\|} \rangle\Big| \leq \epsilon, \qquad \Big|\langle \tilde{g}_j - \mu_j, \, \tfrac{\mu_i}{\|\mu_i\|} \rangle\Big| \leq \epsilon, \qquad \Big|\langle \tilde{g}_i - \mu_i, \, \tfrac{\mu_i}{\|\mu_i\|} \rangle\Big| \leq \epsilon, \qquad \Big|\langle \tilde{g}_j - \mu_j, \, \tfrac{\mu_j}{\|\mu_j\|} \rangle\Big| \leq \epsilon, \tag{22}$$

*then*

$$\left|\widehat{C}_{ij} - C_{ij}^{\star}\right| \leq \frac{6\,\epsilon}{m_0} + \frac{4\,\epsilon^2}{m_0^2}. \tag{23}$$

*Proof.* Write $\tilde{g}_i = \mu_i + \delta_i$, $\tilde{g}_j = \mu_j + \delta_j$. Decompose the numerator and denominator in the cosine:

$$\langle \tilde{g}_i, \tilde{g}_j \rangle - \langle \mu_i, \mu_j \rangle = \langle \delta_i, \mu_j \rangle + \langle \mu_i, \delta_j \rangle + \langle \delta_i, \delta_j \rangle, \tag{24}$$

and

$$\|\tilde{g}_i\| = \|\mu_i\| \sqrt{1 + 2\langle \delta_i, \mu_i \rangle / \|\mu_i\|^2 + \|\delta_i\|^2 / \|\mu_i\|^2} \tag{25}$$

Using Assumption 4,

$$|\langle \delta_i, \mu_j / \|\mu_j\| \rangle| \leq \epsilon \tag{26}$$

and

$$|\langle \delta_i, \mu_i / \|\mu_i\| \rangle| \leq \epsilon \tag{27}$$

imply

$$|\langle \delta_i, \mu_j \rangle| \leq \epsilon \|\mu_j\| \tag{28}$$

and

$$|\langle \delta_i, \mu_i \rangle| \leq \epsilon \|\mu_i\| \tag{29}$$

A second-order expansion of the cosine in $(\delta_i, \delta_j)$ with the above controls yields the bound. The constants 6 and 4 arise from collecting the linear and quadratic contributions in $\epsilon/m_0$. □

Combining Lemma 1 and Lemma 2 with a union bound over all unordered pairs $(i, j)$ shows that the empirical cosines are uniformly close to their population counterparts.

**Proposition 3** (Uniform cosine accuracy with high probability). *Assume Assumption 2, Assumption 3, and Assumption 4. For any $\epsilon > 0$ there exist absolute constants $c, C > 0$ such that if*

$$n_{\text{eff}} \geq C\,\frac{\sigma^2}{m_0^2\,\epsilon^2} \tag{30}$$

*then, with probability $1 - \delta$,*

$$\max_{i<j} \left|\widehat{C}_{ij} - C_{ij}^{\star}\right| \leq \epsilon \tag{31}$$

*Proof.* For each unordered pair $(i, j)$, apply Lemma 1 with the four unit vectors $\mu_j / \|\mu_j\|$, $\mu_i / \|\mu_i\|$, and use Lemma 2 to convert these directional deviations into a cosine deviation bound. A union bound over the $O(K^2)$ pairs yields the claimed logarithmic factor. The constants absorb the quadratic term in $\epsilon$ by requiring $\epsilon \leq m_0$. □

## B.5 EXACT EDGE RECOVERY AND GROUP RECOVERY

We first show that a uniform cosine error smaller than the margin $\gamma$ implies exact equality of empirical and population conflict graphs.

**Theorem 2** (Exact conflict-graph recovery under the margin). *Assume Assumptions 1–5. If*

$$\max_{i<j} \left| \widehat{C}_{ij} - C_{ij}^\star \right| \le \epsilon \qquad with \qquad \epsilon < \gamma, \tag{32}$$

*then the empirical conflict graph equals the population graph:*

$$\widehat{G} = G^\star. \tag{33}$$

*Equivalently, for every $i \ne j$,*

$$C_{ij}^\star \le -(\tau + \gamma) \;\Rightarrow\; \widehat{C}_{ij} < -\tau \quad and \quad C_{ij}^\star \ge -(\tau - \gamma) \;\Rightarrow\; \widehat{C}_{ij} > -\tau. \tag{34}$$

*Proof.* For any pair $(i, j)$, if $C_{ij}^\star \le -(\tau + \gamma)$, then $\widehat{C}_{ij} \le -(\tau + \gamma) + \epsilon < -\tau$, hence $\{i, j\} \in \widehat{E}$. If $C_{ij}^\star \ge -(\tau - \gamma)$, then $\widehat{C}_{ij} \ge -(\tau - \gamma) - \epsilon > -\tau$, hence $\{i, j\} \notin \widehat{E}$. $\qquad\square$

Combining Proposition 3 and Theorem 2 yields a high-probability statement.

**Corollary 1** (High-probability exact recovery of $G^\star$). *Under Assumptions 1–5, there exists a universal constant $C > 0$ such that if*

$$n_{\text{eff}} \ge C \frac{\sigma^2}{m_0^2 \gamma^2} \log\left(\frac{K^2}{\delta}\right), \tag{35}$$

*then $\Pr(\widehat{G} = G^\star) \ge 1 - \delta$.*

**Theorem 3** (Group recovery under the component model). *Under model (A) and the conditions of Corollary 1, with probability at least $1 - \delta$, the connected components of $\widehat{G}$ equal $\mathcal{P}^\star$.*

*Proof.* Immediate from $\widehat{G} = G^\star$ and the definition of $\mathcal{P}^\star$. $\qquad\square$

**Theorem 4** (Group recovery under the multipartite model). *Under model (B) and the conditions of Corollary 1, with probability at least $1 - \delta$, $\chi(\widehat{G}) = m$ and any optimal coloring of $\widehat{G}$ yields color classes equal to $\{P_r\}_{r=1}^m$ up to label permutation.*

*Proof.* If $\widehat{G} = G^\star$, then $\widehat{G}$ is complete $m$-partite. Proposition 1 gives $\chi(\widehat{G}) = m$. Theorem 1 implies identifiability up to permutation by any optimal coloring. $\qquad\square$

## B.6 QUANTITATIVE PROBE-BUDGET REQUIREMENT

Combining the bounds above yields the following sample-complexity statement.

**Corollary 2.** *Under assumptions 1–5, there exist absolute constants $c, C > 0$ such that the following holds. If the EMA parameters $(R, \beta)$ are chosen to ensure*

$$n_{\text{eff}} \ge C \frac{\sigma^2}{m_0^2 \gamma^2} \qquad \left( equivalently, \quad \sum_{t=1}^R w_t^2 \le c \frac{m_0^2 \gamma^2}{\sigma^2} \frac{1}{\log(K/\delta)} \right) \tag{36}$$

*then $\Pr(\widehat{G} = G^\star) \ge 1 - \delta$, and consequently Theorems 3–4 apply. In particular, for fixed $\beta$ and large $R$, $n_{\text{eff}} \to \frac{1+\beta}{1-\beta}$ (i.e., it saturates). Thus, to meet the required budget as $K$ grows, one increases $n_{\text{eff}}$ by choosing $\beta$ closer to 1 (e.g., $1 - \beta \asymp 1/\log(K^2/\delta)$), or by switching to a unnormalized averaging approach.*

## B.7 SUMMARY OF THE RECOVERY ARGUMENT

We summarize the logical flow leading to consistency of the scheduler.

(i) Assumptions: Assumptions 1–5 define the conditions in which in which across-group population cosines lie below $-(\tau + \gamma)$, within-group cosines lie above $-(\tau - \gamma)$, EMA gradients concentrate with effective sample size $n_{\text{eff}}$, and all included tasks have non-negligible gradient norm.

(ii) Uniform cosine accuracy: Lemmas 1–2 together with Proposition 3 yield a high-probability uniform cosine approximation:

$$\max_{i<j}\bigl|\widehat{C}_{ij} - C^{\star}_{ij}\bigr| \ \leq \ \epsilon, \tag{37}$$

with probability at least $1 - \delta$, where $\epsilon$ decreases as $n_{\text{eff}}$ increases.

(iii) Exact recovery of edges: If the approximation tolerance satisfies $\epsilon < \gamma$, Theorem 2 converts the uniform bound into exact edge recovery of the conflict graph:

$$\widehat{G} \ = \ G^{\star}. \tag{38}$$

(iv) Recovery of the grouping: Given $\widehat{G} = G^{\star}$, Theorem 3 implies group recovery under the component model (groups are the connected components). Under the multipartite model, Proposition 1 and Theorem 1 yield $\chi(\widehat{G}) = m$ and Theorem 4 shows that any optimal coloring returns the true parts (up to label permutation).

**Quantitative consequence.** Assume Assumptions 1–5 and fix $\delta \in (0, 1)$. Let $m_0 = \min_i \|\mu_i\|$ and let $\sigma^2$ be the variance proxy from Assumption 2. If the EMA probe budget satisfies

$$n_{\text{eff}} \ \geq \ C \, \frac{\sigma^2}{m_0^2 \, \gamma^2} \, \log\!\Bigl(\frac{K^2}{\delta}\Bigr) \tag{39}$$

for a universal constant $C > 0$, then with probability at least $1 - \delta$ the empirical conflict graph equals the population graph: $\widehat{G} = G^{\star}$. Consequently:

(i) under the component model (A), the connected components of $\widehat{G}$ coincide with $\mathcal{P}^{\star}$.

(ii) under the multipartite model (B), $\chi(\widehat{G}) = m$ and any optimal coloring of $\widehat{G}$ recovers $\mathcal{P}^{\star}$ up to permutation of labels.

## C  DESCENT BOUNDS FOR SCHEDULED VERSUS AGGREGATED UPDATES

We compare two update procedures over a single refresh: a scheduled sequence of per-group steps (i.e., the approach we propose in our paper) and a single aggregated step that combines all groups at once. Both use the same step size $\eta$ and the same gradient information measured at the start of the refresh, and our analysis operates at the level of $L$-smooth (descent) upper bounds. We identify when the scheduled bound is strictly tighter and summarize implications under PL / strong convexity.

Throughout, $F : \mathbb{R}^d \to \mathbb{R}$ is differentiable and $L$-smooth, i.e.

$$F(y) \ \leq \ F(x) + \langle \nabla F(x), y - x \rangle + \tfrac{L}{2}|y - x|^2, \quad \forall x, y. \tag{40}$$

We write $\nabla F(x) = \sum_{r=1}^{m} G_r(x)$, where each $G_r(x)$ is the group gradient for color $r$ (any fixed linear aggregator of task gradients assigned to color $r$ for the current refresh). We use a refresh step size $\eta \in (0, 1/L]$.

## C.1 SINGLE REFRESH BASELINES AND NOTATION

### C.1.1 SINGLE AGGREGATED STEP

**Definition C.1** (Aggregated step). *Starting from the same point $x$, with step size $\eta \in (0, 1/L]$ and group gradients $G_r^0 := G_r(x)$ (with $\nabla F(x) = \sum_{r=1}^m G_r^0$), define*

$$x^{\mathrm{agg}} := x - \eta \sum_{r=1}^m G_r^0. \tag{41}$$

**One-shot $L$-smoothness bound.** Applying $L$-smoothness with $y = x^{\mathrm{agg}}$ yields

$$F(x^{\mathrm{agg}}) \leq F(x) - \eta \Big\langle \nabla F(x), \sum_{r=1}^m G_r^0 \Big\rangle + \frac{L\eta^2}{2} \Big\| \sum_{r=1}^m G_r^0 \Big\|^2. \tag{42}$$

### C.1.2 SCHEDULED GROUP SEQUENCE OVER ONE REFRESH

**Definition C.2** (Scheduled refresh). *Starting from the same point $x$, define*

$$x_0 := x, \qquad x_r := x_{r-1} - \eta\, G_r(x_{r-1}) \quad (r = 1, \ldots, m), \qquad x^{\mathrm{sch}} := x_m. \tag{43}$$

**Order and notation.** The within refresh order $(1, \ldots, m)$ may be fixed or randomly permuted each refresh. We write $H(\cdot)$ for the Hessian of $F$ and take $\eta \in (0, 1/L]$.

Our goal is to compare upper bounds derived from $L$-smoothness for $F(x^{\mathrm{sch}})$ and $F(x^{\mathrm{agg}})$.

## C.2 TELESCOPING BOUND FOR SCHEDULED UPDATES

**Lemma 3** (Smoothness Expansion for Two Scheduled Groups). *Let $m = 2$ and $G_r^0 := G_r(x)$. For any $\eta \in (0, 1/L]$,*

$$F(x^{sch}) \leq F(x) - \eta \langle \nabla F(x), G_1^0 \rangle + \frac{L\eta^2}{2} \|G_1^0\|^2$$
$$- \eta \langle \nabla F(x), G_2(x_1) \rangle + \frac{L\eta^2}{2} \|G_2(x_1)\|^2 + \eta^2 \int_0^1 \langle H(x - t\eta G_1^0)\, G_1^0,\, G_2(x_1) \rangle dt. \tag{44}$$

*Proof sketch.* Apply the $L$-smoothness inequality at the first step to bound $F(x_1)$. For the second step, use $L$-smoothness at $x_1$ and expand

$$\nabla F(x_1) = \nabla F(x) - \int_0^1 H(x - t\eta G_1^0)\, \eta G_1^0\, dt \tag{45}$$

by the fundamental theorem of calculus along the segment $x \to x_1$. $\qquad \square$

### C.2.1 START-OF-REFRESH REDUCTION UNDER PER-GROUP LIPSCHITZNESS

We adopt the following assumption whenever we compare bounds solely in terms of start-of-refresh measurements. It will be used throughout Sections C.3–C.6

**Assumption 6** (Per-group lipschitzness). *Each group map $G_r(\cdot)$ is $L_r$-lipschitz:*

$$\|G_r(u) - G_r(v)\| \leq L_r \|u - v\| \quad \text{for all } u, v. \tag{46}$$

*Under this assumption, for $m = 2$ we have $G_2(x_1) = G_2^0 + \delta_2$ with $\|\delta_2\| \leq L_2 \eta \|G_1^0\|$, hence*

$$\|G_2(x_1)\| \leq \|G_2^0\| + L_2 \eta \|G_1^0\| \tag{47}$$

*For general $m$*

$$\|G_r(x_{r-1})\| \ \leq \ \|G_r^0\| + L_r \, \eta \sum_{p<r} \|G_p^0\| \qquad (r = 2, \ldots, m) \tag{48}$$

*When these substitutions are made in scheduled bounds, the induced drift contributions are collected into a nonnegative penalty $R_m(x; \eta)$*

## C.3 Upper bounds for scheduled and aggregated updates (general $m$)

Applying $L$-smoothness $m$ times yields the scheduled upper bound

$$\begin{aligned}
\mathrm{UB_{sch}}(x; \eta) := F(x) \ - \ \eta \sum_{r=1}^{m} \big\langle \nabla F(x), G_r(x_{r-1}) \big\rangle \ + \ \frac{L\eta^2}{2} \sum_{r=1}^{m} \|G_r(x_{r-1})\|^2 \\
+ \ \eta^2 \sum_{1 \leq p < q \leq m} \int_0^1 \big\langle H\big(x - t\eta G_p(x_{p-1})\big) \, G_p(x_{p-1}), \ G_q(x_{q-1}) \big\rangle \, dt.
\end{aligned} \tag{49}$$

The aggregated upper bound is the one-shot bound from Equation 42, restated as

$$\mathrm{UB_{agg}}(x; \eta) := F(x) \ - \ \eta \Big\langle \nabla F(x), \sum_{r=1}^{m} G_r^0 \Big\rangle \ + \ \frac{L\eta^2}{2} \Big\| \sum_{r=1}^{m} G_r^0 \Big\|^2 \tag{50}$$

The integrals in Equation 49 are over ordered pairs $p < q$ along the specific sequence $x_0 \to x_1 \to \cdots \to x_m$; the bound therefore depends on the within-refresh order. Randomizing the order yields an expected version.

In Sections C.4–C.6 we express the scheduled bound in terms of $\{G_r^0\}$ under the per-group lipschitzness assumption. The associated drift terms are aggregated into $R_m(x; \eta)$.

## C.4 Scheduled and aggregated gap at a common linearization

Define the shorthand

$$I_{pq}(x; \eta) \ := \ \int_0^1 \big\langle H(x - t\eta G_p^0) \, G_p^0, \ G_q^0 \big\rangle \, dt \tag{51}$$

By expanding $\mathrm{UB_{sch}}$ around $\{G_r^0\}$ and collecting the lipschitz drift penalties into $R_m(x; \eta) \geq 0$, we obtain:

**Theorem 5** (Upper-bound gap under per-group lipschitzness)**.** *Assuming per-group lipschitzness, for any partition $\{G_r\}$ and $\eta \in (0, 1/L]$,*

$$\mathrm{UB}_{sch}(x; \eta) - \mathrm{UB}_{agg}(x; \eta) \ \leq \ \eta^2 \sum_{1 \leq p < q \leq m} \Big( - L \, \langle G_p^0, G_q^0 \rangle \ + \ I_{pq}(x; \eta) \Big) \ + \ R_m(x; \eta). \tag{52}$$

*Using $\|H(\cdot)\|_{\mathrm{op}} \leq L$ and Cauchy-Schwarz (Steele, 2004)*

$$I_{pq}(x; \eta) \ \leq \ L \, \|G_p^0\| \, \|G_q^0\| \tag{53}$$

*which gives the envelope*

$$\mathrm{UB}_{sch}(x; \eta) - \mathrm{UB}_{agg}(x; \eta) \ \leq \ L\eta^2 \sum_{p<q} \big( \|G_p^0\| \, \|G_q^0\| - \langle G_p^0, G_q^0 \rangle \big) \ + \ R_m(x; \eta) \ \geq \ 0 \tag{54}$$

**Interpretation** This shows that without additional structure, the scheduled smoothness bound can be looser than the aggregated bound. The gap is governed by Hessian-weighted cross terms $I_{pq}$

**Proposition 4** (Drift penalty bound under per-group lipschitzness). *Assume each group map $G_r$ is $L_r$-lipschitz. Then for $r \geq 2$,*

$$\|G_r(x_{r-1})\| \leq \|G_r^0\| + L_r \eta \sum_{p<r} \|G_p^0\| := \|G_r^0\| + L_r \eta\, S_{r-1}, \tag{55}$$

*and the scheduled start substitution error satisfies*

$$\begin{aligned} R_m(x;\eta) \leq \eta^2 \Big( \sum_{p=1}^m \|G_p^0\| \Big) \sum_{r=2}^m L_r\, S_{r-1} \\ + \frac{L\eta^2}{2} \sum_{r=2}^m \Big( 2\,\|G_r^0\|\, L_r \eta S_{r-1} + (L_r \eta S_{r-1})^2 \Big), \end{aligned} \tag{56}$$

*so $R_m(x;\eta) = O(\eta^2)$ with constants controlled by $\{L_r\}$ and $\{\|G_r^0\|\}$.*

## C.5 SUFFICIENT CONDITIONS FOR A TIGHTER SCHEDULED BOUND

The terms $I_{pq}(x;\eta)$ encode Hessian-weighted interactions between groups and determine when scheduling is advantageous at the bound level.

**Assumption 7** (Hessian-weighted negative cross-terms). *There exist nonnegative margins $\{\Gamma_{pq}\}_{p<q}$ such that*

$$I_{pq}(x;\eta) = \int_0^1 \langle H(x-t\eta G_p^0)\, G_p^0,\, G_q^0\rangle dt \leq -\Gamma_{pq} \|G_p^0\|\, \|G_q^0\| \quad \textit{for all } p < q \tag{57}$$

**Theorem 6** (Strict upper-bound improvement under per-group lipschitzness and negative Hessian-weighted cross-terms). *Assuming per-group lipschitzness and 57, for any $\eta \in (0, 1/L]$,*

$$\mathrm{UB}_{sch}(x;\eta) - \mathrm{UB}_{agg}(x;\eta) \leq \eta^2 \sum_{p<q} \Big( -L\langle G_p^0, G_q^0\rangle - \Gamma_{pq} \|G_p^0\|\, \|G_q^0\| \Big) + R_m(x;\eta) \tag{58}$$

*In particular, if*

$$\sum_{p<q} \Big( \Gamma_{pq} \|G_p^0\|\, \|G_q^0\| + L\langle G_p^0, G_q^0\rangle \Big) > \frac{R_m(x;\eta)}{\eta^2} \tag{59}$$

*then $\mathrm{UB}_{sch}(x;\eta) < \mathrm{UB}_{agg}(x;\eta)$*

## C.6 PL OR STRONG CONVEXITY: STANDARD RATE AND UPPER-BOUND GAINS FOR SCHEDULING

Assume $F$ satisfies the Polyak–Łojasiewicz (PL) inequality with parameter $\mu > 0$:

$$\tfrac{1}{2}|\nabla F(x)|^2 \geq \mu\,(F(x) - F^\star), \qquad \forall x \tag{60}$$

For any $\eta \in (0, 1/L]$, the single aggregated update satisfies the standard GD bound

$$F(x^{\mathrm{agg}}) \leq F(x) - \eta\Big(1 - \tfrac{L\eta}{2}\Big) |\nabla F(x)|^2 \leq \Big(1 - 2\mu\eta\big(1 - \tfrac{L\eta}{2}\big)\Big)\big(F(x) - F^\star\big) \tag{61}$$

Define the upper-bound gain (under per-group lipschitzness, so both bounds are expressed at start-of-refresh):

$$\Delta\mathrm{UB}(x;\eta) := \mathrm{UB}_{\mathrm{agg}}(x;\eta) - \mathrm{UB}_{\mathrm{sch}}(x;\eta) \geq 0 \tag{62}$$

whenever 59 holds. Since $F(x^{\text{sch}}) \leq \text{UB}_{\text{sch}}(x; \eta)$ and $\text{UB}_{\text{agg}}(x; \eta)$ upper-bounds the one-shot decrease term in 61, we obtain the bound-level contraction

$$F(x^{\text{sch}}) - F^\star \;\leq\; \left(1 - 2\mu\eta\big(1 - \tfrac{L\eta}{2}\big)\right)\left(F(x) - F^\star\right) \;-\; \Delta_{\text{UB}}(x; \eta). \tag{63}$$

Consequently, under per-group lipschitzness and 59, the scheduled refresh satisfies the standard gradient-descent contraction and, in addition, achieves an extra nonnegative decrement $\Delta_{\text{UB}}(x; \eta)$ in the upper bound.

### C.7 WHY THE ASSUMPTIONS ARE MILD

The assumptions we use are mild. They are standard and naturally align with our training pipeline.

#### C.7.1 $L$-SMOOTHNESS

This is the same regularity used throughout the main paper and in our baselines. Each task loss we optimize is $L_i$-smooth, so the overall objective is $L$-smooth. We only use this to apply the standard smoothness (descent) inequality (Nesterov, 2004; Beck, 2017).

#### C.7.2 PER-GROUP LIPSCHITZNESS OF $G_r$

Each $G_r$ is a fixed linear combination of the task gradients assigned to group $r$. If each task gradient is $L_i$-lipschitz, then $G_r$ is lipschitz with constant $L_r \leq \sum i \in rL_i$. In other words, this property falls out of task-level smoothness. The same smoothness estimates we already use for step-size selection upper-bound the $L_r$.

#### C.7.3 NEGATIVE HESSIAN-WEIGHTED CROSS-TERMS

The condition we use asks that, over the short moves we actually take ($\eta \leq 1/L$), groups that are separated by the scheduler continue to exhibit negative interaction under the local Hessian (i.e., the Hessian-weighted cross-terms remain negative). This aligns with how the scheduler is built. It separates tasks that exhibit sustained negative interactions and it periodically refreshes assignments so the local geometry does not drift far. Thus the assumption matches the mechanism we deploy.

#### C.7.4 PL AND STRONG CONVEXITY

We invoke PL only to convert a per-refresh decrease into a standard contraction factor. We do not require global strong convexity. A local PL inequality around the iterates is enough, which is commonly observed after warm-up and annealing we already use (Karimi et al., 2016; Zhou et al., 2021; Liu, 2025).

### C.8 CONCLUDING REMARKS

This appendix formalizes a bound-level comparison between scheduled and aggregated updates. Without additional structure the scheduled bound need not be tighter, but under per-group lipschitzness and negative Hessian-weighted cross-terms it becomes strictly tighter, and under PL the scheduled refresh inherits the standard GD contraction with an additional nonnegative decrement. In practice, these conditions arise naturally once the task-group assignments stabilize, so the scheduler will typically achieve tighter descent bounds without changing step sizes or gradient information.

## D COMPUTATIONAL COMPLEXITY OF ONE REFRESH (AND AMORTIZED OVER TRAINING)

We analyze the computational and memory complexity of the proposed interference-aware scheduler per refresh and its amortized cost over training. The former accounts for the cost of a single refresh operation while the latter represents the average cost distributed across all training steps. We distinguish the work required by the underlying multi-task training objective (e.g., backpropagation

to obtain gradients) from the scheduler overhead (EMA maintenance, cosine computation, conflict graph construction, and color).

## D.1 NOTATION

- $K \in \mathbb{N}$ – *number of tasks*
- $d \in \mathbb{N}$ – *dimension of the gradient EMA vector per task*
- $R \in \mathbb{N}$ – *refresh period (number of training steps between graph rebuilds)*
- $\beta \in [0, 1)$ – *exponential moving average (EMA) parameter*
- $T \in \mathbb{N}$ – *total number of training steps*
- $\mathsf{G} > 0$ – *time to compute one backward pass to obtain a task gradient at a refresh*
- $\tau \in (0, 1)$ – conflict threshold; an undirected edge $\{i, j\}$ is present iff $\widehat{C}_{ij} < -\tau$
- $T_{\text{refresh}} > 0$ – *time cost of a single scheduler refresh*
- $S_{\text{refresh}} > 0$ – *peak additional memory used during a refresh*
- $N_{\text{refresh}} \in \mathbb{N}$ – *number of refreshes over $T$ steps with period $R$ (satisfies $N_{\text{refresh}} \in \{\lfloor T/R \rfloor, \lceil T/R \rceil\}$ and $N_{\text{refresh}} \leq T/R + 1$)*
- $r \in \mathbb{N}$ – *dimension of the sketch space used for cosine computation (number of columns of the random projection matrix)*

## D.2 PER-REFRESH COMPLEXITY

At a refresh, the scheduler performs a finite sequence of deterministic operations on the current collection of task-wise exponential moving averages (EMAs) of gradients. Let

$$M \in \mathbb{R}^{K \times d} \tag{64}$$

denote the matrix whose $i$-th row $m_i^\top$ is the EMA for task $i$. We fix a random matrix $R \in \mathbb{R}^{d \times r}$ with $r \ll d$ and form a lower dimensional sketch

$$M_f := MR \in \mathbb{R}^{K \times r}. \tag{65}$$

A refresh first updates these rows through a scalar EMA rule

$$m_i \leftarrow \beta m_i + (1 - \beta) g_i \tag{66}$$

using the most recent probe (or reused) gradient $g_i$. It then constructs the cosine-similarity matrix in the sketch space

$$\widehat{C} = \widetilde{M}_f \widetilde{M}_f^\top \tag{67}$$

where $\widetilde{M}_f$ is the row-normalized version of $M_f$. It thresholds $\widehat{C}$ at $-\tau$ to obtain the conflict adjacency. Finally, it applies a graph-coloring routine to the resulting simple graph (Welsh & Powell, 1967).

EMA maintenance uses a constant number of vector operations per task: one multiply-add on each of the $d$ coordinates of $m_i$. Aggregating over all $K$ tasks gives a time proportional to $Kd$. The storage required to hold all EMAs is the $K \times d$ array $M$, so the working set devoted to EMAs is $\Theta(Kd)$ numbers.

The construction of $\widehat{C}$ in the sketched space proceeds in three stages: (i) forming the sketch $M_f = MR$, (ii) normalizing each row of $M_f$, and (iii) multiplying $\widetilde{M}_f$ by its transpose. The sketching multiply touches every entry of $M$ and $R$ and therefore costs $O(Kdr)$ time. Row normalization touches each entry of $M_f$ exactly once and therefore costs $\Theta(Kr)$ time. The Gram product $\widetilde{M}_f \widetilde{M}_f^\top$ consists of $K^2$ dot products of length $r$, which is $O(K^2 r)$ time (Kågström et al., 1998). The cosine matrix itself occupies $K^2$ entries. If it is retained after thresholding, it uses $\Theta(K^2)$ space. If dropped right after graph construction, that $\Theta(K^2)$ storage is only temporary.

Thresholding linearly scans the off-diagonal of $\widehat{C}$, adding an undirected edge when $\widehat{C}_{ij} < -\tau$; this costs $\Theta(K^2)$ time. The result is either a dense $K \times K$ boolean array requiring $\Theta(K^2)$ space, or a sparse adjacency whose size depends on the number of conflicts (e.g., $\Theta(kK)$ when retaining the $k$ most negative entries per row).

Putting these pieces together yields the following statement.

**Proposition 5** (Per-refresh scheduler overhead with random projections). *Under the standard RAM model with dense matrix multiplication in the sketch space costed as $O(K^2 r)$, the time required by a single scheduler refresh is*

$$T_{refresh} = \Theta(Kd) + O(Kdr) + O(K^2 r) + O(K^2) = O(Kdr + K^2 r), \qquad (68)$$

*and the additional space required by the scheduler during the refresh is*

$$S_{refresh} = \Theta(Kd) + \Theta(Kr) + \Theta(K^2), \qquad (69)$$

*where the $\Theta(K^2)$ term is transient if $C$ is not retained after coloring and the $\Theta(Kr)$ term is transient if the sketch $M_f$ is discarded between refreshes and recomputed from $M$.*

*Proof.* The EMA update costs $\Theta(Kd)$ by a direct count of coordinate-wise multiplication and addition. Forming the random projection sketch $M_f = MR$ touches each entry of $M$ and $R$, and therefore costs $O(Kdr)$. Row-normalizing $M_f$ then costs $\Theta(Kr)$, since it processes all $Kr$ entries once.

In the sketched space, the Gram matrix $\widehat{C} = \widetilde{M}_f \widetilde{M}_f^\top$ requires $K^2$ inner products of length $r$, which is $O(K^2 r)$ time. Thresholding scans $O(K^2)$ entries and is therefore $\Theta(K^2)$. The greedy coloring performs a sort of $K$ keys and then assigns at most one color per edge incident on the current vertex, which is $O(K^2)$ in the worst case. These $\Theta(K^2)$ terms are dominated by the $O(Kdr)$ and $O(K^2 r)$ contributions once $r \geq 1$ and $K$ is nontrivial.

Summing these contributions and absorbing lower-order terms yields $T_{\text{refresh}} = \Theta(Kd) + O(Kdr) + O(K^2 r) + O(K^2) = O(Kdr + K^2 r)$. $\qquad \square$

### D.3 Amortized cost over training

Let $R \in \mathbb{N}$ denote the refresh period as the scheduler executes a refresh once every $R$ training steps. Consider a training run of length $T$ steps. The number of refreshes executed is $\lfloor T/R \rfloor$ or $\lceil T/R \rceil$ depending on whether one refresh occurs at step 0. In either case it is bounded by $T/R + 1$. Multiplying the per-refresh time $T_{\text{refresh}}$ by the number of refreshes and dividing by $T$ shows that the amortized scheduler time per training step satisfies

$$\frac{1}{T} N_{\text{refresh}} T_{\text{refresh}} \leq \frac{1}{T}\left(\frac{T}{R} + 1\right) T_{\text{refresh}} = \frac{1}{R} T_{\text{refresh}} + \frac{1}{T} T_{\text{refresh}} \qquad (70)$$

Letting $T \to \infty$ (or simply taking $T$ large compared to one refresh) eliminates the $T^{-1} T_{\text{refresh}}$ boundary term, yielding the asymptotic amortized bound

$$\frac{1}{R} T_{\text{refresh}} = \frac{1}{R} O(Kdr + K^2 r) = O\left(\frac{Kdr + K^2 r}{R}\right). \qquad (71)$$

If probe gradients are computed only at refreshes, their contribution $KG$ per refresh adds $\frac{1}{R}\Theta(KG)$ to the amortized time per step. If, instead, the training loop already computes task-wise gradients each step and these are reused to update the EMAs, then the probe term is absent and the amortized scheduler overhead remains $O\big((Kdr + K^2 r)/R\big)$.

The amortized space usage is simpler. The EMA matrix $M$ must be retained throughout training and therefore contributes $\Theta(Kd)$ at all times. The cosine matrix $\widehat{C}$ and the adjacency are constructed only during the refresh. They're released after coloring, so the $\Theta(K^2)$ space does not persist. Consequently, the persistent memory overhead attributable to the scheduler is $\Theta(Kd)$, while the peak overhead during a refresh is $\Theta(Kd) + \Theta(K^2)$.

### D.4 Conditions for negligible overhead

Let the amortized per-step costs be

$$C_{\text{sched}} = \frac{a}{R}\,(Kdr + K^2 r), \qquad C_{\text{probe}} = \frac{b}{R}\,KG, \tag{72}$$

where $a, b > 0$ are platform-dependent constants and $G$ denotes the per-task backpropagation cost of the optional probe at a refresh. For fixed $R$,

$$\frac{C_{\text{sched}}}{C_{\text{probe}}} = \frac{a}{b}\,\frac{Kdr + K^2 r}{KG} = \frac{ar}{b}\,\frac{d + K}{G}. \tag{73}$$

Hence $C_{\text{sched}}$ is negligible relative to $C_{\text{probe}}$ whenever

$$\frac{C_{\text{sched}}}{C_{\text{probe}}} \leq \varepsilon \quad \Longleftrightarrow \quad r\,(d + K) \leq \frac{b}{a}\,\varepsilon\,G. \tag{74}$$

### D.5 Reducing Time Complexity

In this section, we detail approaches that can be taken under certain circumstances to optimize time complexity.

#### D.5.1 Random projections

We replace the EMA matrix $M \in \mathbb{R}^{K \times d}$ by a lower-dimensional sketch $\widetilde{M} = MR$ with $R \in \mathbb{R}^{d \times r}$ and $r \ll d$ (Dasgupta & Gupta, 2003). The sketching multiply costs $O(Kdr)$ and the cosine Gram becomes $O(K^2 r)$ instead of $\Theta(K^2 d)$. Storage for the sketched EMAs is $O(Kr)$. By the Johnson-Lindenstrauss (JL) random projection guarantee, if we map the $K$ task-EMA vectors from $\mathbb{R}^d$ to $\mathbb{R}^r$ using a suitable random matrix with $r = \Theta(\epsilon^{-2} \log K)$, then after row normalization all pairwise inner products (hence cosines) are preserved within $\pm\epsilon$ with high probability. We assume a uniform row-norm floor $\min_i \|m_i\| \geq m_0 > 0$ (which can be enforced in practice by skipping tasks with $\|m_i\| < \nu \ll m_0$) so cosine errors remain controlled. Choosing $\epsilon < \gamma$, where $\gamma$ is the cosine margin from the recovery analysis, ensures that every pair remains on the same side of the threshold $-\tau$. Therefore the set $\{(i,j) : \widehat{C}_{ij} < -\tau\}$ and the resulting coloring are unchanged with high probability.

In short, dimensionality drops from $d$ to $r$, the refresh cost drops from $\Theta(K^2 d)$ to $O(Kdr + K^2 r)$, and decisions are preserved as long as the chosen $r$ makes the embedding error smaller than the margin.

#### D.5.2 Deterministic covariance sketching via frequent directions

We maintain a deterministic sketch $B \in \mathbb{R}^{\ell \times d}$ of the row space of $M$ using Frequent Directions and either project rows onto $\text{span}(B)$ or form an approximate Gram from the sketch (Liberty, 2013; Ghashami et al., 2016a). Maintaining the sketch costs $O(Kd\ell)$, the cosine Gram in the sketch space costs $O(K^2 \ell)$, and storage for the sketch is $O(\ell d)$. Frequent Directions gives a spectral-norm bound

$$|MM^\top - \widehat{MM^\top}|_2 \leq \epsilon |M|_F^2 \tag{75}$$

when $\ell = \Theta(\epsilon^{-2})$, which yields a uniform bound on inner-product and squared-norm errors. Assuming a row-norm floor $\min_i \|m_i\| \geq m_0 > 0$ and applying a standard cosine perturbation bound after row normalization, one obtains

$$\left| \cos(m_i, m_j) - \widehat{\cos}(m_i, m_j) \right| \leq \frac{2\,\epsilon\,\|M\|_F^2}{m_0^2} + O\!\left( \frac{\epsilon^2\,\|M\|_F^4}{m_0^4} \right) \tag{76}$$

Taking $\epsilon$ small enough so that the right-hand side is $< \gamma$ ensures that all threshold decisions and the resulting coloring are preserved deterministically. Thus the effective dimension drops from $d$ to $\ell$ in the worst case, and the refresh cost becomes $O(Kd\ell + K^2 \ell)$.

Table 3: Runtime and Taskonomy Tiny validation metrics for SON-GOKU scheduling variants.

| Family | Variant | Runtime | | | Taskonomy Tiny | | | |
|---|---|---|---|---|---|---|---|---|
| | | Elapsed (s) ↓ | Imgs/s ↑ | Refresh (ms) ↓ | Depth Eucl. RMSE ↓ | Depth Eucl. MAE ↓ | Normal mean (deg) ↓ | Reshading MAE ↓ |
| Baseline | FD (conservative) | 76.37±1.84 | 50.28 ± 1.21 | 488.94±27.35 | **25.58** ±0.92 | **7.98** ±0.41 | **56.63** ±1.87 | **0.238** ±0.018 |
| Random proj. | 128 random dim. | 72.32±2.76 | 53.10±1.95 | 366.70±34.12 | 26.19±1.37 | 9.05±0.63 | 70.37±3.91 | 0.334±0.039 |
| Freq. Directions | 128 FD width $r$ | 73.85±1.57 | 52.00±1.33 | 391.15±29.44 | 27.49±1.68 | 9.82±0.71 | 70.97±4.22 | 0.490±0.065 |
| | 256 FD width $r$ | 75.11±1.23 | 51.13±1.09 | 440.05±23.08 | 26.34±1.05 | 8.72±0.52 | 62.37±2.98 | 0.339±0.033 |
| Edge sampling | 25% sampling rate | 72.38±3.41 | 53.05±2.37 | 342.26±48.73 | 126.01±18.92 | 49.50±7.31 | 77.48±6.85 | 5.959±0.821 |
| Incremental Gram | 1e-3 threshold $\epsilon$ | **70.07**±2.04 | **54.80**±2.11 | **293.36**±31.29 | 34.41±4.67 | 15.36±1.88 | 93.86±9.73 | 0.621±0.079 |

### D.5.3 EDGE SAMPLING FOR CONFLICT GRAPHS WITH ADAPTIVE REFINEMENT

We reduce the number of cosine evaluations by computing $\widehat{C}_{ij}$ for only $\tilde{O}(K \log K)$ randomly chosen task pairs to build a provisional conflict graph and then refining by evaluating additional pairs that are near the threshold or needed to certify connectivity and chromatic structure. We still compute all $K$ row norms once in $O(Kd)$ time for normalization, and the first pass costs $O(Kd \log K)$ for the sampled dot products. The total cost adds only the refinement work, which remains small when only few pairs are ambiguous. Under a planted separation model with margin $\gamma$ and reasonably dense cross-group conflicts, one can show with high probability that the sampled graph already captures the correct inter-group connectivity, so the coloring or component structure is recovered after the first pass and only boundary pairs need refinement. This reduces the pairwise work from $K^2$ to near $K \log K$ while preserving the final decisions under stated assumptions (Erdős & Rényi, 1960).

### D.5.4 INCREMENTAL GRAM UPDATES

We avoid rebuilding the full cosine matrix when only a small subset of tasks has meaningfully changed since the last refresh. If $s$ rows of $M$ cross a chosen change threshold, we first renormalize these rows and then recompute both the corresponding $s$ rows and $s$ columns of the Gram by taking dot products against all $K$ rows, which costs $O(sKd)$, with an additional $O(sd)$ to update norms, instead of $\Theta(K^2 d)$, and we leave all unchanged entries as they are. This update is exact for the affected entries, so conflict edges and coloring decisions are preserved by construction, and the reduction is deterministic whenever $s \ll K$. To prevent slow drift in the unchanged entries, we can periodically force a full rebuild and reset the change counters.

### D.5.5 EXPERIMENTAL ANALYSIS

We evaluated each optimizations' speed and relative impact on performance on the Taskonomy Tiny subset. We use the Tiny subset here as this experiment does not require large-scale training for valid results.

Results are presented in Table 3. We can see that, in practice, every approach actually does improve speed over the SON-GOKU baseline (Frequent Directions with high $r$, note that we use $r$ in the same way that Ghashami et al. 2016b use $\ell$). However, every approach also degrades performance (by performance we are referring the main objective metric, like loss or accuracy, not speed) to varying extents[1]. Edge sampling and incremental gram updates have an extremely negative impact on performance. Such approaches may need additional fine-tuning or may only be practical in specific settings. Interestingly, we observe a slight decrease in performance when using lower width for Frequent Directions, demonstrating the tradeoffs between speed and performance that come with such an approach.

Overall, every approach achieves the desired effect of increasing speed, but does so at varying costs in performance. Each approach will require careful fine tuning and usage in real-world deployment to properly weigh this tradeoff. Based on our experimentation, it appears that Frequent Directions offers the most consistent and reasonable tradeoff that clearly scales with the FD width.

---

[1]Please note that varying performance for the SON-GOKU baseline across experiments is due to changes in the backbone model architecture and the Taskonomy subset.

# E DESCENT PRESERVATION UNDER $\tau$-COMPATIBILITY

## E.1 PROOF OF PROPOSITION 6

**Proposition 6.** *Let $S \subseteq \{1, \ldots, K\}$ be a $\tau$-compatible task set. That is, every pair of gradients satisfies*

$$\langle g_i, g_j \rangle \geq -\tau \|g_i\| \|g_j\|, \qquad \forall i \neq j \in S, \quad 0 \leq \tau < 1 \tag{77}$$

*Then*

$$\left\| \sum_{k \in S} g_k \right\|^2 \geq \left(1 - \tau(|S| - 1)\right) \sum_{k \in S} \|g_k\|^2. \tag{78}$$

*Proof.* We begin with the polarization identity for any finite set of vectors:

$$\left\| \sum_{k \in S} g_k \right\|^2 = \sum_{k \in S} \|g_k\|^2 + 2 \sum_{\substack{i,j \in S \\ i < j}} \langle g_i, g_j \rangle. \tag{79}$$

### E.1.1 LOWER-BOUNDING THE CROSS TERMS

Because $S$ is $\tau$-compatible, inequality (77) gives

$$\langle g_i, g_j \rangle \geq -\tau \|g_i\| \|g_j\|. \tag{80}$$

Insert this bound into (79) to obtain

$$\left\| \sum_k g_k \right\|^2 \geq \sum_k \|g_k\|^2 - 2\tau \sum_{i<j} \|g_i\| \|g_j\|. \tag{81}$$

### E.1.2 SYMMETRIZING THE MIXED SUM

Observe that

$$\sum_{i<j} \|g_i\| \|g_j\| = \frac{1}{2} \sum_{\substack{i,j \\ i \neq j}} \|g_i\| \|g_j\|. \tag{82}$$

Substituting (82) into (81) yields

$$\left\| \sum_k g_k \right\|^2 \geq \sum_k \|g_k\|^2 - \tau \sum_{\substack{i,j \\ i \neq j}} \|g_i\| \|g_j\|. \tag{83}$$

### E.1.3 BOUNDING THE MIXED SUM VIA CAUCHY-SCHWARZ

Apply the Cauchy-Schwarz inequality in $\mathbb{R}^{|S|}$ to the vectors $a = (\|g_1\|, \ldots, \|g_{|S|}\|)$ and $\mathbf{1} = (1, \ldots, 1)$:

$$\sum_k \|g_k\| = \langle a, \mathbf{1} \rangle \leq \|a\| \|\mathbf{1}\| = \left( \sum_k \|g_k\|^2 \right)^{1/2} \sqrt{|S|}. \tag{84}$$

Using $\left( \sum_k a_k \right)^2 \leq |S| \sum_k a_k^2$ and (85),

$$\sum_{i \neq j} \|g_i\| \|g_j\| = \left( \sum_k \|g_k\| \right)^2 - \sum_k \|g_k\|^2, \tag{85}$$

we obtain the standard estimate

$$\sum_{i \neq j} \|g_i\| \|g_j\| \leq (|S| - 1) \sum_k \|g_k\|^2. \tag{86}$$

Hence,

$$\tau \sum_{i \neq j} \|g_i\| \|g_j\| \leq \tau (|S| - 1) \sum_k \|g_k\|^2. \tag{87}$$

### E.1.4 COMBINING BOUNDS

Insert (87) into (83):

$$\left\|\sum_k g_k\right\|^2 \geq \sum_k \|g_k\|^2 - \tau(|S|-1)\sum_k \|g_k\|^2 = \left(1 - \tau(|S|-1)\right)\sum_k \|g_k\|^2, \qquad (88)$$

which is (78). □

### E.2 INTERPRETATION AND PRACTICAL IMPLICATIONS

Equation (78) guarantees that whenever we restrict an SGD step to a $\tau$-compatible group (i.e., a set of tasks whose gradients are not too conflicting) the resulting joint update preserves at least a $\left(1 - \tau(|S|-1)\right)$ fraction of the summed squared step lengths.

Below, we provide a strictly stronger version that is assumption free.

**Proposition 7** (Data-Dependent Lower Bound via the Aggregate Conflict Ratio). *Define the aggregate conflict ratio*

$$\tau_{\text{eff}}(S) := \frac{\sum\limits_{i \neq j} \left(-\langle g_i, g_j \rangle\right)_+}{\sum\limits_k \|g_k\|^2}, \qquad (x)_+ := \max\{x, 0\}. \qquad (89)$$

*Then, without additional assumptions,*

$$\left\|\sum_{k \in S} g_k\right\|^2 \geq \left(1 - \tau_{\text{eff}}(S)\right)\sum_{k \in S} \|g_k\|^2, \qquad (90)$$

*and under $\tau$-compatibility we always have $\tau_{\text{eff}}(S) \leq \tau(|S|-1)$, so (90) is never weaker than (78).*

Our takeaways from this are as follows:

(i) *Descent direction safety.* The aggregated step is guaranteed to be a descent direction whenever $\tau_{\text{eff}}(S) < 1$ (data-dependent) and, in particular, whenever $\tau(|S|-1) < 1$ (worst-case).

(ii) *Convergence-rate constant.* In analyses for smooth SGD, one may replace $\|g_t\|^2$ by the right-hand side of either (90) (which is tighter) or (78) (worst-case), leading respectively to constants involving $\tau_{\text{eff}}(S_t)$ or $\tau(|S_t|-1)$.

## F CONVERGENCE RATE WITH $\tau$-DEPENDENT CONSTANT

**Theorem 7** (Baseline $O(1/\sqrt{T})$ convergence of the full gradient). *Let $F(\theta) = \sum_{k=1}^K \mathcal{L}_k(\theta, \phi_k)$ be $L$–smooth in the shared parameters $\theta$. Assume the stochastic gradient $g_t$ obtained at step $t$ satisfies $\mathbb{E}[g_t \mid \theta_t] = \nabla F(\theta_t)$ and $\mathbb{E}[\|g_t - \nabla F(\theta_t)\|^2 \mid \theta_t] \leq \sigma^2$. Let the step size be $\eta = \frac{c}{\sqrt{T}}$ with $0 < c \leq \frac{1}{L}$, and suppose the scheduler selects a $\tau$-compatible task set $S_t$ at each step (this will be used below for a refinement). Then*

$$\min_{1 \leq t \leq T} \mathbb{E}\left[\|\nabla F(\theta_t)\|^2\right] \leq \frac{2\left(F_0 - F^\star\right)}{c\sqrt{T}} + \frac{cL\sigma^2}{\sqrt{T}}. \qquad (91)$$

*Proof.* Because $F$ is $L$–smooth, for any $\eta \leq \frac{1}{L}$ the standard non-convex SGD inequality (Ghadimi & Lan 2013, Lemma 3.2) gives

$$\mathbb{E}\left[F(\theta_{t+1})\right] \leq \mathbb{E}\left[F(\theta_t)\right] - \frac{\eta}{2}\mathbb{E}\left[\|\nabla F(\theta_t)\|^2\right] + \frac{\eta^2 L\sigma^2}{2}. \qquad (92)$$

Summing equation 92 over $t = 0, \ldots, T-1$ and using $\mathbb{E}[F(\theta_T)] \geq F^\star$ yields

$$\frac{\eta}{2}\sum_{t=0}^{T-1}\mathbb{E}\left[\|\nabla F(\theta_t)\|^2\right] \leq F_0 - F^\star + \frac{\eta^2 L\sigma^2 T}{2}. \qquad (93)$$

Dividing by $T$, using $\min_t x_t \leq \frac{1}{T}\sum_t x_t$, and substituting $\eta = \frac{c}{\sqrt{T}}$ gives equation 91. □

**Data-dependent $\tau$-refinement for the scheduled gradient energy.** For a finite set $S$, define the aggregate conflict ratio

$$\tau_{\text{eff}}(S) := \frac{\sum_{i \neq j \in S}\left(-\langle g_i, g_j\rangle\right)_+}{\sum_{k \in S}\|g_k\|^2} \in [0, \infty), \qquad (x)_+ = \max\{x, 0\}. \tag{94}$$

Then for every step $t$,

$$\left\|\sum_{k \in S_t} g_{k,t}\right\|^2 \geq \left(1 - \tau_{\text{eff}}(S_t)\right)\sum_{k \in S_t}\|g_{k,t}\|^2. \tag{95}$$

Consequently,

$$\frac{1}{T}\sum_{t=0}^{T-1}\mathbb{E}\Big[\sum_{k \in S_t}\|g_{k,t}\|^2\Big] \leq \underbrace{\frac{1}{T}\sum_{t=0}^{T-1}\mathbb{E}\Big[\frac{1}{1 - \tau_{\text{eff}}(S_t)}\Big]}_{=: \Gamma_T} \cdot \frac{1}{T}\sum_{t=0}^{T-1}\mathbb{E}[\|g_t\|^2]. \tag{96}$$

Using $\mathbb{E}\|g_t\|^2 = \mathbb{E}\|\nabla F(\theta_t)\|^2 + \mathbb{E}\|g_t - \nabla F(\theta_t)\|^2 \leq \mathbb{E}\|\nabla F(\theta_t)\|^2 + \sigma^2$ and the average version of equation 92,

$$\frac{1}{T}\sum_{t=0}^{T-1}\mathbb{E}[\|\nabla F(\theta_t)\|^2] \leq \frac{2(F_0 - F^\star)}{\eta T} + L\eta\sigma^2, \tag{97}$$

we obtain the $\tau$-dependent, data-driven control

$$\frac{1}{T}\sum_{t=0}^{T-1}\mathbb{E}\Big[\sum_{k \in S_t}\|g_{k,t}\|^2\Big] \leq \Gamma_T\left(\frac{2(F_0 - F^\star)}{\eta T} + L\eta\sigma^2 + \sigma^2\right). \tag{98}$$

If, in addition, each $S_t$ is pairwise $\tau$-compatible with $|S_t| = s_t$ and $\tau(s_t - 1) \leq \rho < 1$ uniformly in $t$, then $\tau_{\text{eff}}(S_t) \leq \tau(s_t - 1) \leq \rho$ and hence $\Gamma_T \leq \frac{1}{1-\rho}$. With $\eta = \frac{c}{\sqrt{T}}$, equation 98 becomes

$$\frac{1}{T}\sum_{t=0}^{T-1}\mathbb{E}\Big[\sum_{k \in S_t}\|g_{k,t}\|^2\Big] \leq \frac{1}{1-\rho}\left(\frac{2(F_0 - F^\star)}{c\sqrt{T}} + \frac{cL\sigma^2}{\sqrt{T}} + \sigma^2\right). \tag{99}$$

### F.1 DISCUSSION AND INTUITION

Equation equation 91 is the classical $O(1/\sqrt{T})$ rate for non-convex SGD with unbiased and bounded variance gradients and constant-over-time step size $\eta = c/\sqrt{T}$. Under these conditions, the convergence rate in terms of the full gradient norm $\|\nabla F(\theta_t)\|^2$ does not depend on $\tau$. However, the scheduler's $\tau$ structure does control the per step energy of the scheduled gradient through equation 96– equation 99. Less cross-task conflict (smaller $\Gamma_T$) results in a tighter bound on $\frac{1}{T}\sum_t\sum_{k \in S_t}\|g_{k,t}\|^2$, which is the quantity governed by the descent preservation inequalities used throughout the analysis.

## G BOUNDED STALENESS VIA GREEDY GRAPH COLORING

**Proposition 8** (Staleness Bound). *Let $G = (\mathcal{T}, E)$ be the task–conflict graph whose vertices are tasks and whose edges connect pairs with interference coefficient exceeding the threshold $\tau$. Denote by $\Delta$ its maximum degree. Greedy graph coloring produces a proper coloring $C_1, \ldots, C_m$ with*

$$m \leq \Delta + 1. \tag{100}$$

*If the scheduler activates the color classes in the cyclic order $C_1 \to C_2 \to \ldots \to C_m \to C_1 \to \ldots$, then every task is updated at least once every*

$$s_{\max} = m - 1 \leq \Delta \tag{101}$$

*iterations. In particular, the schedule enforces a bounded inter-update delay of at most $\Delta$ iterations per task, consistent with the bounded-delay assumption of Recht et al. (Niu et al., 2011).*

*Proof.* We proceed in two parts.

**Part A: Color count bound.** A greedy algorithm scans vertices in some order and assigns to each vertex the smallest available color not used by its already colored neighbors. When the $i$-th vertex $v$ is reached, at most $\deg(v) \leq \Delta$ of its neighbors are already colored, so at most $\Delta$ colors are unavailable. Therefore one of the first $\Delta + 1$ colors is always free, implying $m \leq \Delta + 1$ (Lovász, 2006).

**Part B: Staleness of cyclic execution.** Fix any task $T \in \mathcal{T}$ and let it belong to color $C_j$ for some $1 \leq j \leq m$. Under cyclic scheduling, $C_j$ is executed at steps $t = j,\ j + m,\ j + 2m, \dots$. The number of intervening steps between two consecutive executions of $C_j$ is exactly $m - 1$. Hence task $T$ never waits more than $s_{\max} = m - 1$ iterations for an update. Combining with Equation 8 yields $s_{\max} \leq \Delta$. $\qquad\square$

## G.1 Interpretation

The bound (Equation 101) guarantees that the shared parameters used by any task are refreshed at least once every $\Delta$ iterations in the worst case (e.g., when the conflict graph is a clique of size $\Delta + 1$). This aligns with the bounded-delay assumption common in analyses of asynchronous SGD and lock-free training, so convergence proofs built under that assumption apply to our cyclic schedule with delay parameter at most $\Delta$ when iterations are used as the unit of delay (Niu et al., 2011; Lian et al., 2015). In practice $\Delta$ is often much smaller than the total number of tasks, so the scheduler achieves low interference and low parameter staleness simultaneously.

## H Greedy Graph-Coloring Uses at Most $\Delta+1$ Colors

### H.1 Proof of Proposition 9

**Proposition 9** (Coloring Period Bound). *Let $G = (V, E)$ be a finite, simple, undirected graph with maximum degree $\Delta := \max_{v \in V} \deg(v)$. The greedy (first-fit) coloring algorithm (e.g., Welsh–Powell order)[2] produces a proper vertex coloring with no more than*

$$\chi_{greedy}(G) \ \leq \ \Delta + 1 \tag{102}$$

*distinct colors. Consequently, when the scheduler activates the color classes in a cyclic order, the cycle length is bounded by $\Delta + 1$. This is a quantity depending only on the structure of the conflict graph.*

*Proof.* Let the vertices be processed in the chosen order $v_1, v_2, \dots, v_{|V|}$ (e.g., Welsh-Powell). Assume inductively that after coloring the first $k - 1$ vertices the algorithm has used at most $\Delta + 1$ colors. Consider vertex $v_k$. Since $\deg(v_k) \leq \Delta$, at most $\Delta$ neighbors of $v_k$ can appear before $v_k$ in the ordering. Hence, at the moment of coloring $v_k$, at most $\Delta$ colors are forbidden (one for each previously colored neighbor). Among the palette $\{1, 2, \dots, \Delta + 1\}$ there is therefore at least one color still available. Assigning the smallest such color to $v_k$ maintains a proper coloring and never introduces a new color beyond $\Delta + 1$.

Proceeding vertex-by-vertex, no step ever requires more than $\Delta + 1$ colors, establishing equation 102. $\qquad\square$

### H.2 Implications for the scheduler

A coloring with at most $\Delta + 1$ classes means the scheduler's cycle period (the number of batches needed before every task reappears) is bounded by a graph invariant independent of the number of tasks. Even if thousands of tasks exist, as long as each one conflicts with at most $\Delta$ others, the memory footprint (one shared backbone plus $\Delta + 1$ sets of head activations) and the maximum waiting time between successive updates for any task (bounded by $\Delta$, see Proposition 8) remain predictable and small. This guarantee is essential for scaling the scheduler to large, heterogeneous tasks.

---

[2]Order the vertices in non–increasing degree and assign to each the smallest positive integer (color) not used by its previously colored neighbors.

# I BASELINE NON-CONVEX SGD CONVERGENCE RATE

## I.1 PROOF OF THEOREM 8

**Theorem 8** (Classical $O(1/\sqrt{T})$ bound). *Let $F : \mathbb{R}^d \to \mathbb{R}$ be an $L$-smooth, possibly non-convex objective and suppose the stochastic gradient $g_t$ computed at iteration $t$ satisfies*

$$\mathbb{E}[g_t \mid \theta_t] = \nabla F(\theta_t), \qquad \mathbb{E}[\|g_t - \nabla F(\theta_t)\|^2 \mid \theta_t] \le \sigma^2. \tag{103}$$

*Run SGD with the constant step size $\eta = \frac{c}{\sqrt{T}}$, $0 < c \le \frac{1}{L}$, for $T$ iterations starting from $\theta_0$. Then*

$$\min_{0 \le t < T} \mathbb{E}[\|\nabla F(\theta_t)\|^2] \le \frac{2(F_0 - F^*)}{c\sqrt{T}} + \frac{cL\sigma^2}{\sqrt{T}}, \tag{104}$$

*where $F^* = \inf_\theta F(\theta)$.*

*Proof.* The proof is a streamlined restatement of ((Ghadimi & Lan, 2013; Nemirovski et al., 2009)). By $L$-smoothness,

$$F(\theta_{t+1}) \le F(\theta_t) + \langle \nabla F(\theta_t), \theta_{t+1} - \theta_t \rangle + \frac{L}{2}\|\theta_{t+1} - \theta_t\|^2. \tag{105}$$

With $\theta_{t+1} = \theta_t - \eta\, g_t$ and taking conditional expectation,

$$\mathbb{E}[F(\theta_{t+1})] \le \mathbb{E}[F(\theta_t)] - \eta\, \mathbb{E}[\|\nabla F(\theta_t)\|^2] + \frac{\eta^2 L}{2}\, \mathbb{E}[\|g_t\|^2]. \tag{106}$$

Decompose the squared stochastic gradient:

$$\mathbb{E}[\|g_t\|^2] = \mathbb{E}[\|\nabla F(\theta_t)\|^2] + \mathbb{E}[\|g_t - \nabla F(\theta_t)\|^2] \le \mathbb{E}[\|\nabla F(\theta_t)\|^2] + \sigma^2 \tag{107}$$

Thus, and using $\eta \le 1/L$ so that $\eta - \frac{L\eta^2}{2} \ge \frac{\eta}{2}$,

$$\mathbb{E}[F(\theta_{t+1})] \le \mathbb{E}[F(\theta_t)] - \frac{\eta}{2}\, \mathbb{E}[\|\nabla F(\theta_t)\|^2] + \frac{\eta^2 L\sigma^2}{2}. \tag{108}$$

Summing from $t = 0$ to $T - 1$ and telescoping gives

$$\frac{\eta}{2} \sum_{t=0}^{T-1} \mathbb{E}[\|\nabla F(\theta_t)\|^2] \le F_0 - F^* + \frac{\eta^2 L\sigma^2 T}{2}. \tag{109}$$

Dividing by $\eta T$ and inserting $\eta = c/\sqrt{T}$ yields equation 104. $\qquad\square$

## I.2 CONNECTION TO THE SCHEDULER

At $\tau = 0$, pairs with negative inner product are incompatible, so the conflict graph on tasks can be colored into $m$ classes $\{C_1, \ldots, C_m\}$, and a simple policy activates one color class per step. Under a deterministic (cyclic) activation order, the update $g_t = \sum_{k \in S_t} g_{k,t}$ generally satisfies

$$\mathbb{E}[g_t \mid \theta_t] = \sum_{k \in S_t} \nabla \mathcal{L}_k(\theta_t, \phi_{k,t}) \ne \sum_{k=1}^{K} \nabla \mathcal{L}_k(\theta_t, \phi_{k,t}), \tag{110}$$

so it is biased for the full gradient.

### I.2.1 CONSISTENCY WITH THE UNBIASED SGD ASSUMPTION

The analysis in Theorem 8 assumes an unbiased stochastic gradient, $\mathbb{E}[g_t \mid \theta_t] = \nabla F(\theta_t)$. This assumption is met under either of the following implementations.

*(i) Randomized class sampling with scaling.* Draw $J_t \sim \mathrm{Unif}\{1, \ldots, m\}$ independently each step and set

$$\tilde{g}_t = m \sum_{k \in C_{J_t}} g_{k,t}. \tag{111}$$

Table 4: Information on the datasets utilized in experimentation. (*Some samples were removed during preprocessing)

| Dataset | Main Tasks | (+) Aux. Tasks | (-) Aux Tasks | Modalities | Samples |
|---|---|---|---|---|---|
| NYUv2 | Semantic Segmentation Depth Estimation Surface Normal Prediction | – | Color Temp. Estimation | Image | 250* |
| CIFAR-10 | Image Classification | Quadrant Localization Texture Classification | Corruption-Type Prediction Rotation Angle Prediction | Image | 2,500* |
| AV-MNIST | Digit Classification | Digit Parity | | Audio, Image | 56.0k |
| MM-IMDb | Genre Classification | Release Decade | Title-Iniial Classification | Image, Text | 25.9k |
| STOCKS-F&B | 4× Stock Return Prediction | Five-Day Rolling Volatility Sector-Average Next-Day Return | Day of the Week Prediction Lag-0 Reconstruction of Today's Open-Price | Timeseries ×18 | 75.5k |
| STOCKS-HEALTH | 7× Stock Return Prediction | Five-Day Rolling Volatility Sector-Average Next-Day Return | Day of the Week Prediction Lag-0 Reconstruction of Today's Open-Price | Timeseries ×63 | 75.5k |

Then $\mathbb{E}[\tilde{g}_t \mid \theta_t] = \sum_{k=1}^{K} \nabla \mathcal{L}_k(\theta_t, \phi_{k,t}) = \nabla F(\theta_t)$, so Theorem 8 applies (with the variance bound adjusted for the scaled estimator). Equivalently, one may keep $g_t = \sum_{k \in C_{J_t}} g_{k,t}$ and use an effective step size $m\eta$.

*(ii) Deterministic cyclic schedule.* If the classes are visited in a fixed periodic order, then generally $\mathbb{E}[g_t \mid \theta_t] \neq \nabla F(\theta_t)$ at the per-step level. Nonetheless, standard analyses of nonconvex smooth cyclic block updates yield an $O(1/\sqrt{T})$ decay of the average gradient norm under usual step-size conditions, with constants depending on the number of blocks.

Either implementation delivers an $O(1/\sqrt{T})$ convergence guarantee.

## J    EXPERIMENTAL SETUP FOR DATASETS

We evaluate the proposed scheduler alongside numerous baselines and state-of-the-art models across multiple datasets to reliably assess its general performance relative to other approaches. In total, it is evaluated across 6 datasets.

Across all datasets, we incorporate positive and/or negative auxiliary tasks into training. Positive auxiliary tasks share structure or predictive signals with the main tasks (e.g., common features or correlated outputs) and so can improve the learned representations by providing relevant supervision. In contrast, negative auxiliary tasks are uncorrelated or directly conflicting with the main objectives, inducing gradient interference that can slow or degrade primary performance. Including both creates controlled variation in task alignment, letting us test whether SON-GOKU (1) groups compatible tasks, (2) separates conflicting tasks, and (3) maintains main-task performance under interference created by auxiliary tasks.

### J.1    NYUv2

The NYU Depth Dataset v2 (NYUv2) (Silberman et al., 2012) consists of RGB-D indoor scenes with 1,449 densely labeled pairs of RGB and depth images. To demonstrate auxiliary task value in data-scarce conditions, we employ a subset of 250 training samples randomly selected from the original training set.

We formulate a multi-main-task setup with three primary objectives: (1) semantic segmentation (14 classes), (2) depth estimation where the model predicts per-pixel depth values from RGB images, and (3) surface normal prediction where 3-channel surface normals are estimated from RGB input. The negative auxiliary task is color temperature estimation, a synthetically generated task that predicts global color temperature properties designed to interfere with the main tasks by emphasizing global color distribution rather than local semantic and geometric features.

All tasks utilize RGB images as the sole input modality, with depth maps and surface normals serving as prediction targets rather than input features. A ResNet-18 (He et al., 2015) backbone trained from scratch processes the RGB input, with task-specific decoder heads for segmentation (with 32 × upsampling), depth regression, surface normal regression, and color temperature estimation.

## J.2    CIFAR-10

The CIFAR-10 (Krizhevsky et al., 2009) dataset contains 60,000 $32 \times 32$ color images across 10 generic classes. To evaluate our interference-aware scheduler in a data-scarce environment where auxiliary tasks provide maximum benefit, we employ a subset of 2,500 training samples (250 per class) from the original 50,000 training images.

For the multi-task learning setup, we set image classification as the main task and construct three auxiliary tasks synthetically from the RGB images. The positive auxiliary tasks include: (1) quadrant localization, where the model predicts which quadrant contains the primary object, and (2) texture classification using Gabor filter responses clustered into 8 texture categories via k-means clustering. The negative auxiliary tasks consist of: (3) corruption-type prediction, where images are artificially corrupted using 15 different corruption types from the ImageNet-C corruption suite (Hendrycks & Dietterich, 2019), and (4) rotation angle prediction, where images are rotated by 0°, 90°, 180°, or 270° and the model predicts the rotation angle.

All tasks share a ResNet-18 (He et al., 2015) backbone trained from scratch without pretraining, with task-specific heads for each auxiliary task.

## J.3    AV-MNIST

The AV-MNIST benchmark (Vielzeuf et al., 2018) pairs MNIST images (Lecun et al., 1998) with a log-mel spectrogram of the corresponding spoken digit from TIDIGITS (Leonard & Doddington, 1993). It is a synthetic benchmark that has significant noise applied to audio and feature reduction applied to images, making it far more difficult than the original MNIST.

We use all paired samples in our experiments. Our primary task is 10-way digit classification. Following (Vielzeuf et al., 2018), we encode images with a small 4-layer convolutional network and spectrograms with a 2-layer CNN, both built and trained from scratch. These embeddings are projected and fused for processing by a simple MLP in intermediate fusion (Boulahia et al., 2021; Guarrasi et al., 2025), as are the models trained on MM-IMDb and STOCKS. We include only one positive auxiliary class, Digital Parity. This task aims to identify the digits as either even or odd, which has been shown to be a positive auxiliary task for improving representations on MNIST-like datasets (Tacchetti et al., 2018; Mohammadi et al., 2020).

## J.4    MM-IMDʙ

The MM-IMDb dataset (Arevalo et al., 2017) contains 25,959 movies with genre annotations over 23 categories. We extract poster images and plot summaries for every movie in the dataset.

The images and summaries are encoded by a frozen VGG16 (Simonyan & Zisserman, 2014) and Google word2vec (Mikolov et al., 2013) model, respectively. Our main task is movie genre prediction. We add one positive auxiliary task, Release Decade, and one negative auxiliary task, the classification of the title's first word as either a vowel or consonant.

## J.5    STOCKS

The STOCKS datasets we use, introduced in (Liang et al., 2021), contain stock market timeseries data across two categories. Specifically: (1) STOCKS-F&B, which has 14 input and 4 output stocks in the GICS Restaurants or Packaged Food & Meats category (MSCI Inc. & S&P Dow Jones Indices, 2024), and (2) STOCKS-HEALTH, which contains 56 input and 7 output stocks in the Health Care category.

Every input stock consists of 500 trading days, with the goal of predicting returns over the next day. We discretize the continuous return variable $R$ into three non-overlapping categories: (1) $Low$, where $0 \leq R < 0.1$, (2) $Medium$, where $0.1 \leq R < 0.5$, and (3) $High$, where $R \geq 0.5$. Mean Absolute Error (MAE) is calculated by mapping the three classes to numbers ($Low \rightarrow 0$, $Medium \rightarrow 1$, $High \rightarrow 2$) and then deriving MAE as usual. Each input series is encoded by the same CNN-BiLSTM network. This consists of 3 CNNs and 1 BiLSTM (Cui et al., 2018).

We augment the main prediction task with two positive auxiliaries and two negative auxiliaries. The first positive task, Five-Day Rolling Volatility, is calculated as the standard deviation of daily logarithmic returns over a sliding five-trading-day window. This feature captures short-term fluctuations in a stock's price. In Sector-Average Next-Day Return, for each date we compute the mean of the actual next-day returns of all stocks within the same GICS sector, providing a simple measure of sector-level momentum and drift

The negative tasks focus on useless information that is meant to distract the model. Namely, day of the week prediction (in the range of Monday to Friday) and Lag-0 Open-Price Reconstruction, which requires the model to reproduce the same day's opening price verbatim. The first is information that contains little to no signals that would contribute to overall performance, and the second is a trivial identity mapping that contributes no real predictive challenge.

## K    MODELS USED FOR COMPARISON

### K.1    BASELINE MODELS

1. *Uniform.* This baseline assigns equal weights to all tasks throughout training, representing the simplest approach where all task losses are weighted equally.

2. *Gradnorm (Chen et al., 2018).* Balances task learning rates by normalizing gradient magnitudes relative to target loss ratios. This maintains consistent training dynamics across tasks.

3. *MGDA (Sener & Koltun, 2018).* Formulates multi-task learning as a multi-objective optimization problem, finding Pareto-optimal solutions (Lockwood, 2008; Pareto, 2014) through gradient descent in the convex hull of gradients (Fliege & Svaiter, 2000; Miettinen, 1999).

### K.2    STATE-OF-THE-ART MODELS

1. *PCGrad (Yu et al., 2020).* Projects conflicting gradients onto orthogonal subspaces when negative cosine similarity is detected, eliminating destructive interference between task gradients.

2. *CAGrad (Liu et al., 2021).* Extends PCGrad by adaptively adjusting gradient magnitudes based on conflict severity. This proves more nuanced modifications to gradients than binary projection.

3. *Adatask (Yang et al., 2023).* Dynamically reweighs task losses using relative loss changes, adapting to varying task learning rates during training.

4. *FAMO (Liu et al., 2023).* Fast Adaptive Multitask Optimization dynamically adjusts task weights to equalize each task's rate of loss improvement. It uses an online, per-step rule (no pairwise gradient ops), adding negligible overhead while remaining robust to loss-scale differences.

5. *Fair Resource Allocation in MTL (FairGrad) (Ban & Ji, 2024).* Views the shared update as a limited resource and chooses it to maximize an $\alpha$-fair utility of per-task improvements. The parameter $\alpha$ controls the trade-off between average performance and fairness.

6. *Nash-MTL (Navon et al., 2022).* Frames multitask training as a bargaining game and computes a scale-invariant weighted combination of task gradients given by the Nash bargaining solution. Weights are obtained by solving a small inner problem (e.g., via CCP) using the gradient Gram matrix. Updates are balanced across tasks.

### K.3    ABLATION STUDY MODELS

1. *Static One-Shot Coloring.* We run the greedy graph coloring once at the start of training, freeze the resulting task groups, and never recompute the conflict graph. All other hyperparameters ($\tau$, history length $H$, and update interval $R$) match the full scheduler. As training progresses we expect the fixed coloring to grow stale, mixing tasks whose interference relationships have changed. This ablation isolates the benefit of dynamic recoloring, showing how much performance depends on adapting the schedule to evolving gradient conflicts.

2. *Single-Step Conflict Estimation.* Here, we set the history length to $H = 1$, so every recoloring step relies on only the most recent mini-batch gradients to estimate interference. Without aggregation over many past steps, the conflict graph should become highly noisy, causing unstable task groupings from one update window to the next. This variant tests the importance of historical conflict statistics in the scheduler. **Threshold Graph (baseline).** We connect tasks $i$ and $j$ whenever the smoothed cosine $\hat{s}_{ij}(t)$ falls below a global threshold $-\tau(t)$. This is the rule used in the main method and analyzed in our recovery and scheduling theory. It prioritizes the most strongly conflicting pairs globally and serves as the reference against which the other graph rules are compared.

3. *kNN-Symmetric Graph.* For each task we identify its $m$ most conflicting neighbors (those with the smallest smoothed cosine values) and add edges to those neighbors. We then symmetrize the graph by including an undirected edge if either endpoint selects the other. This construction roughly fixes the degree of each node and spreads edges more evenly across tasks. It tests whether enforcing local degree control can outperform or match the global threshold rule when conflict is heterogeneous across tasks.

4. *Signed-Only Graph.* Here we connect two tasks only if their smoothed cosine is strictly negative, $\hat{s}_{ij}(t) < 0$, ignoring the magnitude of the conflict. This yields a much sparser graph that only records clearly antagonistic pairs. This ablation explores an extreme notion of conflict and allows us to test whether discarding moderately conflicting (but still harmful) interactions degrades performance.

5. *Quantile Threshold Graph.* Instead of fixing $\tau$ by hand, we set $\tau(t)$ at each refresh so that only the worst $p\%$ of smoothed cosine values are treated as conflicting. This behaves like an adaptive threshold that tracks how the similarity distribution drifts over training, keeping the graph density approximately stable over time. This variant tests whether such an automatically tuned cutoff provides an advantage over the fixed global threshold.

We evaluate each graph rule under two settings. In the *fixed $\tau$* setting, all rules share the same $\tau(t)$ schedule used in the main experiments, and we simply observe the induced edge density, number of colors, and validation performance. In the *density-matched* setting, we adjust the hyperparameters of each rule (e.g., $m$ for kNN, percentile $p$ for quantile) so that all graphs have approximately the same edge density at each refresh. This isolates the effect of which pairs are marked as conflicting, rather than how many edges are present.

## L  EXPANDED WALL-CLOCK TIME STUDY

We provide more results from our wall-clock time study. The expanded table includes results from testing refresh rates $R \in \{4, 32, 256\}$ for scheduler-based methods.

### L.1  EXPERIMENTAL SETUP FOR WALL-CLOCK TIME STUDY

We benchmark wall-clock time with a controlled synthetic workload to remove the effects of data loading and I/O. For each configuration (number of tasks $K$ and scheduler refresh rates $R$), we pre-generate a fixed sequence of per-task gradient vectors and loss values directly on the target device, and then feed the same exact tensors, in the same exact order, to every method. We set the gradient dimensionality to 1024. Timing uses a high-resolution clock with a device synchronize before starting and after finishing to capture only on-device compute. We also accumulate the norm of the combined gradient into a scalar accumulator (also known as a scalar sink) so the backend must realize the computation, avoiding lazy evaluation. Each MTL approach is run for 900 steps and repeated 10 times.

## M  EXTENDED RELATED WORK

Multi-task learning (MTL) methods have evolved from simple loss-weighting approaches to larger and more sophisticated optimization techniques that manage task conflict and cooperation (Yang et al., 2023). Early adaptive-weighting approaches sought to balance losses automatically (Vandenhende et al., 2022; Fan et al., 2023), while more recent work modifies gradients directly (Yu et al., 2020).

Table 5: We present wall-clock time (seconds $\pm$ standard deviation) across all $K$ and scheduler refresh rates $R \in \{4, 32, 256\}$. We split results into sub-tables by $R$ for readability. Non-scheduler methods do not depend on $R$, so they are shown in the $R=4$ sub-table and omitted in the $R=32, 256$ subtables to avoid redundancy.

(a) $R=4$ (all methods)

| Method | K=3 | K=6 | K=16 | K=40 |
|---|---|---|---|---|
| Uniform | 0.2656 ± 0.1201 | 0.3240 ± 0.0629 | 0.3798 ± 0.1050 | 0.4054 ± 0.1190 |
| GradNorm | 5.4714 ± 0.7137 | 5.1201 ± 0.6112 | 4.9042 ± 0.5869 | 4.7372 ± 0.9286 |
| MGDA | 97.1081 ± 5.4645 | 121.4371 ± 9.0923 | 132.4913 ± 3.1752 | 134.0878 ± 2.2621 |
| PCGrad | 3.6212 ± 0.3517 | 23.1266 ± 0.8773 | 176.7566 ± 2.8171 | 1127.1337 ± 34.2603 |
| CAGrad | 102.8651 ± 18.3422 | 136.1034 ± 2.4218 | 134.3585 ± 4.0791 | 132.7034 ± 1.2412 |
| AdaTask | 2.1816 ± 0.0934 | 2.1032 ± 0.1012 | 2.2853 ± 0.0718 | 2.2278 ± 0.1370 |
| FAMO | 2.0725 ± 0.2073 | 1.9980 ± 0.1998 | 2.1710 ± 0.2171 | 2.1164 ± 0.2116 |
| FairGrad | 3.8020 ± 0.5703 | 15.2079 ± 2.2812 | 108.1450 ± 16.2218 | 675.9065 ± 101.3860 |
| Nash-MTL | 5.7030 ± 1.1406 | 22.8118 ± 4.5624 | 162.2176 ± 32.4435 | 1013.8598 ± 202.7720 |
| SON-GOKU | 2.0904 ± 0.3506 | 3.6770 ± 0.4974 | 6.3225 ± 0.7895 | 14.3280 ± 1.4073 |
| SON-GOKU + AdaTask | 4.1011 ± 0.4174 | 5.2126 ± 0.6066 | 7.6798 ± 0.7107 | 14.7528 ± 1.8671 |
| SON-GOKU + GradNorm | 7.3223 ± 0.4994 | 8.5898 ± 0.8203 | 12.1065 ± 2.5850 | 16.8329 ± 1.9803 |
| SON-GOKU + PCGrad | 2.3489 ± 0.3258 | 3.5925 ± 0.4100 | 6.1549 ± 0.8461 | 12.5729 ± 1.2657 |

(b) $R=32$ (scheduler-based approaches)

| Method | K=3 | K=6 | K=16 | K=40 |
|---|---|---|---|---|
| SON-GOKU | 1.9896 ± 0.3651 | 3.3202 ± 0.5745 | 6.0897 ± 0.9425 | 12.1432 ± 1.2044 |
| SON-GOKU + AdaTask | 3.7718 ± 0.9654 | 5.0511 ± 0.6531 | 7.5903 ± 1.1920 | 14.5182 ± 2.0660 |
| SON-GOKU + GradNorm | 7.0202 ± 1.0711 | 8.1661 ± 0.9355 | 10.7227 ± 2.2088 | 16.5760 ± 1.8418 |
| SON-GOKU + PCGrad | 1.9834 ± 0.3586 | 3.4971 ± 0.3840 | 6.1395 ± 0.9425 | 10.9097 ± 1.5263 |

(c) $R=256$ (scheduler-based approaches)

| Method | K=3 | K=6 | K=16 | K=40 |
|---|---|---|---|---|
| SON-GOKU | 1.7593 ± 0.2280 | 3.0024 ± 0.3942 | 4.8411 ± 0.7302 | 11.4162 ± 1.6076 |
| SON-GOKU + AdaTask | 3.7224 ± 0.2696 | 4.4548 ± 0.5837 | 7.5276 ± 0.6230 | 13.0608 ± 3.2925 |
| SON-GOKU + GradNorm | 6.0221 ± 1.0418 | 7.8659 ± 0.7917 | 9.5029 ± 1.2168 | 15.6860 ± 2.3680 |
| SON-GOKU + PCGrad | 1.6776 ± 0.4104 | 3.0189 ± 0.7854 | 5.9893 ± 1.3797 | 7.1915 ± 0.2021 |

Task scheduling and grouping methods, though far less popular than adaptive weighting techniques (Torbarina et al., 2023), have contributed to the field by controlling the timing of updates.

## M.1 TUNED LOSS WEIGHTING

From early MTL work it became clear that simply summing task losses often favors one objective at the expense of others (Kurin et al., 2022; Zhao et al., 2024; Mueller et al., 2022), especially when losses have different scales or noise levels. To address this, practitioners manually tuned per-task weight coefficients ($\lambda$-values) to rebalance learning (Argyriou et al., 2007; Ando & Zhang, 2005; Evgeniou et al., 2005; Kang et al., 2011; Liang & Zhang, 2020; Lin et al., 2022; Yu et al., 2021), but this process was laborious and dataset-specific. Thus, researchers began to develop automated methods.

## M.2 ADAPTIVE LOSS WEIGHTING

(Kendall et al., 2018) introduced uncertainty weighting, learning each task's homoscedastic (constant-variance) (Bishop, 2006) noise to scale losses automatically and improve depth and semantics on NYUv2 (Silberman et al., 2012).

GradNorm automatically balances multiple loss functions by tuning each task's gradient magnitude so that all tasks train at comparable speeds (Chen et al., 2018). It does this by introducing a single asymmetry hyperparameter $\alpha$ that governs how much each task's loss is scaled. This eliminates the need for expensive grid searches over manual weights. GradNorm was also a major leap empirically as it surpassed exhaustive search baselines on both regression and classification tasks. Dynamic

Weight Averaging (DWA) extended this idea by adjusting weights based on loss rate of change, reducing oscillations between tasks (Liu et al., 2019).

More recently AdaTask applies task-specific learning rates that adapt to each head's gradient norm, yielding significant gains on multi-label classification benchmarks (Yang et al., 2023).

### M.3 GRADIENT-LEVEL CONFLICT MITIGATION

Rather than rescaling losses, gradient surgery methods alter update directions. PCGrad projects gradients that conflict (negative cosine) onto each other's normal plane, significantly boosting efficiency on supervised vision and RL problems (Yu et al., 2020). CAGrad frames task balance as a min-max optimization, finding updates that maximize the worst-case task improvement (Liu et al., 2021). The Multiple Gradient Descent Algorithm (MGDA) computes a Pareto-optimal convex combination of task gradients, ensuring no task is harmed (Sener & Koltun, 2018). More recent variants such as SAM-GS incorporate momentum into conflict detection, smoothing gradient estimates while preserving the benefits of surgery (Borsani et al., 2025).

### M.4 EMPIRICAL TASK GROUPING

Task grouping aims to decide which tasks should train together so that helpful transfer is amplified and harmful interference is limited. It typically groups tasks into subsets that update jointly, rather than updating all tasks at once. This is different from approaches that keep all tasks active or reweight the joint gradient (adaptive loss weighting, gradient surgery).

Early approaches under this category used round-robin and random sampling-based approaches that ignored any task relationships (McCann et al., 2018; Zamir et al., 2020). Standley et al. (2020) exhaustively searches over small subsets to identify beneficial groupings, demonstrating the potential of selective updates but failing to scale beyond eight tasks due to computational complexity.

Task Affinity Groupings (TAG) (Fifty et al., 2021) performs one joint training run to measure inter-task 'affinity'. It quantifies how an update for task $i$ (its gradient) would change task $j$'s loss, and it uses these cross-effects to select partitions of tasks that should share updates. The key idea is to treat grouping as an outcome of measured gradient interactions.

Ayman et al. (Ayman et al., 2023) train a predictor that maps single-task statistics and dataset features to an estimate of whether two or more tasks should be grouped. They then use that predictor to guide a randomized search over groups, which dramatically reduces the number of multi-task trainings (or 'MTL trials') needed to find a good partition.

Using a completely different approach, Towards Principled Task Grouping (PTG) (Wang et al., 2024) formulates grouping as a mathematical program with a theoretically motivated objective capturing beneficial transfer while respecting resource constraints (e.g., compute budgets). It builds a principled optimization over candidate groups that is meant to generalize across application domains.

Scalable Task Grouping via Training Dynamics (STG-MLT) (Sherif et al., 2024) avoids expensive affinity estimation by extracting Data Maps (Swayamdipta et al., 2020) (simple summaries of training dynamics per task) and then clustering tasks using those features. The clusters are intended to push for positive transfer at larger scale. This approach essentially replaces gradient cross-effects with more compact trajectory features that are cheap to compute and easy to cluster.

## N SPACE COMPLEXITY AND MEMORY USAGE

We analyze the space complexity (and in relation, memory usage) of SON-GOKU during both refresh and non-refresh steps. In our analysis, we distinguish memory usage from other components of training (e.g., parameters, optimizer state, activations) from the memory of the multi-task algorithm. The memory usage of the backbone model is irrelevant in assessing SON-GOKU's space complexity, so our following analysis focuses only on the incremental cost added by the scheduler or MTL optimizer.

SON-GOKU maintains Exponential Moving Averages (EMAs) of gradients within a refresh window for each task. It periodically recomputes pairwise interactions before recoloring and scheduling

| Method | Incremental Space Complexity |
|---|---|
| Uniform (equal weights) | $O(1)$ |
| GradNorm | $O(K)$ |
| AdaTask | $O(Kd)$ |
| FAMO | $O(1)$ |
| PCGrad | $O(Kd) + O(K^2)$ |
| MGDA | $O(Kd) + O(K^2)$ |
| CAGrad | $O(Kd) + O(K^2)$ |
| FairGrad | $O(Kd) + O(K^2)$ |
| Nash-MTL | $O(Kd) + O(K^2)$ |

Table 6: Incremental space complexity (i.e., focused only on method itself) of baseline and state-of-the-art multi-task learning methods. SON-GOKU scales better than the majority of existing methods in terms of space complexity while still faithfully modeling task interference. $K$ represents the number of tasks, $d$ represents the number of shared parameters. "Peak" refers to space complexity at a refresh step, while "persistent" is the space complexity between refreshes.

(Section 4). We refresh the schedule every $R$ steps, update EMAs continously, and then rebuild the interference structure at each refresh and proceed with the new groups.

### N.1 SPACE USAGE ACROSS A REFRESH CYCLE

**Between refreshes (persistent memory).** The scheduler keeps (i) an EMA-based similarity structure and (ii) the current conflict graph and color assignments. With sketched EMAs, based on Ghashami et al. (2016a) of width $r \ll d$, the persistent memory footprint is:

$$\boxed{\text{persistent memory} = O(K^2)}$$

dominated by the smoothed similarity matrix and graph coloring metadata.

**At refresh (peak).** Every $R$ steps we form the dense matrix of task–task interactions and recolor. With sketched EMAs, the peak incremental memory during each refresh is

$$\boxed{\text{peak (refresh) memory} = O(K^2 + Kr)}$$

for the $K \times K$ interference matrix plus $K$ sketch vectors of width $r$. These are released immediately after recoloring. We do not retain $K$ full gradients persistently.

**After refresh.** Only the updated smoothed similarities $K \times K$ and graph coloring metadata remain, returning to the persistent $O(K^2)$ footprint until the next refresh.

### N.2 BASELINES FOR COMPARISON

Below, we summarize the incremental space complexity of common MTL optimizers:

- **Loss reweighting** (e.g., GradNorm, AdaTask): Maintain a few scalars, so $O(K)$

- **Gradient-level Conflict Mitigation and Multi-Objective Solvers** (e.g., PCGrad, CAGrad, MGDA, Nash-MTL): Usually retain $K$ task gradients in the shared parameter space during update calculation, so they usually have a space complexity of $O(Kd)$

We describe the space complexity of SON-GOKU alongside several other baselines and state-of-the-art methods in Table 6.

We can see that many state-of-the-art methods' space complexity scales with $d$. Gradient manipulation and multi-objective methods often incur $O(Kd)$ extra memory because they retain $K$ gradients per each task. As the backbone size $d$ grows, their memory footprint scales linearly with $d$, making them

more computationally expensive to use with larger models. This contrasts SON-GOKU, which has a space complexity that grows mainly with the task count $K$ rather than the model size.

Methods like FAMO, which have lower space complexity than SON-GOKU, keep their memory overhead low by adjusting a single set of task weights rather than modeling *which* tasks should (or should not) be updated together. They do not build a conflict graph or schedule incompatible tasks apart, so strongly interfering tasks are still co-updated and can only be down-weighted. This makes methods like FAMO fast and light on memory, but it provides less structure for avoiding negative transfer, among a plethora of other issues. In settings with strong (and potentially changing) interference, this can yield weaker accuracy than approaches like SON-GOKU that explicitly detect conflicts and adapt over time.

### N.3 EXAMPLE

To illustrate the importance of utilizing projections (among other optimizations) and scaling space complexity without $d$, we describe an example.

#### N.3.1 SETUP

Consider fine-tuning a large shared backbone with $d$ parameters (e.g., a billion parameter encoder) on $K$ downstream tasks. Methods such as PCGrad, MGDA, CAGrad, FairGrad, or Nash-MTL typically form and retain $K$ full task gradients in the shared parameter space at each update. Their incremental memory cost therefore scales as $O(Kd)$, on top of the memory already required by the backbone parameters, optimizer states, and activations.

#### N.3.2 MEMORY USAGE

As $d$ grows, this $O(Kd)$ term rapidly dominates the memory budget. For instance, with a backbone of $d \approx 10^9$ parameters and $K = 20$ tasks, storing $K$ full-precision gradients requires tens of gigabytes of additional memory. In practice, this can make gradient-based conflict mitigation or multi objective solvers difficult to deploy. Practitioners would either shrink the model, reduce $K$, or aggressively trade off batch size and activation memory to stay within device limits.

In contrast, SON-GOKU maintains only (i) EMA-based sketches of each task gradient of width $r \ll d$ and (ii) a $K \times K$ conflict structure and its graph coloring. Between refreshes, the incremental memory is therefore $O(K^2)$ (for smoothed similarities and coloring metadata). At a refresh step, SON-GOKU briefly incurs a peak space complexity of $O(K^2 + Kr)$ to rebuild the interference structure from the sketches, and then immediately releases the data. Importantly, all of these terms depend on $K$ and $r$, but not directly on $d$.

#### N.3.3 TAKEAWAY

In setups with large backbones and many tasks, the extra $O(Kd)$ memory required by some other methods can easily exceed available device memory. This would force practitioners to either simplify the model or abandon conflict-aware MTL altogether. SON-GOKU's space complexity instead grows mainly with the number of tasks and the sketch dimension, allowing it to scale to much larger backbones under the same hardware budget while still modeling task interference and adapting the schedule over time.

### N.4 EXPERIMENTAL VALIDATION

We evaluated throughput and memory usage on an isolated environment with the Taskonomy Tiny subset using a U-Net backbone. Our testing included SON-GOKU against other baselines and state-of-the-art methods.

SON-GOKU is clearly the most resource-efficient option in this comparison. It achieves the highest throughput (processing around 68 images per second on the dataset) while also using the lowes tpeak GPU memory, several gigabytes less than other methods. The only methods that slightly outperforms it in reserved memory is CAGrad, but that comes at a substantial cost in speed, demonstrating that SON-GOKU's graph based scheduling adds far less overhead than other methods. As we've stated

Table 7: Throughput and memory usage per training step for SON-GOKU versus other baselines and state-of-the-art methods. Throughput is measured in images per second and memory statistics are in MB. Results and standard deviation presented from five complete trials

| Method | Throughput (imgs/s) ↑ | Peak mem. (MB) ↓ | Reserved mem. (MB) ↓ |
|---|---|---|---|
| SON-GOKU | **68.5** $\pm$ 1.1 | **1959** $\pm$ 47 | 2332 $\pm$ 62 |
| FAMO | 47.6 $\pm$ 2.6 | 4740 $\pm$ 188 | 5152 $\pm$ 143 |
| CAGrad | 42.1 $\pm$ 1.9 | 2008 $\pm$ 33 | **2284** $\pm$ 97 |
| AdaTask | 41.6 $\pm$ 3.4 | 5108 $\pm$ 121 | 5544 $\pm$ 214 |
| PCGrad | 40.0 $\pm$ 1.5 | 5108 $\pm$ 173 | 5614 $\pm$ 129 |
| MGDA | 36.4 $\pm$ 4.2 | 5065 $\pm$ 142 | 5308 $\pm$ 201 |
| GradNorm | 25.5 $\pm$ 2.1 | 5034 $\pm$ 77 | 5542 $\pm$ 178 |

previously, methods that repeatedly solve multi-objective or projection subproblems face huge caveats in terms of time and space complexity. SON-GOKU's advantages over other methods will scale with larger datasets and backbone models.

## O   ON THE CHOICE OF REFRESH RATE

Let $R \in \mathbb{N}$ denote the *refresh period* (we refresh once every $R$ steps), so the refresh *rate* is $1/R$. A full refresh costs $\Theta(K^2 d)$ time to (re)build the cosine Gram from $K$ EMA vectors in $\mathbb{R}^d$. The EMA matrix uses $\Theta(Kd)$ memory persistently (between refreshes). Peak memory usage during a refresh is $\Theta(Kd) + \Theta(K^2)$. From this, it is clear that increasing $R$ reduces overhead (in terms of both speed and memory usage) linearly.

The conflict graph is built from EMAs accumulated over the refresh window. Longer windows (larger $R$) average out gradient noise and stabilize $\widehat{\rho}$, reducing spurious edges. Shorter windows (smaller $R$) can adapt to changing patterns faster but use fewer effective samples, so $\widehat{\rho}$ can be noisier and the schedule can fluctuate. In our analysis (see Appendix F), global $O(1/\sqrt{T})$ nonconvex convergence does not depend on $\tau$, while the gradient energy for each step is controlled by the effective conflict level $\tau_{\text{eff}}$ of the active set. Reducing conflict between tasks (a byproduct of a well-constructed graph) helps to tighten the bounds of our earlier theoretical analysis. This means that we should pick a value of $R$ large enough for stable estimation but not so large that the graph becomes obsolete.

### O.1   TRAINING DYNAMICS

Early in training, both model parameters and task relations typically evolve very rapidly. A faster refresh rate (small $R$) lets SON-GOKU quickly track and exploit the evolving structure of tasks. This means that SON-GOKU's constructed conflict graphs will lag less behind the true nature of task relationships. Later in training, however, the dynamics slow down and a slower refresh rate will allow EMAs to more effectively average out over time. A slower refresh rate essentially makes SON-GOKU more robust to noisy gradients later in training.

### O.2   STRATEGIES FOR ADJUSTING THE REFRESH RATE

To put it simply, three factors matter most when selecting a refresh rate:

1. **Rate of change in task relations.** If edges in the conflict grpah change often, using a higher refresh rate can help to reduce staleness of the conflict graph and color groups. If relations are stable, a lower refresh rate can suffice.

2. **Noise in EMA estimates.** If gradients are noisy at each step, a larger $R$ value is preferred to stabilize the cosine estimates by calculating EMA over a longer range. If there is little noise (or if there is an excess of available compute), a smaller $R$ value can help to more accurately track the structure of task conflicts.

3. **Overhead budget.** If the scheduler's cost needs to be negligible relative to each task's backpropagation, it is best to select an $R$ to satisfy $C\text{sched}/C\text{probe} \leq \varepsilon$, where $C\text{sched} = \frac{a}{R}K^2 d$ and $C\text{probe} = \frac{b}{R}KG$. Equivalently $Kd \leq \frac{b}{a}\varepsilon G$.

We discuss two simple strategies that can be used to select an appropriate $R$ value.

### O.2.1 ANNEAL THE REFRESH RATE

Start with a relatively high rate (small $R$) and increase $R$ over time. This prevents SON-GOKU's graphs from lagging behind the conflict structure early on in training (as task relationships and parameters are changing rapidly). Later on, larger values of $R$ average out noise in $\widehat{\rho}$ and stabilizes the schedule.

### O.2.2 ADAPT THE REFRESH RATE BASED ON TASK CONFLICT VARIABILITY

Use the observable variability in task conflicts to adjust the refresh rate accordingly. Increase the refresh rate (shrink $R$) if a large fraction of edges in $G_\tau$ flip between consecutive refreshes. Decrease the refresh rate (grow $R$) when edge flips in the graph are rare and EMA rows are stable.

A less computationally expensive alternative is to partially updated only the affected rows and columns of the Gram matrix when a small number of EMA rows cross a certain thresholds. This would have a time complexity of $O(sKd)$ for $s \ll K$. Incorporating such an approach with the existing scheduler could help preserve edges and coloring decisions while reducing refresh costs.

## P INCORPORATING SON-GOKU WITH OTHER TASK AFFINITY MEASURES

### P.1 THEORETICAL ANALYSIS AND DISCUSSION

We instantiate SON-GOKU's conflict score using EMA smoothed gradient cosines. We define an interference coefficient $\rho_{ij} = -\frac{\langle \bar{g}_i, \bar{g}_j \rangle}{|\bar{g}_i|,|\bar{g}_j|}$ and build a conflict graph $G_\tau$ by thresholding at a tolerance $\tau$. We then color this graph and schedule one color class at a time. This particular choice is appealing because it is cheap to maintain online and aligns directly with cosine based gradient surgery methods such as PCGrad. At the same time, the SON-GOKU pipeline itself is more general. It only requires *some* symmetric pairwise "conflict score" that can be thresholded to form a graph. In this section we formalize this modular perspective and explain how richer task affinity measures such as TAG's (Fifty et al., 2021) lookahead loss can be used in place of gradient cosine, without changing the scheduling or convergence guarantees that are important to our theoretical analysis.

### P.1.1 SON-GOKU AS A MODULAR CONFLICT GRAPH BASED SCHEDULER

Recall that SON-GOKU operates in four stages:

1. Estimate pairwise interference. For each pair of tasks $(i, j)$, compute a symmetric score $\rho_{ij}$ that is large when $i$ and $j$ are "in conflict" and small (or negative) when they are aligned or neutral. $\rho_{ij}$ is the negative cosine of EMA smoothed gradients.

2. Build a conflict graph. For a threshold $\tau \in (0, 1)$, define an undirected graph

$$G\tau = (V, E\tau), \quad V = 1, \ldots, K, \quad E\tau = (i, j) : \rho_{ij} > \tau, \tag{112}$$

so that edges mark pairs we do *not* want to update together.

3. Color the graph. Apply Welsh-Powell greedy coloring to obtain color classes $\mathcal{C}_1, \ldots, \mathcal{C}m$, each of which contains no edge. I.e., no pair with $\rho_{ij} > \tau$.

4. Schedule color classes. Cycle through color classes over a refresh window, activating exactly one color class per step.

Crucially, all of the scheduling and convergence results (in Section 5) depend only on the active sets $S_t$ that SON-GOKU chooses at each step, and the actual gradients of tasks within those sets, not on the particular formula used to compute $\rho_{ij}$. Below, we provide a few examples.

**Descent within low conflict groups (Section 5.1).** The key inequality

$$\Big|\sum_{k \in S_t} g_{k,t}\Big|^2 \geq \big(1 - \tau(|S_t| - 1)\big) \sum_{k \in S_t} |g_{k,t}|^2. \tag{113}$$

only assumes that gradients inside $S_t$ are pairwise "$\tau$ compatible" in cosine space. It does not require that $S_t$ came from EMA cosine in any specific way.

**Nonconvex convergence (Section 5.2).** The rate

$$\min_{t < T} \mathbb{E}\big[\|\nabla F(\theta_t)\|^2\big] \lesssim \frac{1 + \tau}{\sqrt{T}} \tag{114}$$

uses the same compatibility assumption and is agnostic to how the groups were found.

**Scheduling properties (Section 5.5).** The bound that SON-GOKU uses at most $\Delta + 1$ colors and thus updates each task at least once every $\Delta + 1$ steps depends only on the graph degree $\Delta$, not on the origin of the edges.

From this viewpoint, SON-GOKU is a modular scheduler that takes any symmetric conflict matrix $(\rho_{ij})$ as input and returns (i) a conflict-based partition of tasks, and (ii) a schedule with bounded staleness. Our cosine based construction is just one concrete approach for this interface.

### P.1.2 TAG AND OTHER TASK AFFINITY MEASURES AS CONFLICT SCORES

TAG (Fifty et al., 2021) defines an affinity at each step between tasks $i$ and $j$ by measuring how an update for task $i$ affects task $j$'s loss. Formally, TAG trains all tasks together once, and during this time it repeatedly: (i) takes a gradient step on task $i$ with $\theta' = \theta - \eta g_i$, (ii) evaluates task $j$'s loss at $\theta'$, and (iii) uses the change in loss

$$\Delta L_j^{(i)} \approx L_j(\theta') - L_j(\theta) \tag{115}$$

as a measure of how much $i$ helps or hurts $j$.

Averaging this quantity over training results in an affinity matrix $A_{ij}$, with positive values meaning that updates on $i$ tend to improve $j$, and negative values indicating harmful transfer between tasks. TAG then clusters tasks based on this matrix to decide which tasks should share a network.

Other recent grouping methods construct alternative affinity matrices from training. For example, STG-MTL uses data maps that summarize each task's example-wise training trajectory and then clusters tasks based on these representations, yielding a similarity or affinity between tasks that can be used for grouping (Sherif et al., 2024).

All of these approaches (TAG, STG-MTL, meta grouping, and others) produce symmetric pairwise scores that can be interpreted as how much task $i$ helps task $j$. SON-GOKU can treat any such affinity as a replacement for the cosine-based interference score.

Now, we will provide a more formal analysis. Suppose we have a symmetric affinity matrix $Aij$ where larger values represent more beneficial relationships and smaller values mean more harmful relationships. We can define a conflict score $\rho_{ij} = f(Aij)$ for any monotone decreasing function $f$ (so that more harmful pairs get larger $\rho_{ij}$). The conflict graph is then

$$G_\tau = \big\{(i,j) : \rho_{ij} > \tau\big\} = \big\{(i,j) : A_{ij} < f^{-1}(\tau)\big\}. \tag{116}$$

All subsequent SON-GOKU steps (graph coloring, cyclic scheduling, and the use of $\Delta$ in the bounded staleness theory) operate purely on $G\tau$ and are therefore unchanged.

### P.1.3 TAG AS A LOSS LEVEL PROXY FOR GRADIENT CONFLICT

TAG's lookahead loss has a natural connection to SON-GOKU's cosine base dinterference. Consider a small gradient step on task $i$ with $\theta' = \theta - \eta g_i(\theta)$ and examine task $j$'s loss

$$\Delta L_j^{(i)} := L_j(\theta') - L_j(\theta) \tag{117}$$

A first order Taylor expansion gives

$$\Delta L_j^{(i)} \approx -\eta \langle g_i(\theta), g_j(\theta) \rangle + O(\eta^2).$$ (118)

Thus, up to higher order terms, TAG's affinity at each step is proportional to the negative inner product between gradients.

If $\langle g_i, g_j \rangle > 0$ (aligned gradients), then a step on $i$ decreases $L_j$, so $\Delta L_j^{(i)} < 0$. This means TAG observes positive affinity. On the other hand, if $\langle g_i, g_j \rangle < 0$ (conflicting gradients), then a step on $i$ increases $L_j$, so $\Delta L_j^{(i)} > 0$. So, TAG observes negative affinity.

In contrast, SON-GOKU's default interference coefficient

$$\rho_{ij}^{\cos} = -\frac{\langle \tilde{g}_i, \tilde{g}_j \rangle}{|\tilde{g}_i||\tilde{g}_j|}$$ (119)

is a normalized version of this inner product, averaged via EMA over recent steps. To compare the two, TAG's score is a loss-level, unnormalized directional derivative. Meanwhile, SON-GOKU's $\rho_{ij}^{\cos}$ is a gradient-level, normalized first order approximation.

For small $\eta$ and when we average over enough steps to approximate population quantities, both measure are monotone transforms of the same underlying signal. That is, the sign and magnitude of $\langle g_i, g_j \rangle$. Using TAG's affinity as $\rho_{ij}$ therefore replaces SOn-GOKU's simple cosine with a richer but more expensive signal that directly reflects loss changes.

This interpretation also clarifies why TAG and SON-GOKU are compatible. TAG's lookahead loss is simply a more expressive extimator of the same quantitty that gradient cosine is trying to capture (i.e., how much does $i$ help or hurt $j$).

### P.1.4 CHANGES TO SON-GOKU'S GUARANTEES BY DIFFERENT AFFINITY MEASURES

We have discussed which arguments presented in our theoretical analysis (particularly, Sections 5.1, 5.2, 5.4) are unaffected by the use of different affinity measures like TAG. However, the one part of the theory that *does* depend on the specific cosine-based construction is the graph recovery analysis, where we prove that EMA cosines concentrate around population cosines and that thresholding them recovers the "true" conflict graph with high probability.

If we replace cosine with TAG's affinity (or with other training dynamics-based approaches such as STG-MTL) the structure of our proof would remain similar but the technical details change. For one, instead of bounding the deviation of empirical cosines from population cosines, we would need concentration bounds for the chosen affinity measure. We would again assume a margin around the threshold (e.g., conflicting pairs have $Aij \leq c^\star - \gamma$, aligned pairs have $Aij \geq c^\star + \gamma$). Finally, we would show that, with enough effective samples per pair we can ensure the uniform estimation error is below $\gamma$, so that the estimated graph matches the population one.

TAG already averages loss changes for each step across changes, and Fifty et al. 2021's own analysis shows that this averaged affinity is stable enough to create grouping decisions in practice. Deriving a formal concentration result for TAG's affinity would be an interesting extension for future works. But it is of course orthogonal to the core contribution of SON-GOKU. Once any affinity measure yields a reliable conflict graph, our earlier graph coloring and scheduling theory applies without any changes.

### P.1.5 COMPUTATIONAL CONSIDERATIONS AND REFRESH WINDOWS

Using TAG style scores as $\rho_{ij}$ changes the cost profile of the scheduler but not its structure. TAG requires extra forward and backward passes to evaluate lookahead losses, or a preliminary joint training run whose logs are replayed to compute affinities (Fifty et al., 2021). Our cosine-based approach, by contrast, is fully online. It maintains EMA gradients and sketches and refreshes the conflict graph periodically with amortized overhead $O\left(\frac{Kr(d+K)}{R}\right)$ per step. However, SON-GOKU already separates measurement from scheduling via the refresh window $R$. We only rebuild the conflict graph every $R$ steps and reuse the coloring in between. This design is naturally compatible with more expensive.

Table 8: Validation performance on Taskonomy for SON-GOKU with cosine-based affinities versus SON-GOKU with TAG-based affinities using a U-Net backbone.

| Metric | SON-GOKU (cosine) | SON-GOKU + TAG |
|---|---|---|
| **Depth (Euclidean)** | | |
| AbsRel ↓ | $0.5432 \pm 0.0068$ | $\mathbf{0.5351} \pm 0.0115$ |
| MAE ↓ | $4.0787 \pm 0.071$ | $\mathbf{4.0052} \pm 0.110$ |
| RMSE ↓ | $18.6199 \pm 0.29$ | $\mathbf{18.3034} \pm 0.38$ |
| **Depth (Z-buffer)** | | |
| AbsRel ↓ | $0.5731 \pm 0.0074$ | $\mathbf{0.5622} \pm 0.0129$ |
| MAE ↓ | $4.0494 \pm 0.069$ | $\mathbf{3.9685} \pm 0.115$ |
| RMSE ↓ | $18.5378 \pm 0.27$ | $\mathbf{18.1485} \pm 0.41$ |
| **Edge Occlusion** | | |
| BCE ↓ | $0.0995 \pm 0.0032$ | $\mathbf{0.0990} \pm 0.0071$ |
| F1 ↑ | $0.1466 \pm 0.010$ | $\mathbf{0.1532} \pm 0.019$ |
| Precision ↑ | $\mathbf{0.5399} \pm 0.021$ | $0.5324 \pm 0.038$ |
| Recall ↑ | $0.0870 \pm 0.0065$ | $\mathbf{0.0957} \pm 0.0127$ |
| **Edge Texture** | | |
| BCE ↓ | $\mathbf{0.0779} \pm 0.0010$ | $0.0781 \pm 0.0018$ |
| F1 ↑ | $\mathbf{0.9734} \pm 0.0007$ | $0.9729 \pm 0.0015$ |
| Precision ↑ | $\mathbf{0.9772} \pm 0.0008$ | $0.9766 \pm 0.0016$ |
| Recall ↑ | $\mathbf{0.9768} \pm 0.0007$ | $0.9762 \pm 0.0014$ |
| **Keypoints2D** | | |
| MSE ↓ | $0.0039 \pm 0.0002$ | $\mathbf{0.0038} \pm 0.0000$ |
| **Surface Normals** | | |
| 11.25° within ↑ | $0.4262 \pm 0.0079$ | $\mathbf{0.4364} \pm 0.0135$ |
| 22.5° within ↑ | $0.6432 \pm 0.0068$ | $\mathbf{0.6535} \pm 0.0111$ |
| 30° within ↑ | $0.7340 \pm 0.0059$ | $\mathbf{0.7428} \pm 0.0094$ |
| Mean angle (deg) ↓ | $22.9079 \pm 0.31$ | $\mathbf{22.3352} \pm 0.48$ |
| Median angle (deg) ↓ | $14.5397 \pm 0.27$ | $\mathbf{14.1300} \pm 0.44$ |
| **Principal Curvature** | | |
| L1 ↓ | $0.0691 \pm 0.0014$ | $\mathbf{0.0672} \pm 0.0026$ |
| **Reshading** | | |
| MAE ↓ | $0.1352 \pm 0.0023$ | $\mathbf{0.1330} \pm 0.0039$ |

One can run a TAG-like procedure periodically (or once at a few milestones), obtain $Aij$, map it to $\rho_{ij}$, and then reuse the resulting graph and coloring for many steps. As long as the underlying task relationships do not drift too quickly between these TAG refreshes, the assumptions used in our analysis (bounded drift, concentration within a refresh window) remain reasonable. We provide further justification for the assumptions in their respective appendices.

In this sense SON-GOKU can be viewed as a scheduler that amortizes the cost of any affinity estimator (TAG, STG-MTL, meta grouping, etc.) over many optimization steps, while preserving its descent, convergence, and bounded staleness guarantees.

## P.2 EXPERIMENTAL ANALYSIS

We benchmark SON-GOKU with TAG-style affinities on the Taskonomy dataset (focusing on NN tasks). For TAG lookahead loss, we apply a 0.1 step size when probing inter-task effects. Training and test was repeated across two random seeds, and results are compared against a baseline of SON-GOKU with its default cosine-based approach. Results are presented in Table 8

Across almost all metrics, SON-GOKU with TAG matches or slightly improves on the cosine-based version. The gains are greatest for depth and surface normal prediction, where using TAG-based affinities leads to consistently lower errors and higher within angle scores, while performance on the

Table 9: Base SON-GOKU versus SON-GOKU + TAG in terms of speed. Amortized runtime per training step (ms) averaged across 10 trials.

| Method | Amortized ms / step $\downarrow$ |
|---|---|
| No Refresh (Baseline) | $\mathbf{149.28} \pm 1.63$ |
| SON-GOKU (cosine) | $193.41 \pm 6.27$ |
| SON-GOKU + TAG | $255.29 \pm 17.84$ |

remaining tasks is either similar or shows small improvements. There are a few cases (such as edge texture) where the cosine version does very slightly better, but the differences are extremely small. Overall, this shows that using TAG's more in-depth lookahead loss into SON-GOKU preserves the benefits of the scheduler and can provide small improvements in performance across many tasks. This is due to TAG providing a more detailed estimate of how tasks help or hurt eachother, which lets SON-GOKU separate conflicting pairs more accurately, as discussed in Section P.1.3.

We also measured the amortized speed of training with TAG loss. As seen in Table 9, it is significantly slower than training using our standard cosine-based approach. This supports the theoretical analysis we presented previously. This gap in speed comes from the extra forward and backward passes needed to measure TAG's lookahead loss, while cosine-based SON-GOKU only reuses gradients it has already computed. This means that using TAG (and similar affinities) might provide a bit more performance at a very big cost in speed, while the cosine version keeps most of the benefit of scheduling with much lower runtime overhead.

# Q ADDITIONAL ANALYSIS OF BENEFITS FROM NON-CONFLICT GROUPING

## Q.1 RELATED WORK

A consistent theme in the multi-task learning (MTL) literature is that not all task should be trained together in a single shared model, and that grouping compatible tasks while separating incompatible ones tends to improve performance. This directly supports the idea that non-conflict groups (sets of tasks whose gradients or transfer effects are mutually aligned) are beneficial.

Standley et al. 2020 systematically study which tasks should be learned together by exhaustively evaluating subsets of vision tasks and measuring cross-task transfer. They show that some task pairs consistently help each other, while others consistently harm each other when trained jointly, even with strong backbones. Their proposed framework assigns tasks to a small number of networks so that cooperating tasks share a network and competing tasks are separated, achieving better accuracy versus a single model that uses all tasks and versus purely single-task baselines. This is a direct empirical demonstration that forming groups of mutually beneficial tasks, rather than mixing all tasks, leads to better generalization.

Fifty et al. 2021 (TAG) push this further by defining a gradient-based task affinity. They measure how a small gradient step for task $i$ changes task $j$'s loss, and average these "look ahead" effects over training. Tasks with positive mutual influence (updates on $i$ tend to reduce $j$'s loss) are grouped together, while tasks with negative or weak influence are separated. TAG shows that such affinity based groupings yield lower test loss than joint training and random groupings, while being far cheaper than brute-force search over subsets. Conceptually, TAG's groups are exactly non-conflict groups at the loss level. Inside a group, most cross-task updates help or at least do not hurt eachother.

Earlier theoretical work on clustered MTL also supports our work. Jacob et al. 2008 assume that tasks are partitioned into latent clusters with similar linear predictors, and they design a convex penalty that encourages such clustering. Tasks within a cluster are forced to share parameters, while different clusters are only weakly coupled. They show that, when the clustering assumption is approximately correct, this clustered sharing outperforms both fully shared and fully independent models. Although this is formulated at the level of parameters rather than gradients, it encodes the same idea. Tasks that point in similar directions should be grouped, and others should not (Jacob et al., 2008). Results from multi-task feature learning architectures (e.g., Argyriou et al. 2007) provide related generalization bounds that improve when tasks share a low-dimensional subspace, again matching the intuition that aligned tasks should be grouped in a shared representation.

More recently, work on inter-task gradient noise provides an optimziaiton-based justification for separating conflicting tasks. Fan et al. 2023 identifies inter-task gradient noise (ITGN) as a key factor behind insufficient training in MTL. Gradients from other tasks can behave like noise for a given task, degrading its effective signal. By defining a gradient-to-noise ratio (GNR) and maximizing it per task, MaxGNR shows that reducing ITGN improves test performance on standard MTL benchmarks. While MaxGNR does not explicitly form task groups, its analysis supports the same idea. Mixing strongly conflicting gradients is harmful, and methods that avoid such mixtures (either by reweighting or by grouping) should see optimization and generalization benefits.

Overall, these works collectively support the central premise of Appendix Q and the focus of SON-GOKU as a whole. Identifying and grouping tasks that do not conflict (and separating those that do) is a key factor of both optimization stability and generalization performance. SON-GOKU's non-conflict groups can be seen as an explicit, gradient level realization of the kind of beneficial groupings that these prior methods either search for or implicitly encode.

### Q.2 THEORETICAL ANALYSIS

We briefly summarize the aspects of SON-GOKU's theory that are most relevant to non-conflict grouping, and then add a simple calculation that makes the benefit of grouping more explicit.

#### Q.2.1 EXISTING GUARANTEES FOR NON-CONFLICT GROUPS

Let $g_{k,t} = \nabla\theta\ell_k(\theta_t)$ denote the gradient of task $k$ at step $t$. SON-GOKU defines a conflict graph by thresholding an interference coefficient (based on EMA cosine similarity) and schedules one $tau$-compatible group $S_t$ at a time, where $\tau$-compabability means that pairwise cosines within $S_t$ are bounded by below $-\tau$. Under this condition, we prove the following important inequality (Section 5.1, Appendix E). If $\tau(|S_t| - 1) < 1$, then

$$\left|\sum_{k\in S_t} g_{k,t}\right|^2 \geq \left(1 - \tau(|S_t| - 1)\right)\sum_{k\in S_t}|g_{k,t}|^2. \tag{120}$$

This shows that within a non-conflict group, destructive cancellation is quantitatively bounded. The aggregate gradient cannot flip to ascent, and its norm remains comparable to the total gradient energy of the group.

Using this property, SON-GOKU's convergence analysis establishes that (i) the algorithm preserves the standard $O(1/\sqrt{T})$ non-convex rate of SGD (up to a constant depending on $\tau$), and (ii) a scheduled sequence of group updates over a refresh window achieves a descent bound that is never worse than a single mixed update using all tasks at once, and strictly better when cross-group interactions are sufficiently negative. Appendix F further connects the structure of conflicts to the average gradient energy

$$\frac{1}{T}\sum_{t=1}^{T}\sum_{k\in S_t}|g_{k,t}|^2, \tag{121}$$

showing that lower cumulative interference yields tighter bounds on this quantity and thus moves us toward updates with higher effective signal-to-noise ratio.

The recovery analysis in Appendix B then guarantees that, under mild assumptions on drift and margin around the threshold, SON-GOKU's EMA-based conflict estimates recover the underlying low-conflict structure with high probability. This means that the non-conflict groups SON-GOKU uses in practice are a statistically grounded approximation to the true alignment structure of tasks, rather than arbitrary partitions.

### Q.2.2   SIGNAL-TO-NOISE GUARANTEES FOR NON-CONFLICT GROUPS

To complement these results, consider a simple toy model for two tasks that mirrors the gradient noise perspective in MaxGNR (Fan et al., 2023). Suppose

$$g_1 = \mu_1 + \xi_1, \qquad g_2 = \mu_2 + \xi_2, \tag{122}$$

where $\mu_k = \mathbb{E}[g_k]$ is the mean descent direction for task $k$, and $\xi_k$ is zero mean isotropic noise with covariance $\sigma^2 I$, independent across tasks. Define the signal-to-noise ratio (SNR) at each step of a gradient $g$ as

$$\mathrm{SNR}(g) \; := \; \frac{|\mathbb{E}[g]|^2}{\mathbb{E}\big[|g - \mathbb{E}[g]|^2\big]}. \tag{123}$$

We consider the following approaches:

1. Single mixed update (all tasks at once). Let $g_{\mathrm{mix}} = g_1 + g2$. Then
$$\mathbb{E}[gmix] = \mu_1 + \mu2, \quad \mathbb{E}|gmix - \mathbb{E}[g_{\mathrm{mix}}]|^2 = 2\sigma^2 d, \tag{124}$$
   where $d$ is the dimension. The SNR of the mixed update is proportional to
$$|\mu_1 + \mu_2|^2 \; = \; |\mu_1|^2 + |\mu_2|^2 + 2|\mu_1|, \; |\mu_2| \cos\theta \tag{125}$$

2. Sequential non-conflict updates (update one group at a time). If we perform two sequential updates, first with $g_1$ and then with $g_2$, the average per-step SNR is proportional to $|\mu_1|^2 + |\mu_2|^2$, since each step sees noise of variance $\sigma^2 d$.

Comparing the two,

$$\mathrm{SNR}_{\mathrm{mix}} \; \leq \; \mathrm{SNR}_{\mathrm{seq}} \quad \Longleftrightarrow \quad |\mu_1|^2 + |\mu_2|^2 + 2|\mu_1||\mu_2|\cos\theta \; \leq \; |\mu_1|^2 + |\mu_2|^2. \tag{126}$$

Thus, whenever the mean descent directions of two tasks are non-positively aligned ($\cos\theta \leq 0$), a single mixed update has no better signal-to-noise ratio than sequential updates, and in the genuinely conflicting case ($\cos\theta < 0$) it is strictly worse.

SON-GOKU's non-conflict groups are precisely designed so that within each group, pairwise cosines are bounded away from strongly negative values. This ensures that inside a group, the aggregate gradient faces limited cancellation (by the group-level inequality above), and across groups, strongly negative interactions are never mixed in the same update, so we avoid exactly the SNR degradation shown by this toy model.

This simple argument is fully consistent with the empirical and theoretical trends in the literature. Standley et al. 2020 and TAG show that grouping compatible tasks and separating incompatible ones improves test performance while Fan et al. 2023 connects inter-task gradient noise to under-optimization in MTL. SON-GOKU's non-conflict groups give a principled way to implement this insight

## R   SCALING OF SON-GOKU TO LARGER BACKBONES

### R.1   THEORETICAL ANALYSIS AND REASONING

In this subsection we study how SON-GOKU behaves when we increase the capacity of the shared backbone (e.g., by widening or deepening the network). Intuitively, a larger backbone gives the model more representational degrees of freedom, which can make it easier for different tasks to carve out partially separate feature subspaces. This has two important consequences. For one, task interference can genuinely shrink as capacity grows, so the absolute room for improvement of any multi-task learning (MTL) method, including SON-GOKU, may decrease. At the same time, task gradients are still coupled through shared optimization, data, and regularization, so interference does not necessarily vanish, especially with many tasks or mismatched objectives.

Below we formalize this picture and explain why SON-GOKU is designed to (i) reduce to joint training when interference is negligible and (ii) remain robust and useful when conflicts persist, even on large backbones.

### R.1.1 Effect of Model Capacity on Task Interference

Consider $K$ tasks trained on a shared backbone, with (stochastic) per-task gradients $g_{i,t} \in \mathbb{R}^d$ at step $t$. SON-GOKU summarizes interference via cosine-based interference coefficients

$$\rho_{ij,t} := -\frac{\langle \tilde{g}_{i,t}, \tilde{g}_{j,t} \rangle}{\|\tilde{g}_{i,t}\| \|\tilde{g}_{j,t}\|} \tag{127}$$

Increasing the backbone size changes the geometry of these gradients. A wider or deeper network can essentially allocate separate channels or subspaces to different tasks, so that conflicting updates get placed into distinct parts of the representation. In this overparameterized setting, the model may approximate each task extremely well, allowing gradients for different tasks to decouple and align more often. Empirically and theoretically, interference is very well known to depend strongly on the match between tasks and the shared representation. Some tasks cooperate, others compete, and this structure does not disappear just because we add parameters (Standley et al., 2020).

Even with a large backbone, there are several reasons why negative interactions can persist. One big issue comes down to shared optimziation and finite resources. The backbone is still trained with a single optimizer on a finite dataset with finite training steps and explicit or implicit regularization. The optimization path couples tasks through shared layers, so gradients remain misaligned for objectives with different inductive biases (e.g., geometric versus semantic, short-term versus long-term goals). Additionally, when the number of tasks $K$ is large, increasing capacity is not enough to give each task a completely separate "subnetwork." Some features must be reused, so there is competition over shared directions, and conflicts can reappear as tasks compete for the same representational dimensions (Pascal et al., 2021). Furthermore, if some tasks are over-represented in the data or have larger losses, their gradients dominate updates and push shared features toward their own optima, potentially harming less frequent or harder tasks, regardless of the backbone's absolute size.

Looking at this from the lens of gradient geometry, larger backbones often shift the cosine distribution between tasks toward zero and reduce variance, but they do not guarantee that all cosines become small and nonnegative. SON-GOKU is designed to exploit whatever conflict structure remains while not overreacting when conflicts are rarer.

### R.1.2 SON-GOKU Under Scenarios with Low Interference

Suppose we are in a scenario with almost no interference where, for most pairs $(i, j)$,

$$\mathbb{E}\big[\cos(g_{i,t}, g_{j,t})\big] \geq -\varepsilon, \tag{128}$$

with $\varepsilon$ small and cosines concentrated near zero or positive values. For a reasonable threshold $\tau > 0$, the probability that the EMA-based $\rho_{ij,t}$ exceeds $\tau$ becomes very small. In this limit, (i) the conflict graph $G_\tau$ would become very sparse or even empty; (ii) Welsh-Powell coloring produces one or very few color classes. In the extreme case of no edges, all tasks share a single color; and (iii) SON-GOKU's scheduler then activates all tasks together at each step, so the update coincides with standard joint training.

If $E_\tau = \emptyset$, the active set at every step is

$$S_t = 1, \ldots, K \tag{129}$$

and the update rule is identical to vanilla MTL with a shared backbone and summed (or weighted summed) loss. The only remaining overhead is maintaining EMA statistics and coloring a graph with no edges.

This clearly shows that SON-GOKU will do no harm at larger backbones, given that $\tau$ is properly configured. If the gradient geometry exhibits no strongly negative cosines (for the chosen $\tau$), SON-GOKU reduces to joint training and does not introduce artificial conflicts or distortions. As previously mentioned, the only configurable parameter in this setting is the threshold $\tau$. As capacity grows and cosines shift closer to zero, it can be sensible to use a slightly more conservative threshold (e.g., larger $\tau$) so that we still isolate genuinely harmful pairs without splitting tasks unnecessarily. As long

as there is a margin separating truly conflicting pairs from near-conflicting paris, such a choice of $\tau$ exists.

### R.1.3 Comparison to Methods That Always Modify Gradients

Many existing MTL methods *always* modify gradients, regardless of whether conflicts are present. Gradient surgery and multi-objective methods such as PCGrad and CAGrad change the direction of the update by projecting or optimizing over gradients at every step, to avoid negative inner products or approximate optimal directions (Sener & Koltun, 2018; Yu et al., 2020; Liu et al., 2021; Navon et al., 2022; Ban & Ji, 2024; Shi et al., 2023; Borsani et al., 2025). Methods like GradNorm automatically rescale task losses or gradients (Chen et al., 2018; Kendall et al., 2018; Liu et al., 2019; Yang et al., 2023; Liu et al., 2023; Fan et al., 2023; Liang & Zhang, 2020; Lin et al., 2022). There exist a plethora of other methods that similarly modify gradients or other parameters at all times (Caruana, 1997; Ando & Zhang, 2005; Evgeniou et al., 2005; Argyriou et al., 2007; Kang et al., 2011; Ruder, 2017; Pascal et al., 2021).

These approaches can be very effective when conflicts and imbalances are strong, but in settings with low interference and high capacity they can become unstable. For example, when cosines are small and positive, the true conflict is near zero. However, stochastic noise can make some cosines appear slightly negative. A surgery method like PCGrad sees these as "conflicts" and projects away components that were actually harmless, adding noise to the optimization path. Methods that explicitly solve a multi-objective problem at each step optimize a combination of gradients whose directions become almost collinear when interference is low. In that case the optimization can become counterproductive. Basically, tiny numerical differences in gradients can lead to large changes in the chosen combination, creating unstable trajectories that are not justified by any genuine underlying conflict (Sener & Koltun, 2018; Yu et al., 2020; Liu et al., 2021; Navon et al., 2022; Shi et al., 2023; Ban & Ji, 2024; Borsani et al., 2025). Finally, as backbones get larger, gradient norms often shrink and their distribution shifts. Methods that rely heavily on relative magnitudes must then be retuned or recalibrated, and may continue to reweight tasks even when they would naturally co-exist without interference (Chen et al., 2018)

By contrast, SON-GOKU's design is deliberately based on events. It monitors normalized cosines which are invariant to global rescaling of gradients and less sensitive to noise at each step. It only introduces edges in the conflict graph once cosines cross a threshold and stay there consistently enough to not be erased by smoothing. And in the case when cosines are all near zero or positive, the conflict graph simply goes away, and scheduling essentially becomes joint training.

In other words, as we scale up the backbone and genuine conflicts no longer exist, SON-GOKU turns itself off, while many other methods continue to adjust gradients based on weak and noisy signals.

### R.1.4 Larger Backbones and Conflict Graph Estimation

The graph theory part of SON-GOKU (graph construction, coloring, and scheduling) depends on the actual interference structure, not on the specific estimator used. In Appendix B.4 of the main paper we show that, under mild assumptions, EMA cosines naturally concentrate around the true (or "population") cosines and that thresholding them recovers the true conflict graph with high probability.

Increasing model capacity typically helps this estimation step, for two reasons. Firstly, it enables higher signal-to-noise ratio in cosines. As the backbone gets larger and representations become more structured, gradients for each task often become more stable across steps. This reduces the variance of cosine estimates and makes it easier to distinguish pairs that are persistently conflicting (true gradient cosine is strongly negative) from pairs that are truly neutral or cooperative (true gradient cosine is near zero or positive). Under the same number of samples per pair, the probability that an EMA cosine is misclassified relative to the threshold $\tau$ decreases. This effectively improves the sample complexity of graph recovery. The second reason that increasing model capacity actually helps SON-GOKU is that it enables sparser conflict graphs and better scheduling guarantees. When capacity pushes more task pairs into the non-conflicting regime, the conflict graph becomes sparser and its maximum degree $\Delta$ decreases. By standard graph-coloring results, any proper coloring uses at most $\Delta + 1$ colors. A smaller $\Delta$ thus implies there are fewer groups, so each group is larger and

Table 10: Validation loss averaged across tasks on Taskonomy Tiny for different encoder widths (UNet-style encoder).

| Encoder width | Joint | SON-GOKU |
|---|---|---|
| 64 | $5.232 \pm 0.020$ | **5.203** $\pm 0.018$ |
| 128 | $5.050 \pm 0.018$ | **4.940** $\pm 0.017$ |
| 256 | $4.900 \pm 0.015$ | **4.830** $\pm 0.015$ |

more similar to joint training. It also implies each task is updated at least once every $\Delta + 1$ steps, so smaller $\Delta$ directly improves guarantees about how fresh the graph is.

From the standpoint of our theoretical analysis (Section 5), this means that larger backbones either make the conflict graph easier to estimate for the same number of refreshes, or make the graph itself simpler (low degree), which directly improves scheduling properties. In the settomg where conflicts still exist but are less noisy, SON-GOKU benefits greatly. We get cleaner separation between conflicting and non-conflicting pairs, and we need fewer colors to isolate those conflicts.

### R.1.5 PERSISTENT INTERFERENCE FOR HIGH CAPACITY MODELS WITH MANY TASKS

While the extreme case of no interference is conceptually simple, many realistic scenarios with large backbones still exhibit substantial negative transfer. When $K$ is large and tasks optimize for different invariances or focus on different aspects of the input, some gradients remain fundamentally incompatible, no matter how wide the backbone is. This is reflected in persistent clusters of negative cosines and in empirical studies showing that some tasks should not be learned together even when using fairly powerful models (Standley et al., 2020). With a single optimizer and a finite number of steps, training is a multi-objective problem in the sense of Sener and Koltun. There is a Pareto front of trade-offs that cannot be simultaneously optimized, independent of capacity (Sener & Koltun, 2018). In such cases, conflicts will persist despite over parameterization. Finally, when some tasks see more data or larger losses, their gradients dominate shared layers. This can lead to a behavior where dominant tasks dictate the representation, harming others even in high-capacity networks.

In these settings, our earlier theoretical results continue to apply. The point is, nothing in our analysis assumes a small backbone. What matters is the geometry of gradients, not the absolute size of the parameter space. As we discuss in Appendices R.1.2 and R.1.4, SON-GOKU will either do no harm or continue to provide the same theoretical benefits we discussed previously.

### R.2 EXPERIMENTAL ANALYSIS

For the experimental analysis we evaluate SON-GOKU against a joint training baseline on the Taskonomy dataset. Specifically, we test each approach on a UNet-style encoder across three different settings, where each setting sets a different backbone size for the encoder (64 vs 128 vs 256 neurons). We also evaluate the speed of SON-GOKU across $R$ values of 8, 16, 32, and 64 with these different backbone sizes.

Table 10 contains the results (in terms of loss) for SON-GOKU against joint training averaged across five trials. We observe that, as we widen the encoder from 64 to 128 and 256 channels, both joint training and SON-GOKU improve, but SON-GOKU consistently does better, and the size of the gap follows the qualitative behavior predicted in our earlier analysis. At 64 channels, the backbone is clearly limited in its capacity, so even though there is interference, the model cannot fully exploit cleaner updates. At 128 channels, the backbone is expressive enough that separating conflicting tasks in time and space actually pays off. This is where we see the largest improvement over joint training. At 256 channels, performance improves again for both methods, but the gap shrinks. As model capacity grows, many conflicts are resolved by the large available representation, and SON-GOKU approaches behavior like joint training while still having a small benefit when there are remaining conflicts (Appendix R.1.2). The training setup employed here only used 4 tasks. In setups with tens (or even hundreds) of tasks, it would of course take a much larger backbone for conflicts to be resolved by the larger parameter space alone.

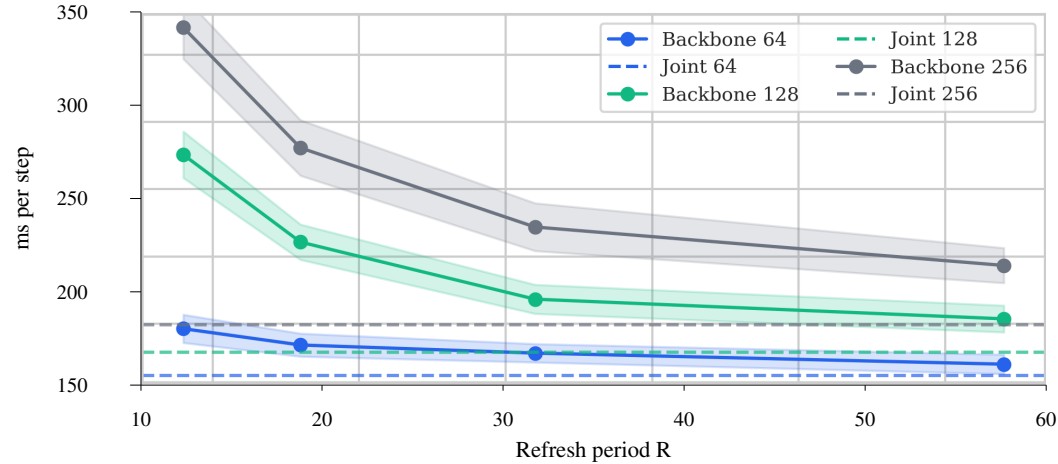

Figure 2: Evaluation of SON-GOKU's speed with varying $R$ (8, 16, 32, 64) on different backbone widths. Plotted against the joint training baseline ($R = \infty$). Highlighted regions represent standard deviation from 15 separate trials. This data was collected during training on Taskonomy Tiny subset.

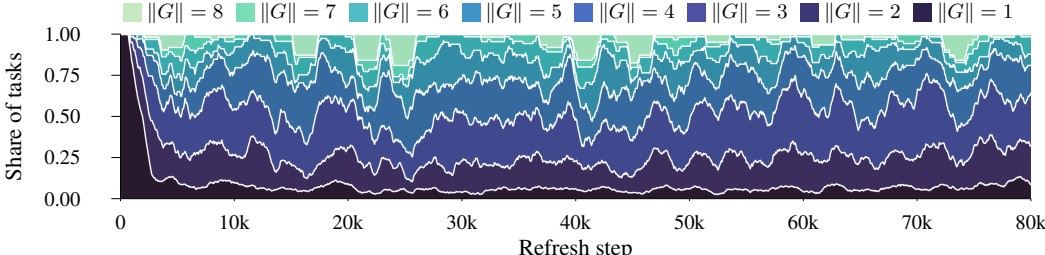

Figure 3: How the fraction of tasks assigned to each group size evolves over the refresh steps in training. This is a stacked area plot showing how the proportion of tasks in each group size $\|G\|$ evolves during training.

For $R = 32$ or $64$, the time for each step is close to the joint training baseline (where $R$ is essentially $\infty$) for all backbone widths, matching our analysis that the extra work scales like $O(Kr(d + K)/R)$ and shrinks as $R$ grows. This demonstrates that SON-GOKU can effectively track interference while adding negligible runtime cost, even for larger backbones.

## S    TRENDS OF SON-GOKU THROUGHOUT TRAINING

In this section we provide detailed visualizations of how SON-GOKU and its components evolve throughout the training process. We train a U-Net model with SON-GOKU on the Taskonomy dataset to gain deeper insight into its behavior, especially in many-task settings. We train for three epochs with a batch size of 64.

## S.1 Task Grouping and Conflict Trends

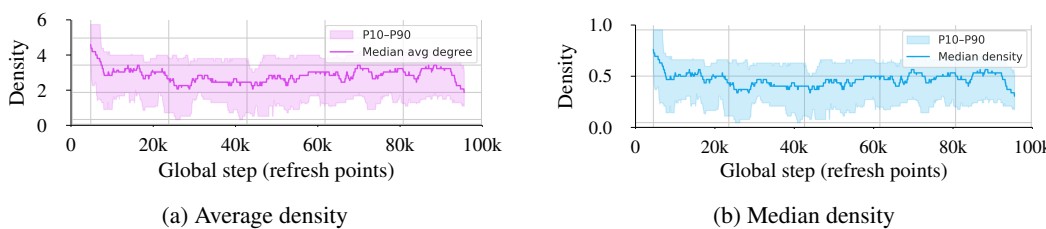

(a) Average density
(b) Median density

Figure 4: Conflict sparsity during training. Subfigure (a) plots the median average node degree (magenta line) with its 10th–90th percentile band (magenta). Subfigure (b) shows the median edge density of the task conflict graph at each refresh step (blue line) with the 10th–90th percentile range across runs (blue band)..

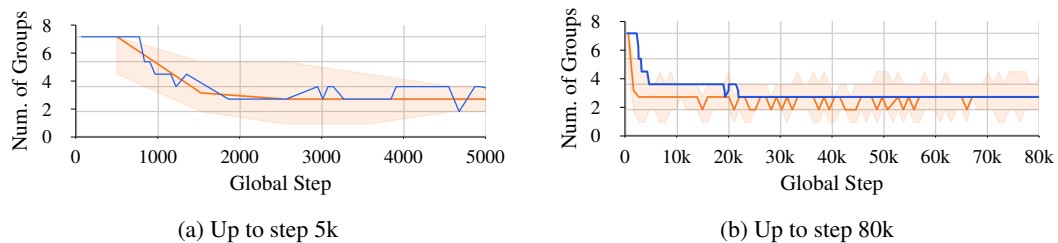

(a) Up to step 5k
(b) Up to step 80k

Figure 5: Grouping behavior throughout training. The blue line represents the number of active color groups at each training step. The orange line represents the median number of groups observed during each refresh period, with the shading showing the full range for that period. Subfigure (a) shows more details from step 0 to step 5,000 in the training process, and Subfigure (b) shows the data from step 0 to step 80,000. Both plots have been lightly smoothed based on moving medians to make them easier to interpret.

### S.1.1 Evolution of Task Group Sizes

Figure 3 shows that SON-GOKU quickly moves away from the trivial approach of training all tasks in one group ($\|G\| = 8$) and instead allocates most stakes to small- and medium-sized groups ($\|G\| \in \{3, 4, 5\}$), with only a small fraction ever isolated on their own. The smooth but continually shifting bands indicate that the scheduler keeps adjusting the granularity of groups over training. It continues to refine task partitions as patterns in task interference change, rather than committing to a fixed grouping of tasks.

### S.1.2 Conflict Sparsity in the Task Graph

Both plots in Figure 4 show that, after an initial phase of dense conflict, SON-GOKU's conflict graph rapidly becomes more and more sparse. Both the node degree and overall edge density drop significantly and then fluctuate around a low level. This indicates that the scheduler quickly resolves most cross-task conflicts and then maintains a partition of tasks into predominantly non-conflicting groups for the rest of training. This idea is supported by our graph visualizations in Appendix S.3.

### S.1.3 Group Count and Schedule Stability

Figure 5 shows that SON-GOKU starts with many groups but soon settles to consistently using three to four groups for most of training. The blue and orange lines across both plots stay close together, meaning the number of groups at each step stays relatively consistent (with slight oscillations, as we see in our graph visualizations in Appendix S.3) with what we see across refresh periods. So the grouping behavior is not fully stable, but converges within a range over time in which it is able to adapt to evolving task relationships. Tuning the SON-GOKU hyperparameters to achieve more stable

Table 11: Validation performance on Taskonomy for SON-GOKU using a U-Net backbone.

| Metric | SON-GOKU |
|---|---|
| **Depth (Euclidean)** | |
| AbsRel ↓ | $0.5432 \pm 0.0068$ |
| MAE ↓ | $4.0787 \pm 0.071$ |
| RMSE ↓ | $18.6199 \pm 0.29$ |
| **Depth (Z-buffer)** | |
| AbsRel ↓ | $0.5731 \pm 0.0074$ |
| MAE ↓ | $4.0494 \pm 0.069$ |
| RMSE ↓ | $18.5378 \pm 0.27$ |
| **Edge Occlusion** | |
| BCE ↓ | $0.0995 \pm 0.0032$ |
| F1 ↑ | $0.1466 \pm 0.010$ |
| Precision ↑ | $\mathbf{0.5399} \pm 0.021$ |
| Recall ↑ | $0.0870 \pm 0.0065$ |
| **Edge Texture** | |
| BCE ↓ | $\mathbf{0.0779} \pm 0.0010$ |
| F1 ↑ | $\mathbf{0.9734} \pm 0.0007$ |
| Precision ↑ | $\mathbf{0.9772} \pm 0.0008$ |
| Recall ↑ | $\mathbf{0.9768} \pm 0.0007$ |
| **Keypoints2D** | |
| MSE ↓ | $0.0039 \pm 0.0002$ |
| **Surface Normals** | |
| 11.25° within ↑ | $0.4262 \pm 0.0079$ |
| 22.5° within ↑ | $0.6432 \pm 0.0068$ |
| 30° within ↑ | $0.7340 \pm 0.0059$ |
| Mean angle (deg) ↓ | $22.9079 \pm 0.31$ |
| Median angle (deg) ↓ | $14.5397 \pm 0.27$ |
| **Principal Curvature** | |
| L1 ↓ | $0.0691 \pm 0.0014$ |
| **Reshading** | |
| MAE ↓ | $0.1352 \pm 0.0023$ |

grouping is possible, but in theory may lead to worse responsiveness to evolving task relationships (see Appendix O and Section 5).

## S.2 CONVERGENCE AND LOSS

To enhance the transparency of our work, we plot the convergence of the SON-GOKU based model across various tasks from the Taskonomy dataset. We plot each metric's changing value against the step number. Across most metrics, we can see a clear pattern of improvement early in training followed by a slower and more stable plateau. However, there are some other metrics where the graph oscillates throughout the training process, demonstrating the difficulty (or perhaps impossibility) of finding a global minima that satisfies all tasks and metrics. This is consistent with a clear pattern established by the literature around the Taskonomy dataset.

To further improve transparency and clarity, we plot the exact final performance values of SON-GOKU on Taskonomy in Table 11 (identical to data and results of cosine-based SON-GOKU presented in Table 8).

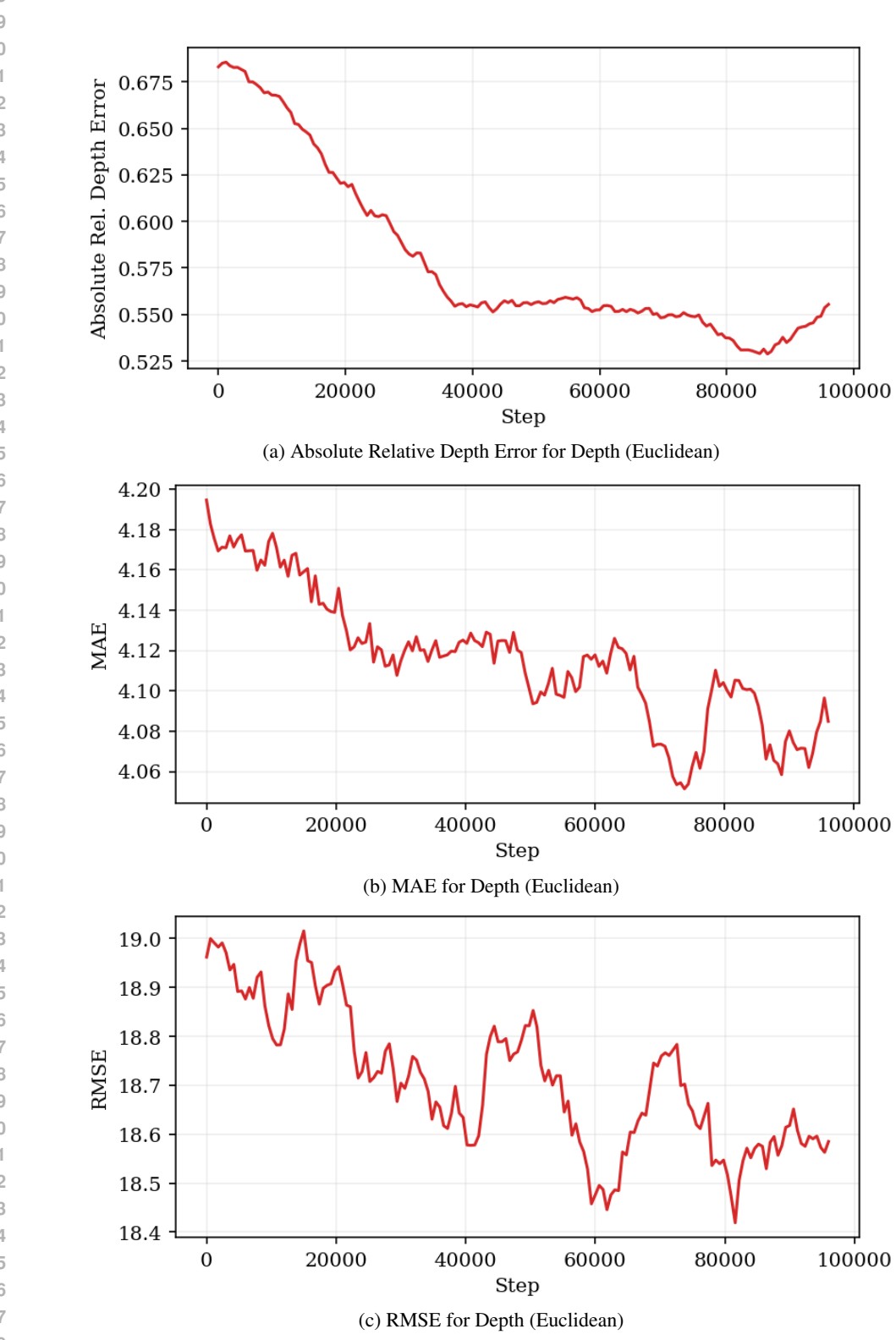

(a) Absolute Relative Depth Error for Depth (Euclidean)

(b) MAE for Depth (Euclidean)

(c) RMSE for Depth (Euclidean)

Figure 6: Convergence curves across all the used Taskonomy tasks and metrics.

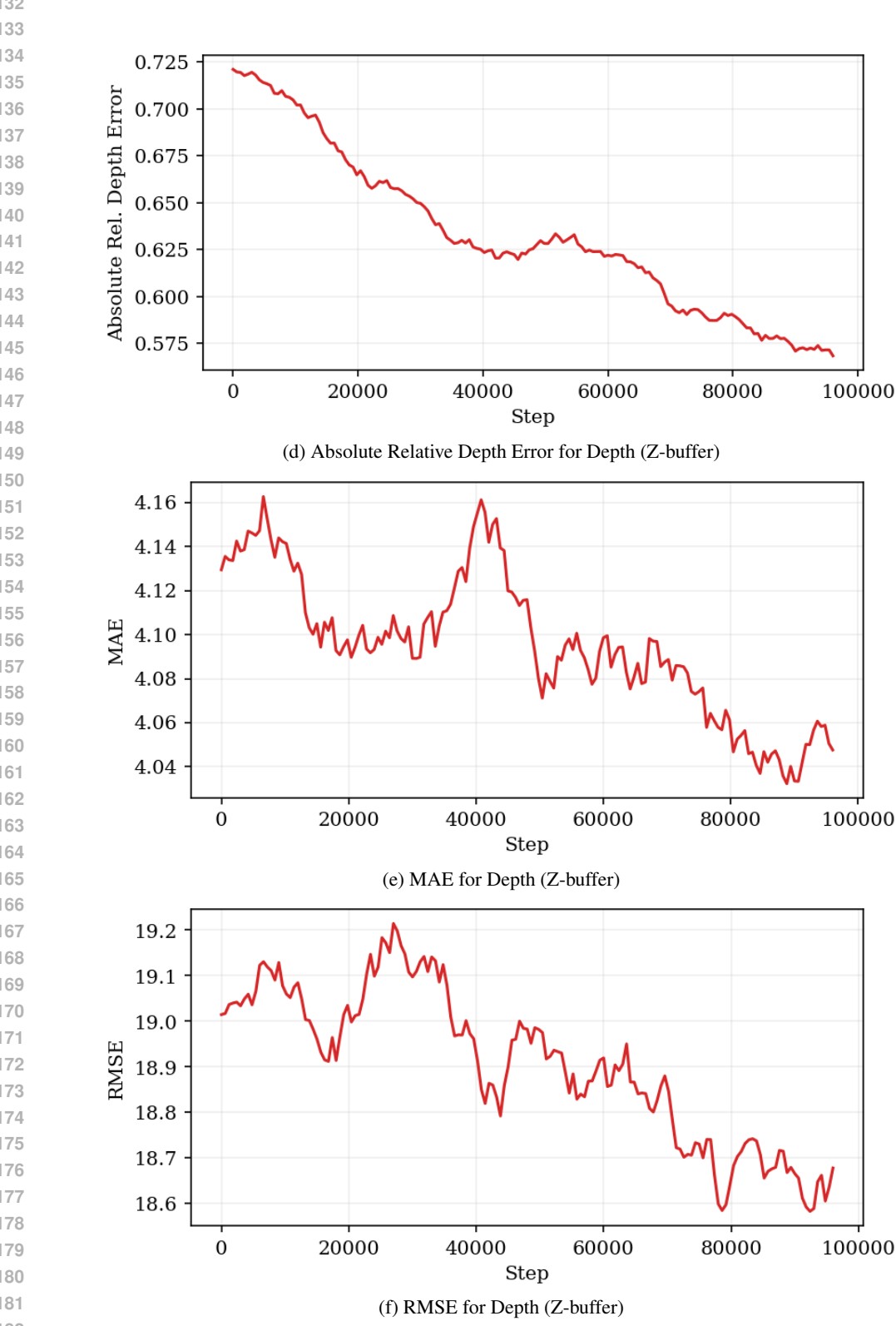

(d) Absolute Relative Depth Error for Depth (Z-buffer)

(e) MAE for Depth (Z-buffer)

(f) RMSE for Depth (Z-buffer)

Figure 6: Convergence curves across all the used Taskonomy tasks and metrics (continued).

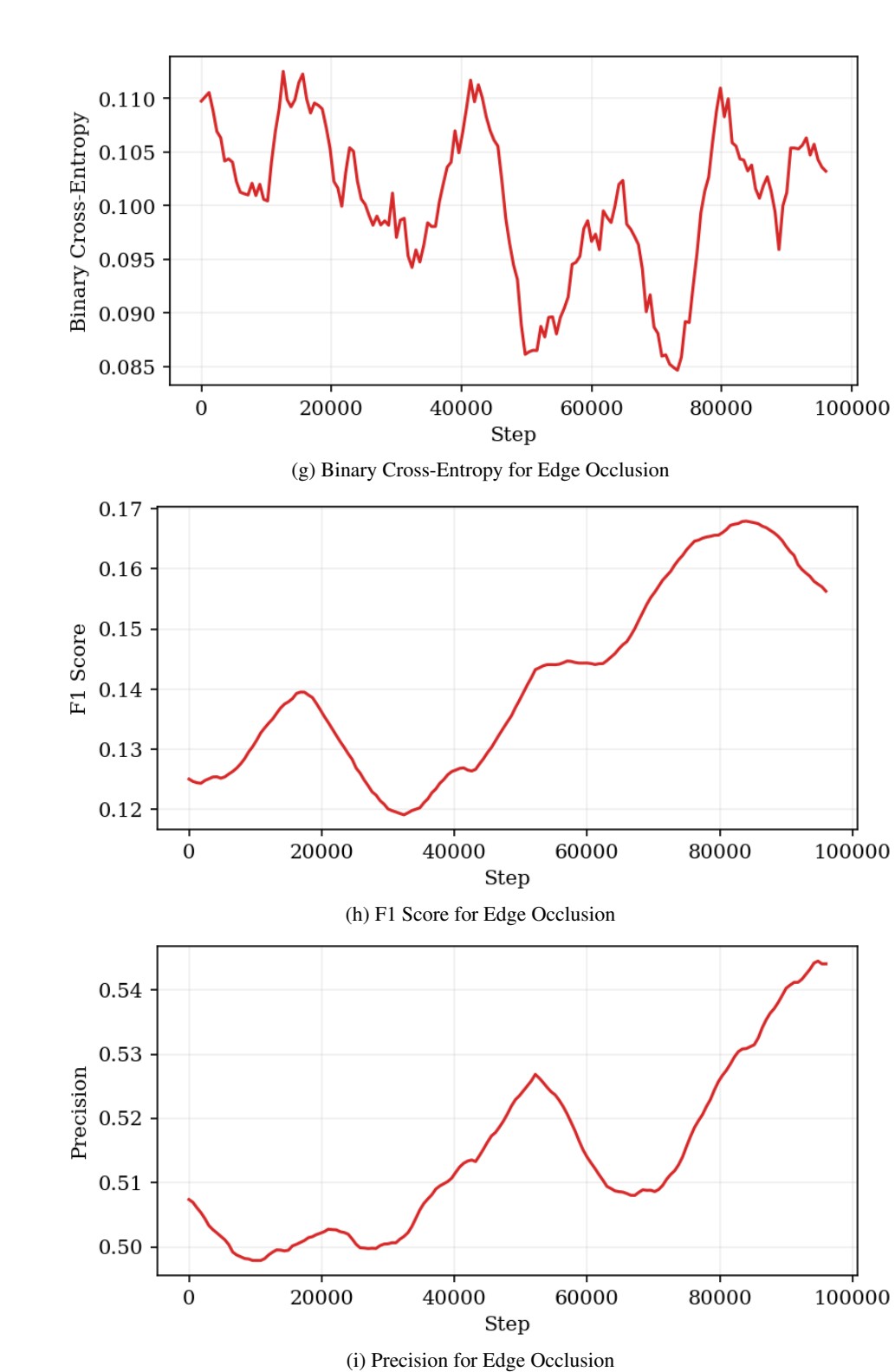

(g) Binary Cross-Entropy for Edge Occlusion

(h) F1 Score for Edge Occlusion

(i) Precision for Edge Occlusion

Figure 6: Convergence curves across all the used Taskonomy tasks and metrics (continued).

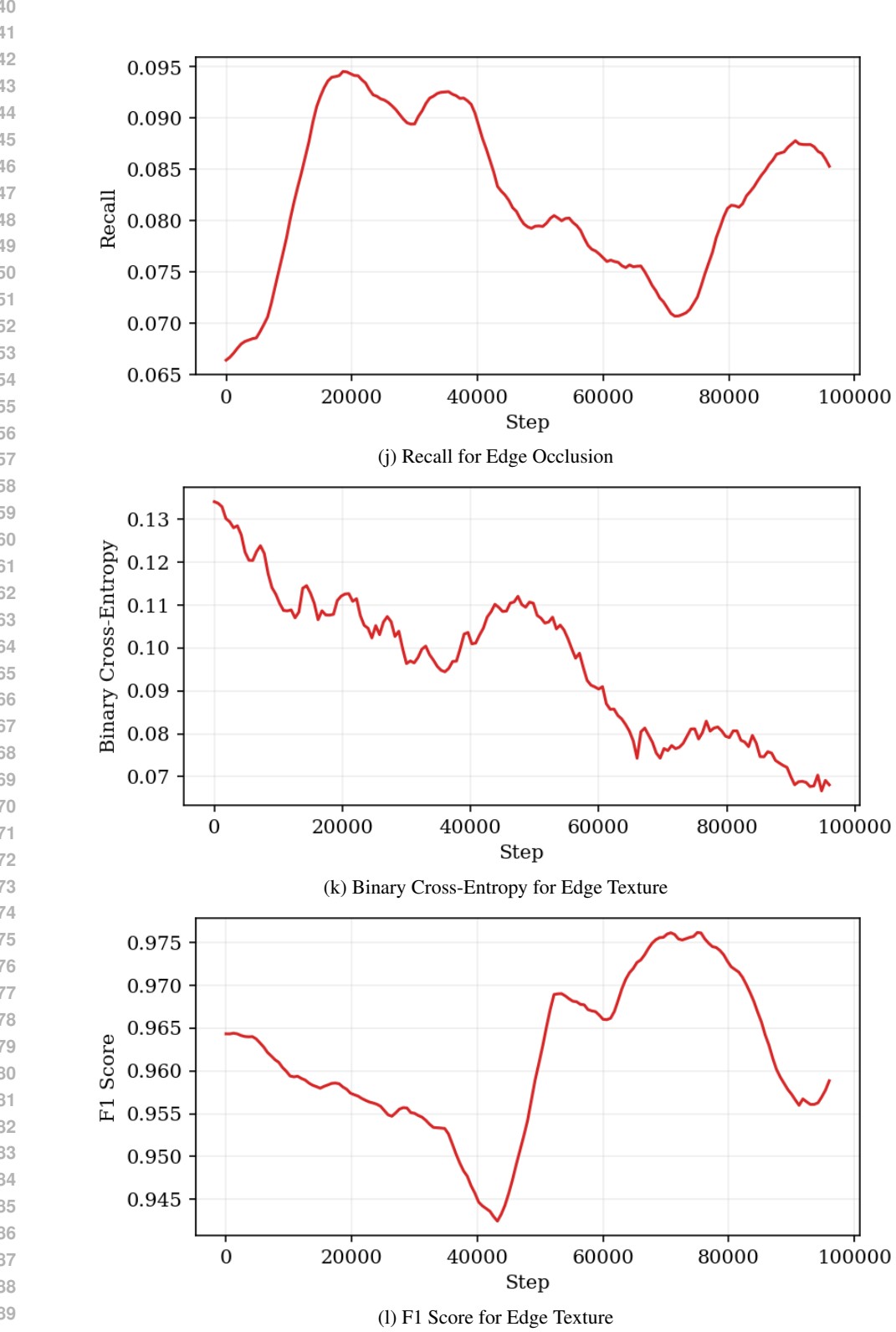

(j) Recall for Edge Occlusion

(k) Binary Cross-Entropy for Edge Texture

(l) F1 Score for Edge Texture

Figure 6: Convergence curves across all the used Taskonomy tasks and metrics (continued).

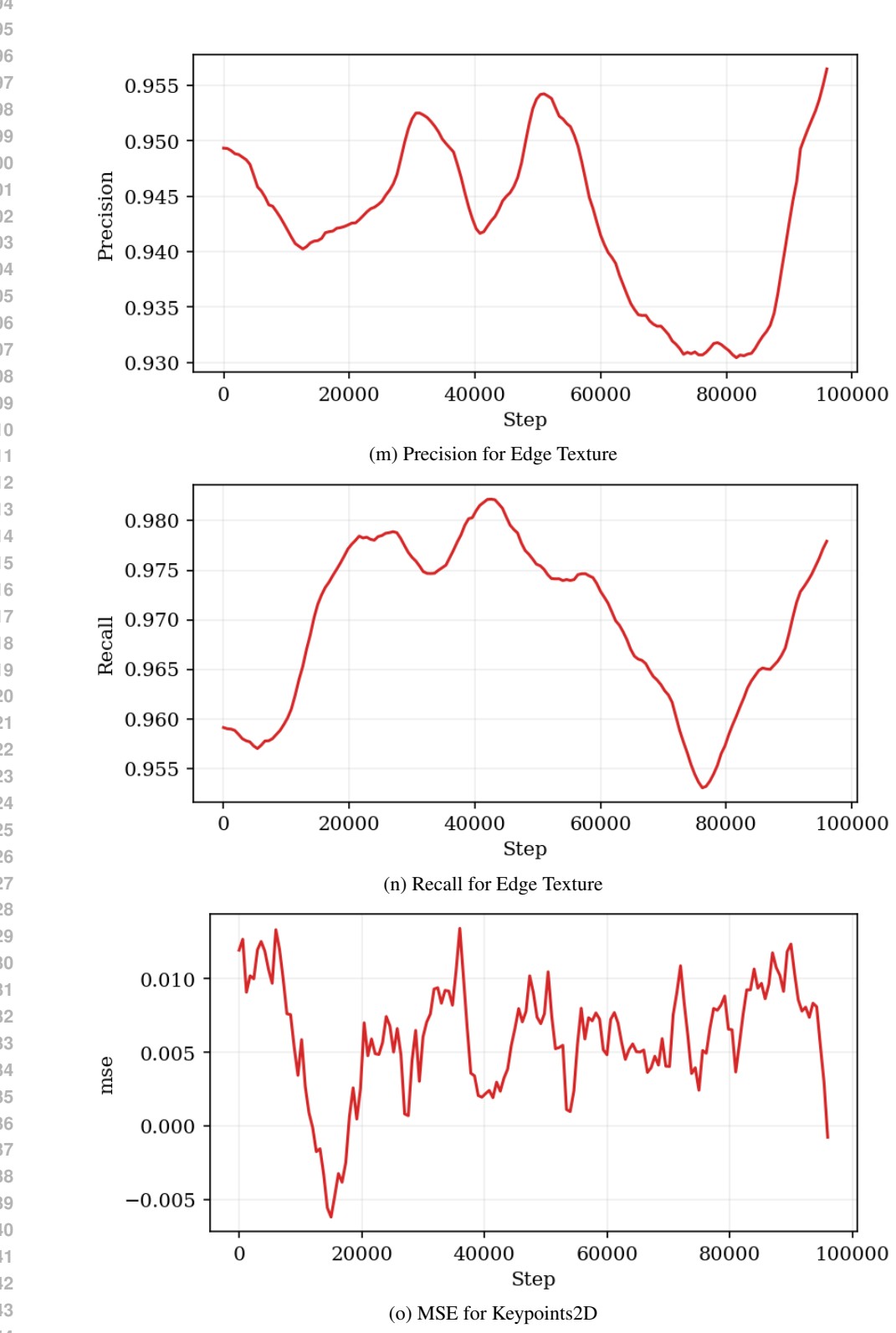

(m) Precision for Edge Texture

(n) Recall for Edge Texture

(o) MSE for Keypoints2D

Figure 6: Convergence curves across all the used Taskonomy tasks and metrics (continued).

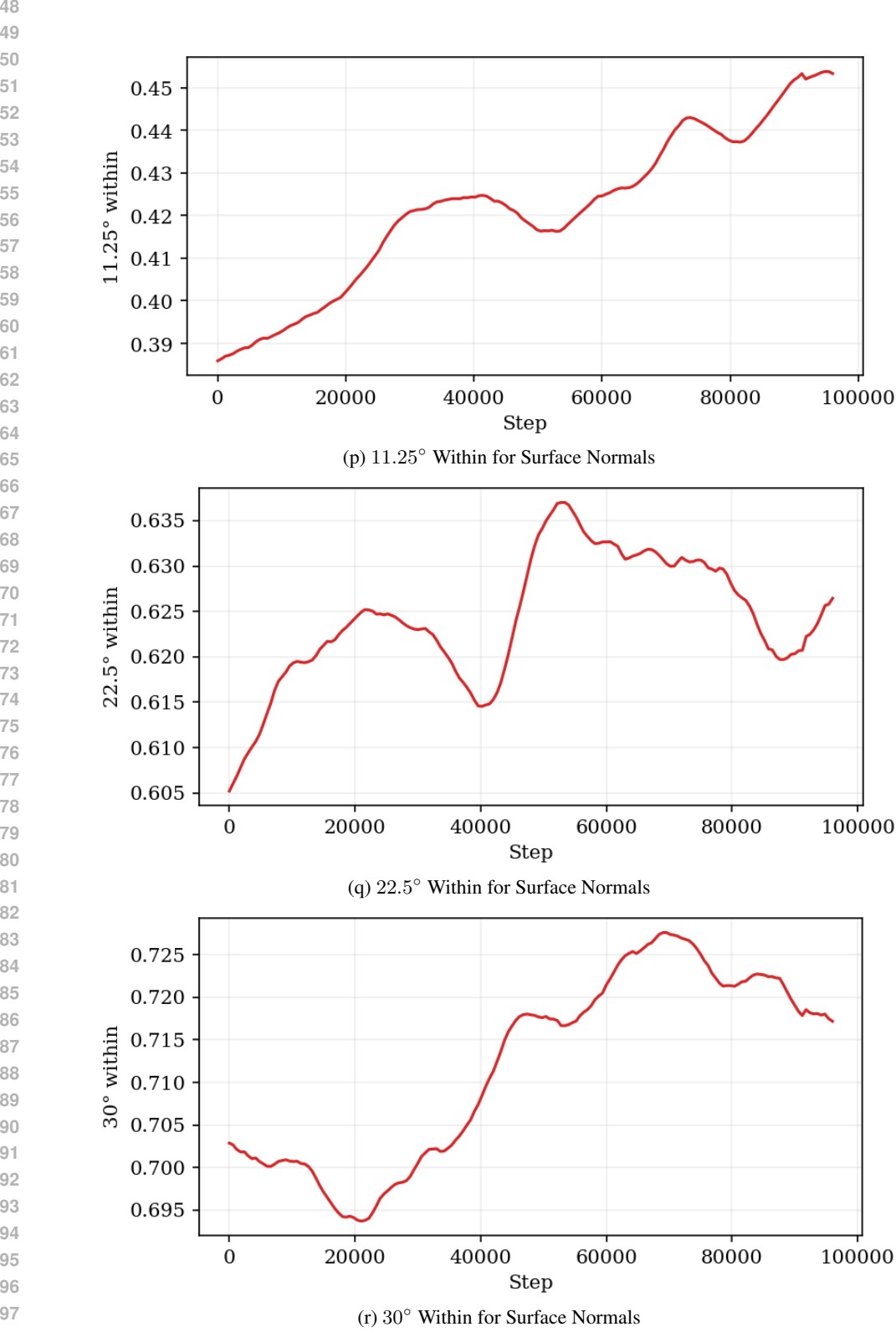

(p) 11.25° Within for Surface Normals

(q) 22.5° Within for Surface Normals

(r) 30° Within for Surface Normals

Figure 6: Convergence curves across all the used Taskonomy tasks and metrics (continued).

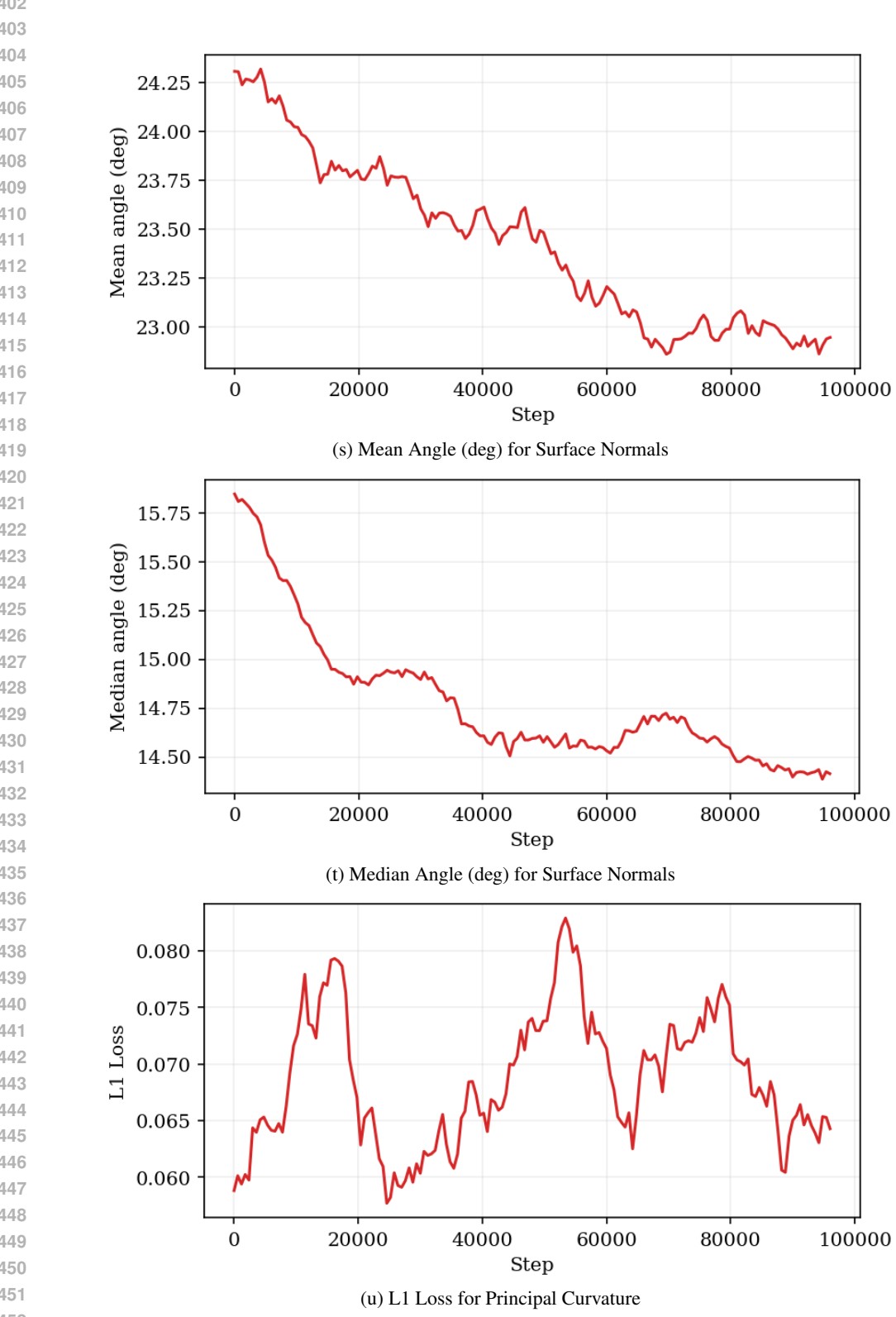

(s) Mean Angle (deg) for Surface Normals

(t) Median Angle (deg) for Surface Normals

(u) L1 Loss for Principal Curvature

Figure 6: Convergence curves across all the used Taskonomy tasks and metrics (continued).

## S.3 GRAPH EVOLUTION

SON-GOKU captures relationships between tasks and groups them accordingly throughout training. It does so online, meaning that it is able to adapt to the constantly changing relationships between tasks. In this instance, SON-GOKU is able to capture a clear consistent relationship between separate tasks, but it still constantly regroups tasks throughout the training process to adapt at different points (Figure 7). Overall, this constantly changing grouping, despite SON-GOKU having identified a somewhat consistent underlying task structure, indicates that SON-GOKU is highly adaptive to evolving cross-task relationships. For more consistent grouping that still optimizes performance, one would need to adjust the scheduler hyperparameters that control the refresh period and EMA history length. Essentially, one would trade off stability against responsiveness.

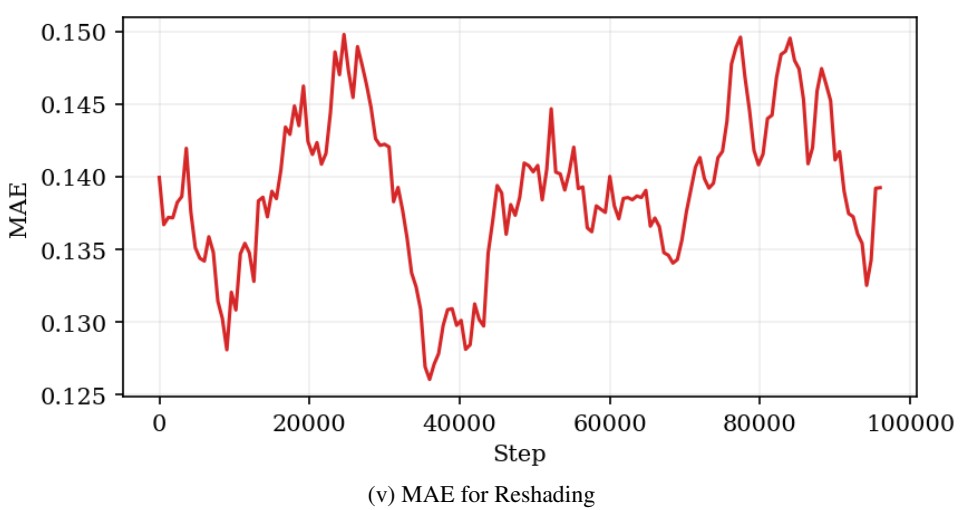

(v) MAE for Reshading

Figure 6: Convergence curves across all the used Taskonomy tasks and metrics (continued).

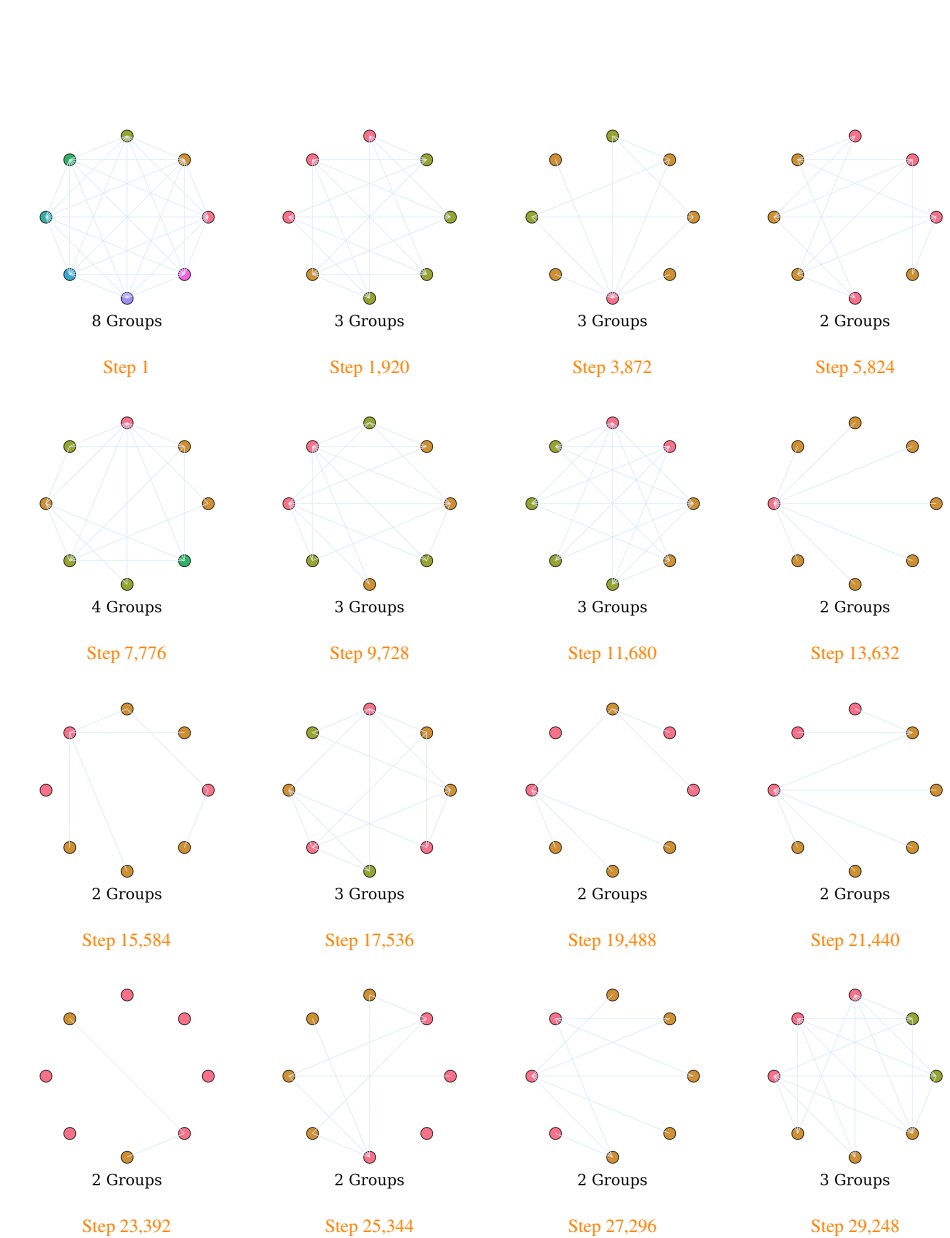

Figure 7: Task-graph evolution during training. Starting with the top node and rotating clockwise, the tasks represented are: Edge Occlusion, Depth Z-Buffer, Depth Euclidean, Reshading, Principal Curvature, Normal, Keypoints 2D, and Edge Texture.

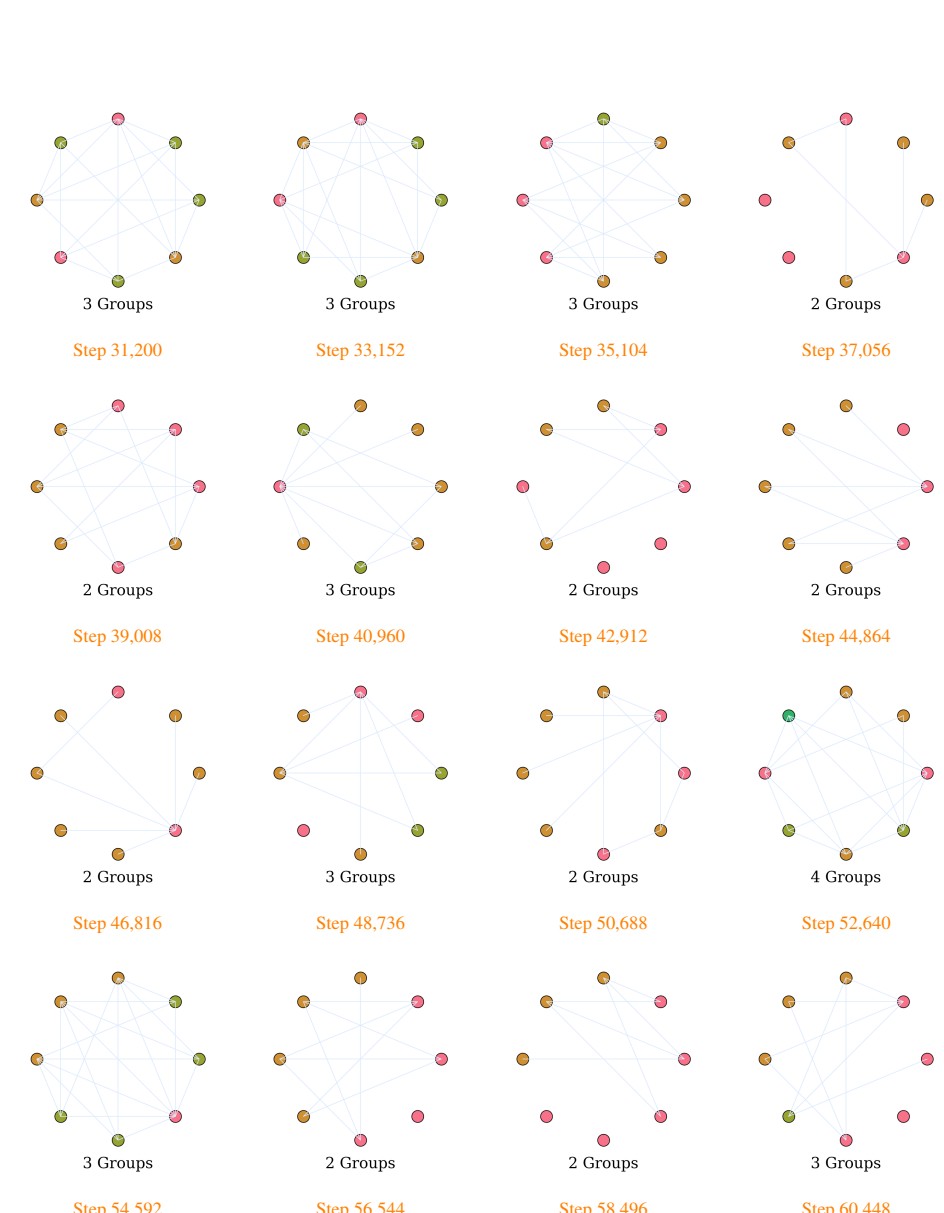

Figure 7: Task-graph evolution during training (continued). Starting with the top node and rotating clockwise, the tasks represented are: Edge Occlusion, Depth Z-Buffer, Depth Euclidean, Reshading, Principal Curvature, Normal, Keypoints 2D, and Edge Texture.

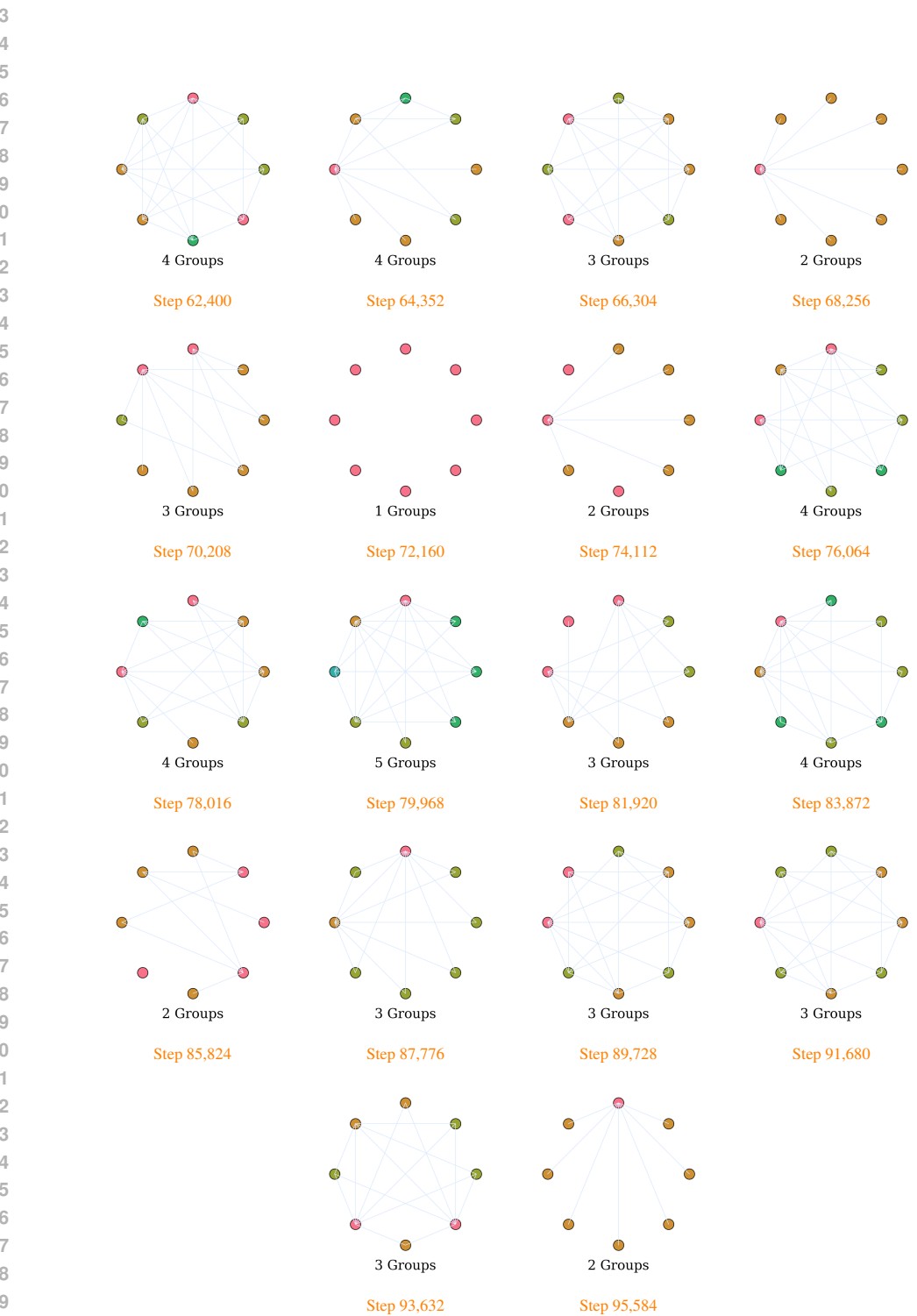

Figure 7: Task-graph evolution during training (continued). Starting with the top node and rotating clockwise, the tasks represented are: Edge Occlusion, Depth Z-Buffer, Depth Euclidean, Reshading, Principal Curvature, Normal, Keypoints 2D, and Edge Texture.