# OpenReview forum: "Graph Coloring for Multi-Task Learning"
_ICLR.cc/2026/Conference — Submitted to ICLR 2026_

### Official Review · Reviewer_bNCj · 2025-10-30

**Soundness:** 3
**Presentation:** 4
**Contribution:** 3
**Rating:** 6
**Confidence:** 4

**Summary:**

This paper presents an interference-aware scheduler named SON-GOKU, which is designed to improve the efficiency and effectiveness of multi-task learning by dynamically grouping tasks based on their gradient conflicts. The proposed method addresses the issue of negative transfer, where conflicting tasks slow down or degrade overall performance. The core idea revolves around constructing a conflict graph from task gradient interference and using greedy graph coloring to partition tasks into non-conflicting groups. SON-GOKU periodically recomputes these groups and activates only one group per training step, aiming to ensure that each mini-batch contains tasks pulling the model in compatible directions. Experiments demonstrate the effectiveness and high scalability of this method.

**Strengths:**

1. The paper is well-motivated and quite novel in solving gradient conflicts. The introduction explains how greedily coloring the graph addresses the existing three issues of grouping methods.
2. The authors provide rigorous theoretical guarantees on grouping, convergence, and conflict graph recovery.
3. The experiments demonstrate that SON-GOKU outperforms baselines (e.g., PCGrad, AdaTask, FAMO) and complements existing optimizers like PCGrad and AdaTask.
4. Using graph relationships to represent gradient conflicts between tasks is quite interesting and provides a new direction for solving multi-task problems.

**Weaknesses:**

1. This paper uses the interference coefficient to measure the relationships between tasks and then constructs a graph. However, the relationships between tasks can also be represented by a causal matrix, which can also be constructed into a graph. Therefore, what are the advantages or differences between the graph construction method in this paper and the causal graph construction?
2. Appendix D.5 discusses multiple approaches for reducing time complexity—random projections, deterministic covariance sketching, edge sampling, incremental Gram updates, and warm-start initializations. While each method is described, its individual impact on the final performance is not clearly distinguished. There could be more experimental results to support the theoretical analysis.
3. This paper lacks some parameter experiments, such as how the tau parameter changes throughout the training process, and how the task graph evolves in specific experiments
4. Although this article provides a detailed description of the method steps, making it highly reproducible, the author does not appear to have provided the code.

**Questions:**

Please see weakness.

---

> ### Author Response · Authors · 2025-11-28
> **Response to Reviewer bNCj (1/7)**
>
> We thank Reviewer bNCj for the thoughtful and detailed review of our submission and for highlighting both the strengths and the remaining concerns. We are glad that you found **the problem well-motivated**, the use of **graph coloring for resolving gradient conflicts to be a novel and interesting direction**, and the **thorough theoretical guarantees** and **empirical results convincing**.
>
> Below, we address your specific points in turn: **(W1)** how our interference based conflict graph differs from causal task graphs and why we don't use causal graphs or matrices; **(W2)** the empirical support for the time-complexity reductions discussed in *Appendix D.5*; **(W3)** the lack of explicit parameter and graph evolution experiments; and **(W4)** the availability of code (repository now available at https://anonymous.4open.science/r/SON-GOKU-ICLR-95AB).

---

> ### Author Response · Authors · 2025-11-28
> **Response to Reviewer bNCj, Weakness 1 (2/7)**
>
> ## **W1: Representing Relationships Between Tasks as a Causal Matrix**
> Our conflict graph is defined **purely in terms** of the **current training dynamics** of a shared model, **not** in terms of structural causal relationships between tasks. We define the **interference coefficient** between tasks $i$ and $j$ as $\rho_{ij} = - \frac{\langle \tilde g_i, \tilde g_j \rangle}{\|\tilde g_i\|\,\|\tilde g_j\|}$, where $\tilde g_i, \tilde g_j$ are **smoothed gradients** with respect to the shared parameters. Positive $\rho_{ij}$ corresponds to **conflicting gradients** (negative cosine) and nonpositive $\rho_{ij}$ corresponds to **alignment or neutrality**.
>
> We then form a **conflict graph** $G_\tau$ by adding an (undirected) edge $(i, j)$ whenever $\rho_{ij} > \tau$. Such pairs of tasks should not be updated together.
>
> By contrast, a **causal graph** (or a "causal matrix") typically aims to encode **structural** and **directional relationships** such as if one task causally influences another (e.g., through label dependencies). Its edges are usually **directed** and *do not* directly refer to the **current gradient geometry** of a specific shared model.
>
> In short, our conflict graph is **symmetric** and **undirected**, encodes **instantaneous gradient interactions** in parameter space, and is directly **tied to the update** $\theta_{t+1} = \theta_t - \eta_t \sum_{k \in S_t} g_{k,t}$. On the other hand, a causal graph is typically **directed** (and possibly weighted), encodes structural **cause-and-effect relationships** between variables and tasks, and is **not directly defined in terms of the gradient vectors** of a particular model.
>
> Our goal in this work is to **improve optimization** by controlling **destructive interference among gradient**s, not to **recover** or use a **structural causal model** of the tasks.
>
> ### **Why cosine-based interference is the right quantity for our scheduler**
>
> The interference coefficient is the **exact quantity** that appears in our **descent** and **convergence analysis**. If the active set $S_t$ at step $t$ is "$\tau$-compatible" (i.e., for all $i \neq j \in S_t$, $\rho_{ij} \le \tau$), we prove the following lower bound on the squared norm of the combined update:
> $$
> \Bigl\|\sum_{k \in S_t} g_{k,t}\Bigr\|^2
> \;\ge\;
> \bigl(1 - \tau\bigl(|S_t| - 1\bigr)\bigr)
> \sum_{k \in S_t} \|g_{k,t}\|^2.
> $$
>
> This guarantees that as long as $\tau(|S_t|-1) < 1$, the **joint update direction cannot flip** to ascent, and the **amount of cancellation is quantitatively controlled by $\tau$** and the group size.
>
> This is crucial. The only information we need to obtain this guarantee are the **pairwise cosines between gradients** (i.e., exactly what $\\rho_{ij}$ measures). Our construction of the conflict graph is therefore tailored so that:
> * Each color class in the graph has **bounded pairwise interference**.
> * The aggregated step over that class is a **provable descent direction** with a clean bound.
>
> We then show that under standard smoothness and noise assumptions, SON-GOKU preserves the usual **nonconvex SGD convergence rate** up to a small factor depending on $\tau$. Again, this argument is expressed in terms of **inner products and cosines of gradients**, not in terms of causal effects.
>
> In contrast, a generic causal matrix typically **does not have a direct interpretation** in the inequality above. Its entries **need not bound gradient inner products**, and so thresholding such a matrix **would not yield the compatibility guarantees** (and SGD convergence bounds) we prove.

---

> ### Author Response · Authors · 2025-11-28
> **Response to Reviewer bNCj, Weakness 1 (3/7)**
>
> ### **Practical and statistical differences from a causal matrix**
> There are important practical differences between our **interference based graph** and a **causal graph over tasks**. In terms of data and estimation burden, the interference coefficient is deliberately defined so that it can be estimated at **practically no cost** from **quantities we already compute during training**. We only need the **task gradients** that arise in **standard backpropagation**, plus short exponential moving averages and periodic refreshes to keep the cosine estimates stable. The only extra cost is **computing the pairwise cosines at refresh**, which leads to an amortized overhead of $\Theta(K^2 d / R)$ per step for $K$ tasks, shared parameter dimension $d$, and refresh period $R$. By contrast, a causal matrix between tasks is usually defined in terms of **how intervening on task $i$ affects task $j$**. In practice, that would require **interventional** or **tightly controlled observational data**. For example, it could require **repeatedly training** with and without task $i$, and then measuring how the performance of task $j$ changes. Doing this robustly for **all pairs of tasks** is typically going to be far more **expensive** and often **infeasible** in an online training loop (i.e., calculating it on the fly during training instead of precomputing), because any change in model architecture, optimizer, or data distribution would in principle require re-estimating the causal matrix. Our approach, in contrast, gives **online**, **training time** feedback about task interactions at essentially no extra data cost.
>
> There is also a difference in **temporal resolution** and **adaptivity**. Because the interference graph is recomputed every $R$ steps from recent EMA smoothed gradients, it can track how **relationships between tasks evolve** as the shared **representation changes during training**. When task gradients drift from aligned to conflicting (or vice versa), the conflict graph and its coloring **adjust accordingly**. A causal matrix, on the other hand, is usually treated as a **fixed property of the underlying problem**, so the causal relations between labels or tasks **do not change** just because the network is in a different training phase. To make a causal matrix respond to optimization dynamics, one would need to run a potentially **heavy causal discovery procedure** over and over again. **Our ablation** between Static One-Shot coloring and the full SON-GOKU scheduler shows that **this kind of adaptivity is highly important**. Using a single, frozen grouping leads to **stale partitions** and **worse performanc**e, whereas dynamic recoloring based on recent gradient statistics is **empirically important for good results**.
>
> Finally, our grouping problem is fundamentally **symmetric**, whereas causal graphs are typically **directional**. For scheduling, what matters is whether **updating tasks $i$ and $j$ together** causes their gradients to **cancel** or **clash**. This is naturally captured by a **symmetric cosine based quantity** $\rho_{ij} = \rho_{ji}$, which tells us whether the two tasks are **safe to update** in the **same step**. Causal matrices, in contrast, are usually **asymmetric**. They distinguish "$A \rightarrow B$" from "$B \rightarrow A$" and may encode **long range** or **mediated effects** that are important for **reasoning about interventions**, but **not directly relevant** to **whether two tasks can share a particular gradient step** safely. Even if we had access to a perfect causal graph over tasks, the **optimization decision** of **which tasks to update together at a given step** does not follow from that structure in any straightforward way, whereas the **interference coefficient** is designed precisely for this decision.

---

> ### Author Response · Authors · 2025-11-28
> **Response to Reviewer bNCj, Weakness 1 (4/7)**
>
> ### **How a causal graph could relate to our framework**
> We agree that causal relations between tasks or labels are **interesting** and **might be useful** in addition to the gradient based conflict we model. Our framework is actually **quite modular**, and can be split into the steps of **(1)** estimating some pairwise interaction score between tasks, **(2)** threshold this score to obtain a sparse graph, and **(3)** coloring the graph greedily and schedule color classes sequentially.
>
> In principle, one could **plug in a score** derived from a **causal matrix** in step (1), as long as the **resulting graph is interpreted as edges representing pairs that should not be updated together**. However, our current theoretical guarantees (descent preservation, convergence rate, and exact recovery of the population conflict graph) are specifically proved for interaction scores that are derived from **gradient cosines**. Additionally, a **causal adjacency matrix** does not directly control **gradient inner products**, so the inequalities and bounds we rely on would **no longer hold** without **substantial new assumptions** and analysis.
>
> We see exploring such causal variants as an interesting future direction, but it is orthogonal to the contribution of the present paper, which is a gradient interference based scheduler with provable optimization properties.
>
> ### **Summary to reviewer bNCj**
> To directly address your question, our graph is not intended as a causal graph. It is an interference graph whose edges are derived from empirically measured cosine similarities of task gradients and whose primary purpose is to **control destructive interference in shared parameter updates**. This construction has two main advantages for our setting. Firstly, it is **tightly coupled** to **optimization theory**. The interference coefficients are **exactly the quantities** appearing in our **descent and convergence bounds**, so thresholding them yields groups with **provably safe** and **often superior** descent behavior. It is also **practical and adaptive**, as it can be estimated **online** with **minimal overhead** from gradients already used for learning, and it naturally tracks **evolving task relations** during training.
>
> We have **clarified this distinction** in the revised version by **explicitly contrasting our conflict graph with causal graphs** in *Section 3.1.1* & *3.1.2* (Interference Coefficient & Conflict Graph) and by emphasizing that SON-GOKU is designed for **controlling interference at optimization-time** rather than **structural causal modeling**. Changes are highlighted in the color orange.

---

> ### Author Response · Authors · 2025-11-28
> **Response to Reviewer bNCj, Weakness 2 (5/7)**
>
> ## **W2: Adding Experimental Results to Support Theoretical Analysis in Appendix D.5**
> We appreciate Reviewer bNCj's careful reading of *Appendix D.5* and for highlighting the gap between the theoretical complexity discussion and the empirical results. Our intention in Appendix D.5 is to **present a list of optional acceleration strategies** (random projections, deterministic covariance sketching, edge sampling, incremental Gram updates, and warm-start initializations) that can be **layered on top of** the core SON-GOKU scheduler without changing its semantics (i.e., the underlying notion of task interference and graph coloring) but **with different tradeoffs between accuracy and efficiency**.
>
> In our current draft, all our main results (*Table 1* and *Table 2*) are reported with the full implementation of SON-GOKU, where the conflict matrix is computed exactly at each refresh. *Appendix D* and *Section 7.4* focus on overall amortized complexity and wall-clock behavior as a function of $K$ and $R$, contrasting SON-GOKU with existing baselines (e.g., PCGrad, FairGrad, Nash-MTL, FAMO, AdaTask), rather than isolating each approximation in *D.5*.
>
> We agree that **empirically validating** the effect of **each acceleration technique** (both on speed and final performance) would strengthen and better support the theoretical complexity analysis. Therefore, we are now running **additional ablation experiments** that enable one acceleration technique at a time on top of the default scheduler (and combinations thereof). These experiments are currently underway. Our goal is to **quantify how much each technique reduces the $O(K^2 d / R)$ cost** in practice and how much, if at all, it **degrades conflict estimation** and **downstream performance**.

---

> ### Author Response · Authors · 2025-11-28
> **Response to Reviewer bNCj, Weakness 3 (6/7)**
>
> ## **W3: Lacking Parameter Experiments**
> We agree that explicitly showing how **the hyperparameters** and the **task graph** evolve over training would make the scheduler much easier to interpret, and we appreciate you pointing this out.
>
> In the current draft, **we already describe how $\tau$ and the conflict graph behave**, but mostly in **text** and **theory** rather than with explicit visualizations. Specifically:
> * *Algorithm 1* and *Section 4.4* specify that at each refresh we **(i)** recompute the interference matrix from EMA gradients, **(ii)** rebuild the conflict graph by thresholding negative cosine similarities at $-\tau$ (written as $\rho_{ij} > \tau $ or, equivalently, $ \cos(\theta_{ij}) < -\tau $), and **(iii)** recolor this graph to obtain a new schedule.
> * *Section 4.4.2* defines the deterministic **warmup and annealing schedule**. We start from $\tau = 1$ (no edges, all tasks co-updated) and **logarithmically anneal** to **target $\tau$** after an initial warmup phase, so $\tau$ **does not fluctuate stochastically** but **follows a simple schedule** shared across runs.
> * *Section 5.1* and *Appendix E* explain how $\tau$ controls the tradeoff between **group size** and **interference**. If the active set $S_t$ is $\tau$-compatible, the joint update retains at least a $(1 - \tau (|S| - 1))$ fraction of the summed squared step lengths, so smaller $\tau$ enforces **more conservative** but **lower conflict** groups.
> * *Section 5.5* shows how the **evolving conflict graph**, via the Welsh-–Powell coloring, **induces schedules** with at most $\Delta+1$ groups, where $\Delta$ is the maximum conflict degree, giving **guaranteed bounds of staleness** in terms of **how often each task is updated**.
>
> We agree, however, that these dynamics are currently only *implicit* and that the paper would benefit from specialized parameter and graph evolution experiments. In the revised version, we will therefore add an Appendix section for at least one of our multi-task benchmarks (most likely the newly added Taskonomy dataset, due to the sheer amount of data and tasks it provides) that:
> * **Tracks graph statistics over refresh rounds**, including edge density, maximum degree $\Delta$, number of colors $m$, and the distribution of group sizes, to show how the conflict graph and its coloring evolve as the shared representation improves.
> * **Visualizes the task graph itself at a few representative epochs** using diagrams to show how the different tasks and groups evolved throughout training. This will highlight how strongly conflicting tasks are **gradually separated** into **different color classes** while benign or synergistic tasks **remain grouped**.
>
> Similar visualizations have been very effective in **related work** on multi-task grouping and dataset behavior (e.g., Standley et al. for task affinity patterns; Swayamdipta et al. for dataset cartography), and we believe they will make SON-GOKU's behavior much more transparent.
>
> To summarize, $\tau$ itself follows a **simple predefined annealing schedule** (so its trajectory does not need to be displayed), while the conflict graph and its coloring evolve in a **data-dependent way** that **we already analyze theoretically** but do not yet visualize. We will add plots and graph snapshots in our revised paper to directly address your request for parameter experiments and to give an intuitive picture of how the task graph change over time in actual experiments. We are currently in the process of training and testing on the Taskonomy dataset and will provide the new visualizations and analyses once training has completed.

---

> ### Author Response · Authors · 2025-11-28
> **Response to Reviewer bNCj, Weakness 4 (7/7)**
>
> ## **W4: Providing Code Implementation for SON-GOKU**
> Thank you for pointing this out. We completely agree that releasing code is important for making the work more reproducible and more usable in practice, beyond the **detailed algorithmic descriptions** and **appendices** already in the paper. Therefore, we have created an **anonymized code repository** (to comply with double-blind peer review) available at this URL: https://anonymous.4open.science/r/SON-GOKU-ICLR-95AB
>
> The repository is set up in such a way that developers can download a Python package to interface with SON-GOKU. It also has code to download, load, and preprocess all the datasets that we used in our paper. We have designed it to be easy-to-use while still providing flexibility to tune hyperparameters and configure the framework to suit developers' needs. It allows for experimentation with SON-GOKU itself as well as its ablations (including the new Graph Construction ablations from our discussion with Reviewer hC6i). The README.md file contains instructions for downloading the package and running experiments. If the paper is accepted, we will provide a de-anonymized GitHub repository link.

---

> > ### Author Response · Authors · 2025-11-28
> > **Response to Reviewer bNCj, References**
> >
> > # **References**
> >
> >
> >
> > Ban, H., & Ji, K. (2024). Fair Resource Allocation in Multi-Task Learning. *ArXiv (Cornell University)*. https://doi.org/10.48550/arxiv.2402.15638
> >
> > Liu, B., Feng, Y., Stone, P., & Liu, Q. (2023). FAMO: Fast Adaptive Multitask Optimization. *ArXiv (Cornell University)*. https://doi.org/10.48550/arxiv.2306.03792
> >
> > Navon, A., Shamsian, A., Achituve, I., Maron, H., Kawaguchi, K., Chechik, G., & Fetaya, E. (2022). Multi-Task Learning as a Bargaining Game. *ArXiv (Cornell University)*. https://doi.org/10.48550/arxiv.2202.01017
> >
> > Standley, T., Zamir, A. R., Chen, D., Leonidas Guibas, Malik, J., & Savarese, S. (2020). Which Tasks Should Be Learned Together in Multi-task Learning? *International Conference on Machine Learning*. https://doi.org/10.48550/arxiv.1905.07553
> >
> > Swabha Swayamdipta, Schwartz, R., Lourie, N., Wang, Y., Hannaneh Hajishirzi, Smith, N. A., & Choi, Y. (2020). Dataset Cartography: Mapping and Diagnosing Datasets with Training Dynamics. *ArXiv (Cornell University)*. https://doi.org/10.18653/v1/2020.emnlp-main.746
> >
> > Welsh, D. J. A. (1967). An upper bound for the chromatic number of a graph and its application to timetabling problems. *The Computer Journal*, *10*(1), 85–86. https://doi.org/10.1093/comjnl/10.1.85
> >
> > Yang, E., Pan, J., Wang, X., Huang, Y., Shen, L., Chen, X., Liu, X., Jiang, J., & Guo, G. (2023). AdaTask: A Task-Aware Adaptive Learning Rate Approach to Multi-Task Learning. *Proceedings of the ... AAAI Conference on Artificial Intelligence*, *37*(9), 10745–10753. https://doi.org/10.1609/aaai.v37i9.26275
> >
> > Yu, T., Kumar, S., Gupta, A., Levine, S., Hausman, K., & Finn, C. (2020). Gradient Surgery for Multi-Task Learning. *ArXiv (Cornell University)*. https://doi.org/10.48550/arxiv.2001.06782
> >
> > Zamir, A. R., Sax, A., Shen, W., Guibas, L., Malik, J., & Savarese, S. (2018). Taskonomy: Disentangling Task Transfer Learning. *2018 IEEE/CVF Conference on Computer Vision and Pattern Recognition*. https://doi.org/10.1109/cvpr.2018.00391

---

> ### Author Response · Authors · 2025-12-04
> **Update on Experiments for Reviewer bNCj (1/3)**
>
> We would like to again thank Reviewer bNCj for their very thoughtful review of our paper. In the time since our original response, we have finished running **several additional experiments** and provided **additional theoretical analysis**, reasoning, and **commentary** in our paper to further address your concerns. When reading the revised paper, it may be helpful to note that text in the **color orange** was **written specifically to address your concerns**, though text in other colors may also be helpful to you. Also please keep in mind that we have already made numerous revisions to the paper even before experiments concluded, so not all changes made to address your concerns will be covered in this "Update on Experiments", those other changes are discussed in our previous responses.

---

> ### Author Response · Authors · 2025-12-04
> **Update on Experiments for Reviewer bNCj, Weakness 2 (2/3)**
>
> ## **Weakness 2**
> > "Appendix D.5 discusses multiple approaches for reducing time complexity—random projections, deterministic covariance sketching, edge sampling, incremental Gram updates, and warm-start initializations. While each method is described, its individual impact on the final performance is not clearly distinguished. There could be more experimental results to support the theoretical analysis."
>
> To address this concern, we have provided extensive experimental validation in Appendix D.5.5 under an additive ablation study. Our new experiments focus on the tradeoffs between speed and performance for each of the proposed approaches to reducing time complexity. Each approach is isolated under a fixed setting. Results are presented in the table below.
>
> | Family            | Variant                      | Elapsed (s) $\downarrow$   | Imgs/s $\uparrow$        | Refresh (ms) $\downarrow$   | Depth Eucl. RMSE $\downarrow$      | Depth Eucl. MAE $\downarrow$       | Normal mean (deg) $\downarrow$      | Reshading MAE $\downarrow$        |
> |-------------------|------------------------------|-----------------------------|---------------------------|------------------------------|-------------------------------------|-------------------------------------|-------------------------------------|------------------------------------|
> | Baseline          | FD (conservative)            | $76.37 \pm 1.84$           | $50.28 \pm 1.21$         | $488.94 \pm 27.35$           | $\mathbf{25.58} \pm 0.92$           | $\mathbf{7.98} \pm 0.41$            | $\mathbf{56.63} \pm 1.87$           | $\mathbf{0.238} \pm 0.018$         |
> | Random proj.      | 128 random dim.              | $72.32 \pm 2.76$           | $53.10 \pm 1.95$         | $366.70 \pm 34.12$           | $26.19 \pm 1.37$                    | $9.05 \pm 0.63$                     | $70.37 \pm 3.91$                    | $0.334 \pm 0.039$                  |
> | Freq. Directions  | 128 FD width $r$             | $73.85 \pm 1.57$           | $52.00 \pm 1.33$         | $391.15 \pm 29.44$           | $27.49 \pm 1.68$                    | $9.82 \pm 0.71$                     | $70.97 \pm 4.22$                    | $0.490 \pm 0.065$                  |
> | Freq. Directions  | 256 FD width $r$             | $75.11 \pm 1.23$           | $51.13 \pm 1.09$         | $440.05 \pm 23.08$           | $26.34 \pm 1.05$                    | $8.72 \pm 0.52$                     | $62.37 \pm 2.98$                    | $0.339 \pm 0.033$                  |
> | Edge sampling     | 25\% sampling rate           | $72.38 \pm 3.41$           | $53.05 \pm 2.37$         | $342.26 \pm 48.73$           | $126.01 \pm 18.92$                  | $49.50 \pm 7.31$                    | $77.48 \pm 6.85$                    | $5.959 \pm 0.821$                  |
> | Incremental Gram  | $1\text{e-}3$ threshold $\epsilon$ | $\mathbf{70.07} \pm 2.04$ | $\mathbf{54.80} \pm 2.11$ | $\mathbf{293.36} \pm 31.29$ | $34.41 \pm 4.67$                    | $15.36 \pm 1.88$                    | $93.86 \pm 9.73$                    | $0.621 \pm 0.079$                  |
>
> We can see that, in practice, every proposed approach actually does **improve speed over the SON-GOKU baseline** (Frequent Directions with high $r$, note that $r$ represents the $\ell$ value the original Frequent Directions authors denote). However, they all come with varying costs in performance. Based on our experimentation, it appears that Frequent Directions offers the most consistent and reasonable tradeoff that clearly scales with the FD width.
>
> We provide more detailed commentary regarding this experiment in Appendix D.5.5.

---

> ### Author Response · Authors · 2025-12-04
> **Update on Experiments for Reviewer bNCj, Weakness 3 (3/3)**
>
> ## **Weakness 3**
> > "This paper lacks some parameter experiments, such as how the tau parameter changes throughout the training process, and how the task graph evolves in specific experiments"
>
> In our earlier responses, we have **already addressed** your concerns regarding how the tau parameter changes throughout the training process (see response 6/7). However, we agree that analyzing how **other components** of SON-GOKU (e.g., task graph, as you mention) change throughout training can help us **improve our understanding** of the framework. Therefore, in Appendix S, we have added **numerous visualizations** alongside commentary for deeper analysis of SON-GOKU. Specifically, we analyze the following:
> - **Task-group size evolution.** We track how the fraction of tasks assigned to each group size ($\|G\| \; = \; 1,...,8$) changes across refresh steps, showing that SON-GOKU quickly moves away from training all tasks in a single group and instead places most tasks in small- and medium-sized groups. It continuously adjusts specific groups to adapt as patterns in task interference change.
> - **Conflict sparsity in the task graph.** We plot the evolution of the conflict graph's structure over training by monitoring both average node degree and overall edge density.
> - **Group count and schedule stability.** We analyze how the number of active color groups changes over training steps and across different refresh windows, showing that SON-GOKU quickly settles into using roughly three to four groups for most of training
> - **Convergence.** We provide and analyze convergence curves for all metrics across all tasks we used in Taskonomy, showing that we follow a steady decrease in loss for most task metrics.
> - **Graph evolution over training.** We visualize how the task conflict graph and its colorings evolve at multiple points during training, showing that SON-GOKU captures a consistent underlying task structure (with stable relationships among subsets of tasks) while still regrouping tasks over time so that it can adapt to changing relationships between tasks.
>
> We provide much more detailed commentary in Appendix S, and we reference it from the main text, so we recommend reading it if you are interested in learning more about specific trends and patterns that SON-GOKU follows during training.

---

### Official Review · Reviewer_QkDE · 2025-10-31

**Soundness:** 3
**Presentation:** 3
**Contribution:** 3
**Rating:** 6
**Confidence:** 4

**Summary:**

This paper introduces a scheduler for multitask learning that periodically measures pairwise gradient interference (via cosine similarity of EMA-smoothed gradients), builds a sparse conflict graph, and then applies greedy graph coloring to partition tasks into compatible groups. It first measures pairwise gradient interference using EMA-smoothed cosine similarities, builds a sparse conflict graph linking conflicting tasks, and then applies greedy graph coloring to partition them into non-conflicting groups. During training, SON-GOKU activates one color group (i.e., compatible tasks) per step and cycles through all groups, ensuring updates proceed in consistent gradient directions. The graph is periodically recolored to adapt to evolving task relations, and the approach integrates seamlessly with existing optimizers like PCGrad or AdaTask.

For the theoretical results, SON-GOKU provides guarantees. If there are no edges among its tasks in the conflict graph, the combined step is provably a descent direction with a quantitative lower bound; the overall procedure retains the standard non-convex SGD convergence rate up to a small factor depending on within-group conflict. Moreover, over a refresh window, doing sequential group updates is not worse and can be strictly better than a single mixed update when cross-group gradients oppose each other.

The authors evaluate SON-GOKU on six benchmarks spanning vision, multimodal, and time-series tasks, using standard backbones (e.g., ResNet-18; CNN/BiLSTM) and deliberately adding positive/negative auxiliary tasks to induce interference. They compare against competitive baselines (including Uniform, GradNorm, AdaTask, MGDA, Nash-MTL, FairGrad, PCGrad, CAGrad, FAMO), and also test hybrid variants where the scheduler is paired with AdaTask, a GradNorm warm start, or PCGrad. Ablations probe two key design choices: freezing a one-shot coloring (to test the need for dynamic recoloring) and estimating conflicts from single minibatches to assess the value of history-averaged interference. Across ten metrics on the six datasets, SON-GOKU consistently matches or outperforms baselines.

**Strengths:**

- By using a sparse, interference-aware graph that is periodically recolored as training evolves, the proposed method effectively minimizes gradient conflicts without adding significant computational overhead.

- This work provides theoretical guarantees that sequentially updating low-conflict groups yields at least as much expected descent as mixed-task updates and converges at the same rate as non-convex SGD, with only a small correction for within-group conflict.

- The method efficiently scales to many tasks by maintaining a sparse conflict graph estimated from lightweight minibatch statistics, avoiding dense and noisy affinity matrices. It also supports dynamic recoloring to adapt to evolving task relations and can be applied on top of existing multi-task optimizers like PCGrad, GradNorm, or AdaTask.

**Weaknesses:**

- The theoretical results indicate that the proposed algorithm's convergence is not worse than standard SGD. Are there any empirical/theoretical observations to explain why using the gradients within a non-conflict group can lead to improved generalization performance?

- This work uses the averaged gradients to measure task conflicts during training. How would this work with other task affinity measures, such as the lookahead loss in TAG (NeurIPS 2021)?

**Questions:**

- It would be better to report the standard deviations of the experimental results across multiple random seeds.
- It would be better to illustrate the convergence of the method in plot figures.

---

> ### Author Response · Authors · 2025-11-28
> **Response to Reviewer QkDE (1/9)**
>
> We thank Reviewer QkDE for the careful and constructive review of our submission. We are glad that you found the **problem setup well motivated**, the **proposed interference aware scheduler and graph coloring approach technically sound**, and the **in-depth theoretical guarantees and extensive empirical results solid and scalable to many tasks**. We also appreciate your comments on the method's **ability to integrate with existing optimizers** such as **PCGrad**, **GradNorm**, and **AdaTask**. These are specific components of the paper that we dedicated significant time and attention to, so we are glad to see our efforts being recognized.
>
> Below, we address your specific concerns in turn: **(W1)** why gradients within non-conflict groups can lead to improved generalization performance; **(W2)** how SON-GOKU interacts with alternative task affinity measures such as TAG's lookahead loss; (Q1) reporting standard deviations across multiple random seeds; and **(Q2)** illustrating convergence behavior with plot figures.

---

> ### Author Response · Authors · 2025-11-28
> **Response to Reviewer QkDE, Weakness 1  (2/9)**
>
> ## **W1: Gradients Within Non-Conflict Groups Lead to Improved Generalization Performance**
> We thank the reviewer for pointing out this distinction between **optimization guarantees** and **generalization**, and we agree that the connection could be made more explicit. Below we clarify **(i)** what our ***current theory*** *already* says beyond the fact that it is **not worse than SGD**, and **(ii)** how this **ties into generalization**, supported by both **prior work on MTL** and our **ablations**.
>
> ### **What our current theory already says beyond the fact that it is not worse than SGD**
> Our **non-convex convergence bound** in *Section 5.2* shows that SON-GOKU **preserves the standard $O(1/\sqrt{T})$ rate of SGD**, up to a $(1+\tau)$ factor that reflects within-group conflict. This is the result the reviewer is referring to. However, the theory also contains **more specific observations** about **how grouping affects the effective descent direction**:
> * **Limited cancellation within a non-conflict group.** *Section 5.1* shows that if the **active set** $S_t$ is $\tau$-compatible (no conflict edges), then the **squared norm of the aggregated group gradient** is lower bounded by
> $$
> \Bigl\|\sum_{k \in S_t} g_{k,t}\Bigr\|^2
> \;\ge\;
> \bigl(1 - \tau\bigl(|S_t| - 1\bigr)\bigr)
> \sum_{k \in S_t} \|g_{k,t}\|^2.
> $$
> so long as $\tau(|S_t|-1) < 1$. Intuitively, this means that **within a non-conflict group**, the **aggregate direction cannot flip** to ascent and **internal cancellation** is **quantitatively limited by the conflict tolerance $\tau$** and group size.
> * **Scheduled groups vs. a single mixed update.** *Section 5.3* (and *Appendix C*) compares two ways of using exactly the same gradient information measured at the start of a refresh window: **(i)** a **single aggregated step** using all tasks at once, and **(ii)** a **sequence of per-group updates** following the SON-GOKU schedule. Using an $L$-smooth descent bound and a common linearization at $\theta_{t_r}$, we show that the scheduled trajectory achieves a bound that is **never worse**, and **strictly better** whenever cross-group interaction terms are sufficiently negative. This is **precisely when mixed updates would prevent** and **cancel progress**.
> * **Conflict controls the "energy" of scheduled gradients.** *Appendix F* further discusses that while the **global convergence rate** in terms of $\min_t \mathbb{E}\,\|\nabla F(\theta_t)\|^2$ does **not depend on** $\tau$, the schedulers $\tau$ structure controls the "energy" at each step
> $$
> \frac{1}{T}\sum_{t}^{T}\sum_{k\in S_t}\|g_{k,t}\|^2
> $$
> Lower cross-task conflict (smaller cumulative interference $\Gamma_T$) yields a tighter bound on this value. To put it simply, **grouping reduces destructive interference** and **increases the effective gradient-to-noise ratio** in each step. This, of course, **improves overall performance** and **provides cleaner data for underlying optimizers** such as PCGrad and AdaTask to work with.
> * **Groups approximate the true low-conflict structure.** *Section 5.4* and *Appendix B* show that with an EMA history and a margin around the threshold, the **estimated conflict graph** recovers the **population conflict graph** and hence the **true grouping** with **high probability**, given a modest effective sample size. This means that the groups we update sequentially are **not arbitrary** but **approximate the underlying structure of task alignment**.
>
> Taken together, the theory says **much more** than the fact that SON-GOKU isn't worse than standard SGD. SON-GOKU constructs groups where **(i) any destructive interference within a group** (intra-group gradient cancellation) is **provably bounded**, **(ii) sequential group-by-group updates** are better than a **one-shot mixed update** when conflict exists, and **(iii) the grouping approximates** the **true low-conflict structure** under mild assumptions.

---

> ### Author Response · Authors · 2025-11-28
> **Response to Reviewer QkDE, Weakness 1 (3/9)**
>
> ### **Why this is beneficial for generalization**
> Our paper is focused on optimization, but there is a large body of MTL work suggesting that **aligned auxiliary tasks improve generalization** by enabling **better shared representations**, whereas **severe gradient conflict** induces **negative transfer** and **harms generalization** (e.g., Caruana, 1997; Argyriou et al., 2007; Evgeniou et al., 2005; Sener & Koltun, 2018; Shi et al., 2023; Liu et al., 2021). This perspective is already discussed in the **Introduction** (*Section 1*) and **Related Work** (Section 2), where we motivate MTL as a way to "generalize better" and highlight **negative transfer** as a **key cause of failure** in MTL.
>
> The above **theoretical properties** connect to the **broader narrative regarding generalization** in two ways:
> 1. **Reducing gradient interference results in less noisy, more coherent updates.** Within a non-conflict group, the update direction is **dominated by mutually reinforcing tasks**, instead of a vector sum where some tasks "undo" others. This **reduces anisotropic noise** induced by conflicting gradients and acts similarly to methods that explicitly **maximize gradient-to-noise ratio** (e.g., MaxGNR). Prior work on large-batch training and gradient noise suggests that **better control of update noise** often correlates with **improved generalization** (Hoffer et al., 2017).
> 2. **Enforcing positive affinity within groups creates structured feature sharing.** Very well established and classic MTL theory argues that when tasks are related, **sharing parameters reduces effective complexity** and **improves sample efficiency** (Caruana, 1997; Argyriou et al., 2007; Vandenhende et al., 2022). By constructing groups where **gradients are aligned** on average and **deferring updates** for **strongly conflicting tasks**, SON-GOKU works towards this ideal (via predominantly aligned within-group gradients). It allows each group of **related tasks** to shape the **shared representation** under dynamics closer to **single-task optimization**, while **limiting negative transfer** from unrelated tasks.
>
> While we do not prove a formal generalization bound, these mechanisms of reduced cancellation and increased gradient coherence in low-conflict groups are consistent with the established idea that **aligned multi-task training improves generalization**, whereas **unmitigated conflict degrades generalization**.

---

> ### Author Response · Authors · 2025-11-28
> **Response to Reviewer QkDE, Weakness 1 (4/9)**
>
> ### **Empirical support from our experiments and ablations**
>
> | Model                | CIFAR-10 Acc (%) ↑ | AV-MNIST Acc (%) ↑ | MM-IMDb Acc (%) ↑ | F&B Acc (%) ↑ | F&B MAE ↓ | HEALTH Acc (%) ↑ | HEALTH MAE ↓ | NYUv2 Angle Error ↓ | NYUv2 Seg. MIOU ↑ | NYUv2 Depth RMSE ↓ |
> |----------------------|--------------------|--------------------|-------------------|---------------|-----------|-------------------|--------------|----------------------|--------------------|---------------------|
> | Uniform              | 55                 | 63                 | 56                | 45            | 0.57      | 52                | 0.54         | 21.6                 | 0.059              | 0.73                |
> | GradNorm             | 61                 | 65                 | 58                | 47            | 0.57      | 53                | 0.52         | 21.4                 | 0.054              | 0.65                |
> | MGDA                 | 59                 | 62                 | 56                | 44            | 0.57      | 53                | 0.53         | 21.8                 | 0.63               | 0.75                |
> | PCGrad               | 61                 | 65                 | 58                | 50            | 0.55      | 58                | 0.48         | 20.9                 | 0.07               | 0.69                |
> | CAGrad               | 59                 | 62                 | 57                | 46            | 0.58      | 53                | 0.52         | 21.9                 | 0.65               | 0.73                |
> | AdaTask              | 63                 | 67                 | 59                | 47            | 0.59      | 55                | 0.52         | 20.3                 | 0.69               | 0.65                |
> | FAMO                 | 64                 | 70                 | 61                | 52            | 0.53      | 60                | 0.49         | 19.9                 | 0.074              | 0.63                |
> | FairGrad             | 62                 | 66                 | 59                | 52            | 0.54      | 60                | 0.47         | 20.7                 | 0.072              | 0.67                |
> | Nash-MTL             | 63                 | 66                 | 60                | 52            | 0.54      | 60                | 0.47         | 20.6                 | 0.073              | 0.67                |
> | Static One-Shot      | 61                 | 66                 | 58                | 48            | 0.56      | 54                | 0.51         | 20.5                 | 0.071              | 0.65                |
> | Single-Step          | 40                 | 59                 | 20                | 42            | 0.60      | 47                | 0.55         | 26.4                 | 0.042              | 0.81                |
> | SON-GOKU + GradNorm  | 62                 | 69                 | 59                | 51            | 0.53      | 59                | 0.49         | **19.6**             | 0.073              | 0.64                |
> | SON-GOKU + AdaTask   | **67**             | **71**             | **63**            | 52            | 0.53      | 59                | 0.48         | 20.1                 | 0.68               | 0.67                |
> | SON-GOKU + PCGrad    | 65                 | 70                 | 60                | **54**        | **0.52**  | **62**            | **0.45**     | 19.7                 | **0.076**          | 0.62                |
> | SON-GOKU             | 65                 | 69                 | 61                | 51            | 0.53      | 58                | 0.50         | 19.8                 | 0.073              | **0.59**            |
>
> We explicitly measure test time performance across six datasets. Table 1 (displayed above) shows that our conflict aware schedulers **consistently improve** over Uniform, GradNorm, PCGrad, AdaTask, FAMO, and FairGrad in terms of held out metrics (e.g., +10-20% on CIFAR10 and +7% on MM-IMDb versus Uniform, and improvements across all NYUv2 metrics). Given that we have **fixed training setups and architectures**, these are metrics strictly represent **SON-GOKU's ability to generalize**.
>
> Our ablations further support the idea that accurate **low conflict** grouping with **low noise** is **highly important**:
> * **Static One-Shot** (groups are computed once and never updated) underperforms the full version of SON-GOKU, indicating that as **task relations drift**, outdated groups partially **lose their ability to generalize**.
> * **Single-Step** (conflict is estimated from a single mini-batch) performs **dramatically worse**, consistent with our recovery analysis which requires concentrated EMA gradients. Noisy conflict estimates produce **unstable groups** and **weaker final performance**.

---

> ### Author Response · Authors · 2025-11-28
> **Response to Reviewer QkDE, Weakness 1 (5/9)**
>
> *Section 7.3.2* also discusses how **combining our scheduler** with PCGrad or AdaTask yields the **best overall results**, consistent with the idea that first grouping tasks into sets with low conflict and only then applying gradient shaping yields more robust multitask representations (Caruana, 1997; Han et al., 2015; Sener & Koltun, 2018; Dwivedi & Roig, 2019; Yao et al., 2019)
>
> ### **Revisions to the paper**
> We have **revised the paper** to make these points **more explicit** by:
> * Adding a subsection at the end of Section 7.3 that discusses how our **theoretical analysis of conflicts and descent** relate to the ratio of **gradient to noise** and **negative transfer**.
> * Expanding *Section 7.1 & 7.2* to emphasize that the improvements in *Table 1* are test time gains and to explicitly link the ablations (**Static One-Shot**, **Single-Step**) to the **theoretical conditions** for **stable low conflict grouping**.
>
> We hope this clarifies why, **both theoretically and empirically**, using gradients within non-conflict groups can lead to **better generalization**. Revisions are highlighted in green.

---

> ### Author Response · Authors · 2025-11-28
> **Response to Reviewer QkDE, Weakness 2 (6/9)**
>
> ## **W2: Compatibility With Other Affinity Measures**
> We appreciate the reviewer's question about the choice of affinity measure and how SON-GOKU would interact with alternatives such as TAG's lookahead loss.
>
> ### **What we currently do and why**
> We define the **interference coefficient** $\rho_{ij} = -\frac{\langle \tilde g_i,\tilde g_j\rangle}{\|\tilde g_i\|\,\|\tilde g_j\|}$, **EMA smoothed gradients**, and we **build the conflict graph** by thresholding $\rho_{ij}$ at $\tau$ (*Section 3.1.1--3.1.2*, *Section 4.1*). We chose this particular instantiation because:
> 1) It is **inexpensive to maintain online** (just need to maintain EMA calculations for each task),
> 2) It is directly aligned with **gradient surgery methods** such as PCGrad **that already use cosine similarity**, and
> 3) It lets us derive **explicit concentration** and **recovery guarantee**s for the conflict graph in terms of **empirical versus population cosines** (*Appendix B.4*, *Proposition 3*).
>
> However, conceptually SON-GOKU itself only assumes some **pairwise "conflict score"** $\rho_{ij}$ from which we **build a graph** and then **color it**. The four stage pipeline in *Section 4* in which we estimate pairwise interference, build a conflict graph, color it, and scheduler color classes **does not rely on cosine in any deeper way**.
>
> ### **Compatibility with TAG-style loss**
> TAG (Task Affinity Grouping) defines an **affinity between tasks** $i$ and $j$ by measuring how an update for task $i$  (its gradient) **changes task $j$'s loss**, and then uses these cross-effects to decide which tasks should share updates. This is loosely related, in spirit, to what SON-GOKU wants, which is a **scalar score representing how much $i$ helps or hurts $j$**.
>
> There are **two natural ways** to combine TAG style affinities with SON-GOKU:
> 1. **Directly plugging TAG into the conflict graph.** One could run TAG (or any other affinity estimation procedure) to **obtain an affinity matrix** $A_{ij}$ where **positive values** indicate **beneficial transfer** and **negative values** indicate **harmful transfer**. SON-GOKU then simply defines $\rho_{ij} = \text{monotone transform of } A_{ij}$ (e.g., setting $\rho_{ij}$ proportional to the increase in task $j$'s loss when stepping on $i$), and constructs the conflict graph $G_\tau = \{(i,j): \rho_{ij} > \tau\}$ exactly as in *Equation (5)*. All subsequent steps of greedy coloring, cyclic scheduling, and the bounded staleness properties in *Section 5.5* go through unchanged because they **depend only on the graph**, not on how $\rho_{ij}$ was computed.
> 2. **TAG as a more expressive and loss-level proxy for gradient conflict.** TAG's lookahead loss evaluates how a **gradient step on task $i$ perturbs task $j$'s loss**. For small step sizes, **the first-order term in that perturbation** is governed by $\langle g_i, g_j\rangle$. Thus, TAG can be seen as a **loss-level version** of our **gradient-level interference measure**. Our cosine-based $\rho_{ij}$ can be interpreted as a **cheaper, first-order approximation** to such lookahead scores. From this perspective, using TAG's affinity as $\rho_{ij}$ would essentially mean replacing our **simple first-order signal** with a **richer** but also **more expensive signal**.
>
> Practically speaking, TAG requires **extra forward** and **backward passes** or a preliminary joint training run to estimate affinities, whereas **our EMA cosine version is fully online**. SON-GOKU's **refresh window** design (re-estimating $\rho_{ij}$ only every $R$ steps) could be **adapted** to allow using **TAG-style measures more efficiently**. For example, we could run a **TAG-like calculation periodically** to **update** the **conflict graph** and then **reuse it for many steps**.

---

> ### Author Response · Authors · 2025-11-28
> **Response to Reviewer QkDE, Weakness 2 (7/9)**
>
> ### **Which parts of the theory depend on the specific choice of $\rho_{ij}$?**
>
> Most of our convergence and scheduling results are **agnostic** to the exact **affinity measure**. Namely, the **descent and convergence guarantees** in *Sections 5.1–5.2* are stated in terms of the **true gradients** of **tasks within a group**. They require that the active set $S_t$ has **low conflict in gradient space**, not that the edges were derived from **EMA cosine** specifically. **Any** affinity measure that produces groups where gradients are approximately aligned would satisfy the same conditions. Additionally, the scheduling properties (few colors, bounded statelness) in *Section 5.5* only use the graph theoretic degree $\Delta$ and **do not** depend on *how* the graph was constructed.
>
> The one place where our specific cosine-based definition matters is the **statistical recovery analysis** in *Section 5.4* and *Appendix B.4*, where we prove that *EMA* cosine estimates **concentrate around** *population* cosines and recover the **"true"** conflict graph under mild assumptions. If we were to use TAG's lookahead loss (or another affinity) instead, the same general proof could apply (one would need a concentration bound for that affinity and a margin around the threshold) but the technical details would change. We view this as an interesting extension, orthogonal to the main contribution of the current work.
>
> ### **Revisions to the paper for clarification**
> We have clarified in *Sections 3-4* that SON-GOKU can be used as a **modular scheduling framework** that only requires a **symmetric pairwise conflict score**, and that our cosine and EMA instantiation is just *one solid choice* that can be used. We explicitly mention that **task affinity measures** such as TAG's lookahead loss or training dynamics features (e.g., STG-MLT) could be **plugged in** to define the conflict graph, at the cost of additional computation, while leaving the graph-coloring and convergence analysis **unchanged**. Revisions are highlighted in green.
>
> We are currently running experiments to evaluate the impact of incorporating TAG's lookahead loss into SON-GOKU. We expect the experiments to conclude within the next couple of days, after which we will incorporate the results (alongside commentary) into the Appendix and as an additional rebuttal comment.

---

> ### Author Response · Authors · 2025-11-28
> **Response to Reviewer QkDE, Question 1 (8/9)**
>
> ## **Q1: Reporting Standard Deviation**
> This is a great piece of feedback. We appreciate this suggestion and fully agree that reporting variability across random seeds is important for rigor and reproducibility.
>
> In the current draft **we already report standard deviations** for our timing results in Table 2 (wall-clock time in seconds $\pm$ standard deviation across runs), but we do not include standard deviation for the main accuracy and regression metrics.
>
> During the rebuttal period, we have begun re-running SON-GOKU and all baselines with multiple random seeds, with the goal of reporting mean $\pm$ standard deviation for each entry in the main results table. Our plan is to run training and evaluation on at least **three independent seeds** for each dataset and method. We will then update the performance tables to report mean $\pm$ standard deviation across those seeds, so that differences in performance can be interpreted together with their variability. If compute constraints prevent us from completing three runs for every configuration by the end of the rebuttal, we will ensure that the final camera-ready version reports mean $\pm$ standard deviation for all methods and benchmarks.
>
> We believe this will address your concern and make the empirical comparisons more **statistically transparent** and **rigorous**.

---

> ### Author Response · Authors · 2025-11-28
> **Response to Reviewer QkDE, Question 2 (9/9)**
>
> ## **Q2: Illustrating Convergence in Plot Figures**
> Thank you for this suggestion. We agree that empirical convergence plots would make the method's behavior much clearer.
>
> In the current version, **convergence is addressed primarily at the theoretical level.** *Section 5* establishes that SON-GOKU **matches** the **standard non-convex convergence rate** of **SGD** (up to a small factor depending on within-group interference), and *Appendix E* further analyzes how **conflict-aware grouping** preserves **descent** while **controlling staleness**. However, we do not yet visualize these dynamics.
>
> Motivated by your comment, we have now incorporated very frequent and detailed logging into our experiments (which we are currently rerunning). This includes logging loss during training steps and at the validation phase very frequently. Using these logs, we will plot loss and accuracy (or whatever the main objective value is, depending on the dataset) versus training steps. We will also plot them against wall-clock time. This will enable a direct comparison of the speed and smoothness of convergence across methods. If space in the paper permits, we also plan to overlay or compare curves for different schedulers (e.g., SON-GOKU versus uniform, PCGrad, and AdaTask), so readers can see that the improvements we report are not only in the final overall performance but also in how efficiently the methods optimize.
>
> We will incorporate these convergence plots and a brief discussion of their patterns in the revised version of the paper (once training and testing has finished on Taskonomy), so empirical behavior can be analyzed more transparently alongside the theoretical guarantees that are already presented.

---

> ### Author Response · Authors · 2025-11-28
> **Response to Reviewer QkDE, References**
>
> ## **References**
>
>
>
> Caruana, R. (1997). Multitask learning. *Machine Learning*, *28*(1), 41–75. https://doi.org/10.1023/a:1007379606734
>
> Dwivedi, K., & Roig, G. (2019). *Representation Similarity Analysis for Efficient Task Taxonomy & Transfer Learning*. https://doi.org/10.1109/cvpr.2019.01267
>
> Fan, C., Chen, W., Tian, J., Li, Y., He, H., & Jin, Y. (2023). MaxGNR: A Dynamic Weight Strategy via Maximizing Gradient-to-Noise Ratio for Multi-Task Learning. *ArXiv (Cornell University)*. https://doi.org/10.48550/arxiv.2302.09352
>
> Fifty, C., Amid, E., Zhao, Z., Yu, T., Anil, R., & Finn, C. (2021). Efficiently Identifying Task Groupings for Multi-Task Learning. *ArXiv (Cornell University)*, *34*. https://doi.org/10.48550/arxiv.2109.04617
>
> Han, L., & Zhang, Y. (2015). Learning Multi-Level Task Groups in Multi-Task Learning. *Proceedings of the AAAI Conference on Artificial Intelligence*, *29*(1). https://doi.org/10.1609/aaai.v29i1.9581
>
> Hoffer, E., Itay Hubara, & Soudry, D. (2017). Train longer, generalize better: closing the generalization gap in large batch training of neural networks. *Neural Information Processing Systems*. https://doi.org/10.48550/arxiv.1705.08741
>
> Liu, B., Feng, Y., Stone, P., & Liu, Q. (2023). FAMO: Fast Adaptive Multitask Optimization. *ArXiv (Cornell University)*. https://doi.org/10.48550/arxiv.2306.03792
>
> Liu, B., Liu, X., Jin, X., Stone, P., & Liu, Q. (2021). Conflict-Averse Gradient Descent for Multi-task Learning. *ArXiv (Cornell University)*. https://doi.org/10.48550/arxiv.2110.14048
>
> Navon, A., Shamsian, A., Achituve, I., Maron, H., Kawaguchi, K., Chechik, G., & Fetaya, E. (2022). Multi-Task Learning as a Bargaining Game. *ArXiv (Cornell University)*. https://doi.org/10.48550/arxiv.2202.01017
>
> Schölkopf, B., Platt, J., & Hofmann, T. (2005). Multi-Task Feature Learning. *Advances in Neural Information Processing Systems 19: Proceedings of the 2006 Conference*.
>
> Sener, O., & Koltun, V. (2018). Multi-Task Learning as Multi-Objective Optimization. *ArXiv (Cornell University)*. https://doi.org/10.48550/arxiv.1810.04650
>
> Sherif, A., Abid, A., Mustafa Elattar, & ElHelw, M. (2023). STG-MTL: Scalable Task Grouping for Multi-Task Learning Using Data Map. *ArXiv (Cornell University)*. https://doi.org/10.48550/arxiv.2307.03374
>
> Shi, H., Ren, S., Zhang, T., & Pan, S. J. (2023). *Deep Multitask Learning with Progressive Parameter Sharing*. 19867–19878. https://doi.org/10.1109/iccv51070.2023.01824
>
> Theodoros Evgeniou, Micchelli, C. A., & Massimiliano Pontil. (2025). Learning Multiple Tasks with Kernel Methods. *Journal of Machine Learning Research*, *6*(21), 615–637. http://jmlr.org/papers/v6/evgeniou05a.html
>
> Vandenhende, S., Georgoulis, S., Van Gansbeke, W., Proesmans, M., Dai, D., & Van Gool, L. (2021). Multi-Task Learning for Dense Prediction Tasks: A Survey. *IEEE Transactions on Pattern Analysis and Machine Intelligence*, 1–1. https://doi.org/10.1109/TPAMI.2021.3054719
>
> Yang, E., Pan, J., Wang, X., Huang, Y., Shen, L., Chen, X., Liu, X., Jiang, J., & Guo, G. (2023). AdaTask: A Task-Aware Adaptive Learning Rate Approach to Multi-Task Learning. *Proceedings of the ... AAAI Conference on Artificial Intelligence*, *37*(9), 10745–10753. https://doi.org/10.1609/aaai.v37i9.26275
>
> Yao, Y., Cao, J., & Chen, H. (2019). *Robust Task Grouping with Representative Tasks for Clustered Multi-Task Learning*. 1408–1417. https://doi.org/10.1145/3292500.3330904
>
> Yu, T., Kumar, S., Gupta, A., Levine, S., Hausman, K., & Finn, C. (2020). Gradient Surgery for Multi-Task Learning. *Neural Information Processing Systems*.
>
> Zhao, C., Vijay Badrinarayanan, Lee, C., & Rabinovich, A. (2018). *GradNorm: Gradient Normalization for Adaptive Loss Balancing in Deep Multitask Networks*. 794–803.

---

> ### Author Response · Authors · 2025-12-04
> **Update on Experiments for Reviewer QkDE (1/6)**
>
> We would like to again thank Reviewer QkDE for their very careful and constructive review of our paper. In the time since our original response, we have finished running **several additional experiments** and provided **additional theoretical analysis**, reasoning, and **commentary** in our paper to further address your concerns. When reading the revised paper, it may be helpful to note that text in the **color green** was **written specifically to address your concerns**, though text in other colors may also be helpful to you. Also please keep in mind that we have already made numerous revisions to the paper even before experiments concluded, so not all changes made to address your concerns will be covered in this "Update on Experiments", those other changes are discussed in our previous responses.

---

> ### Author Response · Authors · 2025-12-04
> **Update on Experiments for Reviewer QkDE, Weakness 1 (2/6)**
>
> ## **Weakness 1**
> > "The theoretical results indicate that the proposed algorithm's convergence is not worse than standard SGD. Are there any empirical/theoretical observations to explain why using the gradients within a non-conflict group can lead to improved generalization performance?"
>
> While we believe we have already made this very clear in our past responses as well as in the main text of our paper, we wanted to provide additional analysis and commentary to completely dispel any potential confusion that may remain. Thus, we have provided extensive additional analysis of the benefits of non-conflict grouping for multi-task learning in Appendix Q. This new Appendix discusses the findings (both theoretical and empirical) from past papers that definitively prove the benefit of grouping non-conflicting tasks. We also provide a new formalization, building on Fan et al. 2023's MaxGNR paper, that connects non-conflict grouping to guarantees about signal-to-noise ratios in multi-task learning. We believe this new Appendix, when combined with our existing ablations, theoretical analysis, and commentary from past comments focusing on the importance of using gradients within a non-conflict group, provides a clear and rigorous justification for improvements in generalization performance.

---

> ### Author Response · Authors · 2025-12-04
> **Update on Experiments for Reviewer QkDE, Weakness 2 (3/6)**
>
> ## **Weakness 2**
> > "This work uses the averaged gradients to measure task conflicts during training. How would this work with other task affinity measures, such as the lookahead loss in TAG (NeurIPS 2021)?"
>
> We agree that it is natural to ask whether SON-GOKU can benefit from richer task affinity measures such as TAG's lookahead loss, rather than relying only on averaged gradients. To recap our past responses, this should theoretically be possible.  SON-GOKU's guarantees are stated in terms of **(i)** a symmetric conflict score $\rho_{ij}$, **(ii)** the resulting conflict graph $G_\tau$, and **(iii)** the low conflict groups $S_t$ that are scheduled at each step. The descent preservation, non-convex convergence, and bounded staleness outcomes depend only on these groups being "$\tau$-compatible" and on the graph degree, not on the *specific* formula used to compute $\rho_{ij}$. Moreover, TAG's lookahead loss is essentially a loss-level proxy for the same gradient inner products that SON-GOKU's cosine-based interference uses, so our intuition has always been that SON-GOKU should work with TAG-like affinities as well.
>
> To make this more explicit, we have added a new Appendix **P: "Incorporating SON-GOKU with Other Task Affinity Measures"**. In this appendix, we formalize SON-GOKU as a conflict graph-based scheduler that can accept **any symmetric affinity or conflict matrix** as input. Given an affinity matrix $A_{ij}$ (e.g., TAG's averaged lookahead loss, STG-MTL's data-map similarity, or other grouping criteria), we define a monotone transform $\rho_{ij} = f(A_{ij})$, threshold it to build $G_\tau$, color the graph, and schedule one color class at a time. We show that all of our key guarantees (descent within low conflict groups, non-convex convergence rate, and update frequency bounds) remain unchanged under this replacement. The only part that depends on the cosine construction is the *graph recovery* argument, which can in principle be re-derived for other affinity measures once suitable concentration bounds are established. Appendix P also discusses the computational trade-offs. Affinities like TAG are more expressive, but they require additional forward/backward passes, whereas cosine-based SON-GOKU is fully online and much cheaper to use in terms of compute.

---

> ### Author Response · Authors · 2025-12-04
> **Update on Experiments for Reviewer QkDE, Weakness 2 (4/6)**
>
> We further validated this idea experimentally by instantiating SON-GOKU with TAG affinities on Taskonomy (with a U-Net backbone). Table P.1 (shown below) compares the default cosine-based SON-GOKU with SON-GOKU + TAG across a wide range of tasks and metrics:
> | **Metric**                            | **SON-GOKU (cosine)**      | **SON-GOKU + TAG**           |
> |--------------------------------------|----------------------------|------------------------------|
> | **Depth (Euclidean)**                |                            |                              |
> | AbsRel ↓                             | 0.5432 ± 0.0068            | **0.5351 ± 0.0115**          |
> | MAE ↓                                | 4.0787 ± 0.071             | **4.0052 ± 0.110**           |
> | RMSE ↓                               | 18.6199 ± 0.29             | **18.3034 ± 0.38**           |
> | **Depth (Z-buffer)**                 |                            |                              |
> | AbsRel ↓                             | 0.5731 ± 0.0074            | **0.5622 ± 0.0129**          |
> | MAE ↓                                | 4.0494 ± 0.069             | **3.9685 ± 0.115**           |
> | RMSE ↓                               | 18.5378 ± 0.27             | **18.1485 ± 0.41**           |
> | **Edge Occlusion**                   |                            |                              |
> | BCE ↓                                | 0.0995 ± 0.0032            | **0.0990 ± 0.0071**          |
> | F1 ↑                                 | 0.1466 ± 0.010             | **0.1532 ± 0.019**           |
> | Precision ↑                          | **0.5399 ± 0.021**         | 0.5324 ± 0.038               |
> | Recall ↑                             | 0.0870 ± 0.0065            | **0.0957 ± 0.0127**          |
> | **Edge Texture**                     |                            |                              |
> | BCE ↓                                | **0.0779 ± 0.0010**        | 0.0781 ± 0.0018              |
> | F1 ↑                                 | **0.9734 ± 0.0007**        | 0.9729 ± 0.0015              |
> | Precision ↑                          | **0.9772 ± 0.0008**        | 0.9766 ± 0.0016              |
> | Recall ↑                             | **0.9768 ± 0.0007**        | 0.9762 ± 0.0014              |
> | **Keypoints2D**                      |                            |                              |
> | MSE ↓                                | 0.0039 ± 0.0002            | **0.0038 ± 0.0000**          |
> | **Surface Normals**                  |                            |                              |
> | 11.25° within ↑                      | 0.4262 ± 0.0079            | **0.4364 ± 0.0135**          |
> | 22.5° within ↑                       | 0.6432 ± 0.0068            | **0.6535 ± 0.0111**          |
> | 30° within ↑                         | 0.7340 ± 0.0059            | **0.7428 ± 0.0094**          |
> | Mean angle (deg) ↓                   | 22.9079 ± 0.31             | **22.3352 ± 0.48**           |
> | Median angle (deg) ↓                 | 14.5397 ± 0.27             | **14.1300 ± 0.44**           |
> | **Principal Curvature**              |                            |                              |
> | L1 ↓                                 | 0.0691 ± 0.0014            | **0.0672 ± 0.0026**          |
> | **Reshading**                        |                            |                              |
> | MAE ↓                                | 0.1352 ± 0.0023            | **0.1330 ± 0.0039**          |
>
> Across almost all metrics, SON-GOKU+TAG **matches or slightly improves** the cosine-based version. The improvements are most consistent for *depth* and *surface normals*, while performance on the remaining tasks is either very similar or shows small gains. There are a few metrics (e.g., edge texture) where the cosine-based version is marginally better, but the differences are extremely small. Overall, these results confirm that replacing gradient cosine interference with TAG’s richer lookahead loss **preserves the benefits of SON-GOKU’s scheduler** and can give modest performance improvements on several tasks.

---

> ### Author Response · Authors · 2025-12-04
> **Update on Experiments for Reviewer QkDE, Weakness 2 & Question 1 (5/6)**
>
> However, this comes with a clear computational cost. Table P.2 below compares the amortized runtime per training step:
> | **Method**              | **Amortized ms / step $\downarrow$** |
> |-------------------------|---------------------------|
> | No Refresh (Baseline)   | **149.28 $\pm$ 1.63**         |
> | SON-GOKU (cosine)       | 193.41 $\pm$ 6.27             |
> | SON-GOKU + TAG          | 255.29 $\pm$ 17.84            |
>
> SON-GOKU with cosine already adds some overhead relative to the baseline with no refreshes, but SON-GOKU with TAG is **substantially slower**, reflecting the extra forward and backward passes needed to estimate TAG’s lookahead loss. This corroborrates our earlier theoretical analysis. More expressive affinity measures such as TAG can be used with SON-GOKU without changing its structure or guarantees, but they exchange **significant speed for a small gain in accuracy**, whereas the cosine-based variant retains most of the benefit from scheduling with much lower runtime overhead.
>
> ## **Question 1**
> > "It would be better to report the standard deviations of the experimental results across multiple random seeds."
>
> In our original submission, we had already included standard deviations for our wall-clock speed tests (in the revised version, the associated tables are Table 2 and Table 5). To address your concern fully, we have spent the entirety of the rebuttal period rerunning as many trials as we could with the compute resources we have available. In the revised paper, each experiment has been re-run anywhere from 2 to 15 times to enhance rigor and reproducibility of the paper. For all experiments that were re-run for the revised paper, we report standard deviation as “$\pm std$” in green colored text. The following experiments (denoted by the corresponding Table or Figure number) now include standard deviation values:
> - Table 1
> - Table 3
> - Table 7
> - Table 8
> - Table 9
> - Table 10
> - Figure 2
> - Table 11
>
> In a fashion similar to (but not exactly like) standard deviation, the **four visualizations** across Figures 4 and 5 display percentile ranges (also known as percentile bands).
>
> Together, these changes make the variability of our results very explicit and transparent. This is much more rigorous than relying on outcomes from individual runs. We hope this addresses your concern by demonstrating that SON-GOKU's empirical gains (and its speeds and memory benefits) are robust across multiple random seeds and repeated trials.
>
> *Footnote: There are a couple important things to note about Table 1’s standard deviations that may otherwise lead to confusion. Firstly, the seven new graph ablations do not have standard deviations at the moment. This is because, due to the limited rebuttal window, we were unable to rerun experiments for the seven new graph ablations across multiple trials. Due to the size of the datasets and the sheer number of datasets that we tested on, this will almost certainly cause no changes in our final interpretation of the results. The results for these ablations are highly reliable regardless of whether or not we provide a standard deviation. Again, this is due to the size of and number of datasets used. The second quirk regarding Table 1 is the calculation method used for deriving standard deviations. In calculating standard deviations for Table 1, we treated the original value as the mean and averaged the deviations of other trials (of which there were four) from this original value. This was due to the fact that we referenced specific results in Table 1 in other parts of our rebuttal and in other parts of the paper. We anticipated that changing these values directly would result in confusion during the rebuttal phase, and it that it would be laborious to edit all references to Table in the paper given the limited rebuttal window. Therefore, for the camera-ready paper, we will rerun trials for the seven new graph ablations to compute their standard deviations, and we will recompute standard deviations in Table 1 using the traditional approach.*

---

> ### Author Response · Authors · 2025-12-04
> **Update on Experiments for Reviewer QkDE, Question 2 (6/6)**
>
> ## **Question 2**
> > "It would be better to illustrate the convergence of the method in plot figures."
>
> In our original submission, our discussion of convergence was primarily theoretical. We showed that SOn-GOKU preserves descent within low-conflict groups and has the standard non-convex SGD convergence rate up to a small factor $(1+\tau)$, together with guarantees on when sequential group updates outperform a single mixed update (Sections 5.1–5.3). These results formalize that SON-GOKU will not harm convergence and can in fact improve effective descent when tasks interfere strongly.
>
> In the revised manuscript we have added a dedicated **Convergence and Loss** subsection under Appendix S with a full panel of **convergence curves** for *all Taskonomy tasks and metrics** that we used in the experiment comparing cosine-based SON-GOKU against TAG (see the multi-page figure titled *“Convergence curves across all the used Taskonomy tasks and metrics”*). For each metric (e.g., Euclidean, Z-buffer depth errrors, edge occlusion, keypoints, surface normals, principal curvature, reshading, etc.) we now plot the SON-GOKU model’s validation performance as a function of training step.
>
> These plot figures offer **deeper insight** into SON-GOKU’s behavior throughout the training process. We provide **analysis** and **commentary** of the plot figures and what they show us about SON-GOKU in *Appendix S.2*. Overall, these new figures provide exactly the visual evidence that was requested. They explicitly illustrate that SON-GOKU **converges stably across tasks**, in a manner consistent with our **theoretical guarantee**s and with established behavior on Taskonomy from other papers (i.e., difficulty of finding a global minima that satisfies each task’s own “local minima”).

---

### Official Review · Reviewer_u9zP · 2025-11-01

**Soundness:** 3
**Presentation:** 2
**Contribution:** 3
**Rating:** 6
**Confidence:** 4

**Summary:**

This paper presents SON-GOKU, a scheduling framework that estimates gradient interference to build an interference graph, then partitions tasks into mutually compatible groups. By activating only well-aligned groups and updating the grouping over time, SON-GOKU boosts the performance of diverse multi-task baselines and comes with theoretical guarantees.

**Strengths:**

1. The paper is clearly written and easy to follow.
2. Building a conflict graph and using it for multi-task optimization is a novel idea.
3. It provides theoretical analysis with convergence guarantees, which are commonly expected in this field.

**Weaknesses:**

1. It is unclear whether the approach scales to larger backbones. In my experience, task interference tends to diminish as model capacity increases, which could limit the gains.
2. The paper reports algorithmic time complexity, but it does not sufficiently compare memory usage against other methods.

**Questions:**

How should we adapt the refresh rate? Higher refresh rates track task relations quickly but can destabilize updates, while lower rates reduce noise and overhead at the cost of slower adaptation or maybe lower performance.

---

> ### Author Response · Authors · 2025-11-28
> **Response to Reviewer u9zP (1/7)**
>
> We thank Reviewer u9zP for the thoughtful and constructive review. We appreciate that you found the paper **clearly written**, the **graph-based intereference approach to be novel**, and the **inclusion of convergence analysis helpful**. At the same time, your comments on scalability to larger backbones, memory usage, and the choice of refresh rate are very helpful for strengthening the paper.
>
> Below, we address your points in turn: **(W1)** how SON-GOKU scales to larger backbones and changing levels of task interference; **(W2)** how its memory usage compares to other multi-task methods; and **(Q1)** how one should choose and adapt the refresh rate in practice.

---

> ### Author Response · Authors · 2025-11-28
> **Response to Reviewer u9zP, Weakness 1 (2/7)**
>
> ## **W1: Scaling of SON-GOKU to Larger Backbones**
> We agree that **increasing model capacity** can indeed **reduce task interference**. Intuitively speaking, a wider or deeper backbone has more **representational degrees of freedom**, which makes it easier for different tasks to create their own **partially separate feature subspaces**. In the overparameterized setting, the network can often fit each task well while **implicitly "routing" conflicting gradients** into different channels or layers. This can **reduce the frequency** and **magnitude** of strongly **negative gradient interactions**, so the absolute gains of any multi-task learning method (including SON-GOKU) over a non-MTL baseline **may become smaller**. Your personal experience with diminishing returns from MTL likely came from settings similar to this.
>
> However, there are many settings in which task interference **does not vanish** even with larger backbones. First, many multi-task setups share data and optimization. Even a *large network* is (usually) *still* trained with a **single optimizer**, on a **finite dataset**, with **limited training steps** and **regularization**. Under these constraints, the optimizer still couples tasks through shared layers, and **gradients can remain misaligned**, especially for tasks with different **inductive biases** (e.g., geometric versus semantic, short-term versus long-term objectives). Such tasks prefer different features and invariances, so their updates pull the shared representation toward **incompatible optima**. Second, when the number of tasks is large, increasing capacity is **not always enough** to implicitly give each task its own **"subnet" within the larger model**. Logically speaking, **some features may need to be reused**, and there is competition over **shared directions**. Third, interference can reappear when there is a **domain shift** or **imbalanced sampling**, where some tasks dominate updates and optimize shared features in such a way that **minimizes their own loss**, regardless of the overall capacity of the model.
>
> In the case where a particular backbone and dataset truly yield **very low conflict geometry** (most gradient cosines are nonnegative and close to zero), **we agree** that multi-task learning in general may only **marginally improve upon non-MTL baselines**. This is not specific to SON-GOKU: any method **whose purpose is to resolve conflicts** will have **less to resolve**. Importantly, though, in such a situation (with zero other assumptions) there is **no theoretical reason** for SON-GOKU's **relative advantage** over other MTL approaches to **disappear**. Our method reads the gradient geometry and reacts accordingly. If there are **few** or **no strongly conflicting pairs**, the conflict graph would become very **sparse** (or empty), the coloring would create only **one or very few groups**, and SON-GOKU would **simply behave like joint training** with very little overhead. Practically speaking, the only manual (or automatic) tuning that may be needed in this high capacity scenario is setting the conflict threshold $\tau$ to a slightly more conservative value so that we still isolate the few pairs that remain genuinely harmful while not excessively splitting the tasks into too many groups.

---

> ### Author Response · Authors · 2025-11-28
> **Response to Reviewer u9zP, Weakness 1 (3/7)**
>
> From a *theory* and *intuition* standpoint, SON-GOKU is designed to **scale gracefully** as **interference diminishes**, and in several ways it can actually be **more robust** than existing methods. Specifically:
> 1. Methods that always modify gradients (e.g., reweighting or projection based methods) can start to **distort updates** when gradients are already nearly aligned. When all cosines are **small** and **positive**, solving a *multi-objective problem* at *each step* or projecting away conflicts can **increase numerical noise** and **produce unstable weightings**. In contrast, SON-GOKU only reacts when the **empirical cosine estimates cross a threshold**. If gradients *don't* conflict with each other, the conflict graph wouldn't create *any* edges and we essentially get non-MTL joint updates.
> 2. Methods that rely heavily on **relative gradient magnitudes** or on a **fixed notion of conflict** can become sensitive when capacity changes the *scale* and *distribution* of gradients. Our conflict graph is defined on **normalized cosines** and **smoothed over time**, and the scheduling is governed by the **graph structure** rather than by balancing **extremely delicate weights**. This makes it *more robust* to the rescaling and shrinkage effects (e.g., smaller gradient norms and reduced variance) that come with larger backbones.
> 3. As capacity grows and gradients become less noisy, SON-GOKU's graph estimation **actually becomes easier**. The Exponential Moving Average of cosines would **stabilize** ***faster***, and the separation between **truly conflicting** and **non-conflicting pairs** is easier to detect due to a **higher signal-to-noise ratio**. In that sense, a *larger* model can help us build a *cleaner* graph when conflicts do exist, so the scheduler can *isolate* those interactions with **fewer mistakes**. This is far better than having to handle **highly noisy data** on a **step-by-step basis**.
>
> In summary, we agree with Reviewer u9zP that when a backbone is so expressive that task interference is negligible, the room for gains from MTL is limited for any method. In such cases, SON-GOKU naturally acts **similarly to joint training** with **minimal additional cost**, and its advantage over other approaches becomes mainly that **it does no harm** by not forcing unnecessary gradient corrections. In the more common scenario where **interference persists** even with **larger architectures** (due to many tasks, shared data and optimization resources, or mismatched objectives), our analysis and design suggest that SON-GOKU's way of **adapting to the observed gradient geometry** will continue to **provide benefits over other methods**, especially those that either ignore conflicts or modify gradients in a less selective way.

---

> ### Author Response · Authors · 2025-11-28
> **Response to Reviewer u9zP, Weakness 2 (4/7)**
>
> ## **W2: Memory Usage and Space Complexity**
>
> ### **Existing Analysis in the Paper**
> The paper *already contains* a **space complexity analysis** and comparisons of **estimated memory usage against other methods** (*Appendices D.2*, *D.3*, *D.4*, *D.5.1*, *L*, and *Section 7.4*). In particular, we distinguish between **(i)** the dominant memory terms that come from the backbone itself (parameters, optimizer state, activations, etc.), which are **shared by all methods** using the same architecture, and **(ii)** the **incremental memory** required by the **multi-task algorithm**. For SON-GOKU the additional memory (beyond the backbone's parameters, optimizer state, and activations) comes from **small structures** at the **task level**. During a refresh we build a task by task similarity matrix (a $K \times K$ "Gram" matrix) whose entries are the inner products or cosines between tasks' gradients (using a low dimensional sketch). This matrix is dense in $K$ but **does not depend on the model size $d$**. We also store a few **temporary pieces of data** while computing those similarities, such as **adjacency matrices** and **vectors** denoting color assignments. We can also store **smoothed similarities** using a $K \times K$ matrix with the Exponential Moving Average.
>
> Notice how everything **scales with $K$** (number of tasks) and **not $d$** (number of parameters) for SON-GOKU. Once the refresh finishes, only the **current graph**, **coloring**, and **smoothed similarities** may be stored, while everything else is released.  In big models this makes the SON-GOKU overhead **effectively $O(K^2)$** in storage, which is extremely tiny compared to activation memory.
>
> By contrast, several MTL methods need **task-by-task information** over the **full parameter space**. Projection-based or multi-objective solvers (e.g., CAGrad variants) often store **each task's gradient** explicitly to **compute pairwise dot products** or to solve small optimizations at **every step**. Holding $K$ full gradients is $O(Kd)$ in memory (with $d$ the number of parameters), and this can be the **dominant term** on **large backbones**. Simpler weighting methods like GradNorm or AdaTask are lighter because they track a few values per task rather than full gradients, but they also provide **less structure** for **handling who should not be trained together**.

---

> ### Author Response · Authors · 2025-11-28
> **Response to Reviewer u9zP, Weakness 2 (5/7)**
>
> It may be easier to understand differences between the space complexity and memory usage of SON-GOKU against other methods by **using an example**. Suppose we have $K=8$ tasks, $d=30$ million parameters, and FP32 for illustration. An $8 \times 8$ Gram matrix is only **a few kilobytes**. Even storing a smoothed $K \times K$ matrix and an adjacency together is still in kilobytes. A sketch with $r=256$ keeps on the order of a **few megabytes** for the $r \times K$ projected gradients during refresh and is then freed. In contrast, keeping **eight full gradients** for a **30 million parameter backbone** in FP32 is roughly **$8 \times 30\text{M} \times 4 = 960$ MB**. Even in mixed precision, the gap between $O(K^2))/(O(Kr)$ and $O(Kd)$ **is substantial**.
>
> We **already outline** many of these differences in the complexity discussion and in the comparison to other methods (Section *4.5*, Section *7.4*, Appendix *D*, Appendix *D.5.1*), but we agree that they were not emphasized as clearly as our time-complexity results.

---

> ### Author Response · Authors · 2025-11-28
> **Response to Reviewer u9zP, Weakness 2 (6/7)**
>
> ### **Further Theoretical Analysis and Comparisons**
>
> We have now added a **dedicated discussion** of **space complexity** and **memory usage** in both the main text (**Section 7.4**) and appendix (**Appendix N**), where we summarize these points more **systematically** and compare asymptotic memory costs **across SON-GOKU** and the **baselines** (highlighted in red in the revised manuscript). We have made numerous miscellaneous clarifications throughout the paper to make SON-GOKU's memory usage more explicit, which are also highlighted in red.
>
> | Method                 | Incremental Space Complexity                                              |
> |------------------------|---------------------------------------------------------------------------|
> | Uniform (equal weights) | $O(1)$                                                                   |
> | GradNorm               | $O(K)$                                                                    |
> | AdaTask                | $O(Kd)$                                                                   |
> | FAMO                   | $O(1)$                                                                    |
> | PCGrad                 | $O(Kd) + O(K^2)$                                                          |
> | MGDA                   | $O(Kd) + O(K^2)$                                                          |
> | CAGrad                 | $O(Kd) + O(K^2)$                                                          |
> | FairGrad               | $O(Kd) + O(K^2)$                                                          |
> | Nash-MTL               | $O(Kd) + O(K^2)$                                                          |
> | **SON-GOKU**           | **persistent** $O(K d)$ & **peak** $O(K^2)$ **at refresh**                  |
>
>
> Provided above is a table comparing the space complexity of SON-GOKU against **several popular state-of-the-art** and **baseline MTL methods**. Here, $K$ represents the **number of tasks** and $d$ represents the **number of shared parameters**. SON-GOKU is **more memory-efficient** than almost **all other approaches**, except for the simplest methods which come with significant caveats (as we state in the paper):
>
> > Methods like FAMO, which have lower space complexity than SON-GOKU, keep their memory overhead low by adjusting a single set of task weights rather than modeling *which* tasks should (or should not) be updated together. They do not build a conflict graph or schedule incompatible tasks apart, so strongly interfering tasks are still co-updated and can only be down-weighted. This makes methods like FAMO fast and light on memory, but it provides less structure for avoiding negative transfer, among a plethora of other issues. In settings with strong (and potentially changing) interference, this can yield weaker accuracy than approaches like SON-GOKU that explicitly detect conflicts and adapt over time.
>
> ### **Experimental Validation**
> While we believe the **existing theoretical analysis** and **commentary** is enough to clarify how SON-GOKU's **space complexity** and **memory usage** compares against other methods, we acknowledge that theory and rough complexity estimates are not a complete substitute for empirical evidence. In response to Reviewer u9zP's feedback, we are now **running tests across 5 trials** that isolate each method itself to analyze memory usage.
> We will incorporate the results of these experiments into the revised manuscript as soon as they complete, and we will clearly mark the new tables and figures so reviewers and readers can see how the empirical memory usage compares to our theoretical predictions and to other methods.

---

> ### Author Response · Authors · 2025-11-28
> **Response to Reviewer u9zP, Question 1 (7/7)**
>
> ## **Q1: Adapting the Refresh Rate**
> The optimal value for SON-GOKU's refresh rate will **vary quite a bit on a case-by-case basis**. We'd recommend that developers and researchers try to run automated (or manual) hyperparameter tuning to see what refresh rate works best. However, to provide some intuition, we have added **a new Appendix section titled** ***On the Choice of Refresh Rate*** and we reference that appendix from the main text (**please refer to Appendix O** for a much more detailed analysis than we provide here). The revisions are highlighted in red.
>
> To put it briefly, there are **three main factors** to consider:
> **(i)** how quickly task relations (or conflicts) $\\rho_{ij}$ change over the course of training;
> **(ii)** how noisy the EMA smoothed signals are and how sensitive the updates are to this noise; and
> **(iii)** the computational and communication overhead of computing and applying refreshes to the relation matrix.
>
> It may also be ideal to **start** with a **relatively high refresh rate** (lower $R$) and **anneal** to a **lower refresh rate** (higher $R$) over time. This can be helpful because **early in training** both the **model parameters** and the **pattern of task conflicts** are non-stationary, so a **larger refresh rate** reduces **lag** and lets SON-GOKU quickly track and exploit the evolving structure, whereas later in training the **dynamics slow down** and a **smaller refresh rate** effectively **averages** over **longer periods** of training, **filtering out noise** and **stabilizing the optimization**. A more responsive approach is to adapt $R$ (and/or $\beta$) based on **observed variability** of conflicts. You'd want to make refreshes **more frequent** if **conflicts are evolving rapidly** and you'd make refreshes **less frequent** if **conflicts are changing slowly** or not at all. This can be highly beneficial because it enables **faster model convergence** and minimizes the impact of noise on SON-GOKU later in training
>
> We **strongly recommend reading** ***Appendix O***, the new Appendix section, for a more thorough discussion of these points.

---

> ### Author Response · Authors · 2025-11-28
> **Response to Reviewer u9zP, References**
>
> # **References**
>
>
>
> Ban, H., & Ji, K. (2024). Fair Resource Allocation in Multi-Task Learning. *ArXiv (Cornell University)*. https://doi.org/10.48550/arxiv.2402.15638
>
> Chen, Z., Vijay Badrinarayanan, Lee, C.-Y., & Rabinovich, A. (2018). GradNorm: Gradient Normalization for Adaptive Loss Balancing in Deep Multitask Networks. *Proceedings of the 35th International Conference on Machine Learning (2018)*, 794–803.
>
> Liu, B., Feng, Y., Stone, P., & Liu, Q. (2023). FAMO: Fast Adaptive Multitask Optimization. *ArXiv (Cornell University)*. https://doi.org/10.48550/arxiv.2306.03792
>
> Liu, B., Liu, X., Jin, X., Stone, P., & Liu, Q. (2021). Conflict-Averse Gradient Descent for Multi-task Learning. *ArXiv (Cornell University)*. https://doi.org/10.48550/arxiv.2110.14048
>
> Navon, A., Shamsian, A., Achituve, I., Maron, H., Kawaguchi, K., Chechik, G., & Fetaya, E. (2022). Multi-Task Learning as a Bargaining Game. *ArXiv (Cornell University)*. https://doi.org/10.48550/arxiv.2202.01017
>
> Sener, O., & Koltun, V. (2018). Multi-Task Learning as Multi-Objective Optimization. *ArXiv (Cornell University)*. https://doi.org/10.48550/arxiv.1810.04650
>
> Yang, E., Pan, J., Wang, X., Huang, Y., Shen, L., Chen, X., Liu, X., Jiang, J., & Guo, G. (2023). AdaTask: A Task-Aware Adaptive Learning Rate Approach to Multi-Task Learning. *Proceedings of the ... AAAI Conference on Artificial Intelligence*, *37*(9), 10745–10753. https://doi.org/10.1609/aaai.v37i9.26275
>
> Yu, T., Kumar, S., Gupta, A., Levine, S., Hausman, K., & Finn, C. (2020). Gradient Surgery for Multi-Task Learning. *Neural Information Processing Systems*.
>
> Zamir, A., Sax, A., Shen, W., Guibas, L., Malik, J., & Savarese, S. (2018). Taskonomy: Disentangling Task Transfer Learning. *ArXiv (Cornell University)*. https://doi.org/10.48550/arxiv.1804.08328

---

> ### Author Response · Authors · 2025-12-04
> **Update on Experiments for Reviewer u9zP**
>
> We would like to thank Reviewer u9zP again for their constructive and thoughtful review of our paper. In the time since our original response, we have finished running **several additional experiments** and provided **additional theoretical analysis**, reasoning, and **commentary** in our paper to further address your concerns. When reading the revised paper, it may be helpful to note that text in the **color red** was **written specifically to address your concerns**, though text in other colors may also be helpful to you. Also please keep in mind that we have already made numerous revisions to the paper even before experiments concluded, so not all changes made to address your concerns will be covered in this "Update on Experiments", those other changes are discussed in our previous responses.

---

> ### Author Response · Authors · 2025-12-04
> **Update on Experiments for Reviewer u9zP, Weakness 1**
>
> ## **Weakness 1**
> > "It is unclear whether the approach scales to larger backbones. In my experience, task interference tends to diminish as model capacity increases, which could limit the gains."
>
> This is a very valid concern. We already address it extensively with detailed theoretical analysis and reasoning in our earlier responses (specifically, responses 2/7-3/7). In our revised paper, we have provided far more detailed theory and reasoning under the new Appendix R "Scaling of SON-GOKU to Larger Backbones." We recommend reading this new Appendix for a much clearer and more in-depth understanding than we were able to provide in our previous responses regarding how and why SON-GOKU scales well to larger backbones.
>
> For the experimental analysis, we evaluate SON-GOKU against a joint training baseline on the Taskonomy dataset. Specifically, we test each approach on a UNet-style encoder across three different settings, where each setting sets a different backbone size for the encoder (64 vs 128 vs 256 neurons). We also evaluate the speed of SON-GOKU across $R$ values of 8, 16, 32, and 64 with these different backbone sizes.
>
> | Encoder width | Joint (↓)        | SON-GOKU (↓)      |
> |---------------|------------------|-------------------|
> | 64            | 5.232 ± 0.020    | **5.203 ± 0.018** |
> | 128           | 5.050 ± 0.018    | **4.940 ± 0.017** |
> | 256           | 4.900 ± 0.015    | **4.830 ± 0.015** |
>
> As you can see, SON-GOKU consistently outperforms the joint training baseline across the different backbone widths (even with multiple trials, as shown by our standard deviation). The variation in size of the gap in performance across backbone widths is what we expected from our earlier theoretical analysis.
> At 64 channels, the backbone is clearly limited in its capacity, so even though there is interference, the model cannot fully exploit cleaner updates. At 128 channels, the backbone is expressive enough that separating conflicting tasks in time and space actually pays off.
> At 256 channels, performance improves again for both methods, but the gap shrinks.
> To recap our earlier responses, as model capacity grows, many conflicts are resolved by the large available representation, and SON-GOKU approaches behavior like joint training while still having a small benefit when there are remaining conflicts.
>
> | R | $64^2$ workload (ms/step) | $128^2$ workload (ms/step) | $256^2$ workload (ms/step) |
> | --- | --- | --- | --- |
> | 8  | 146.445 | 275.696 | 370.202 |
> | 16 | 134.335 | 210.775 | 280.735 |
> | 32 | 128.320 | 168.365 | 221.965 |
> | 64 | 120.074 | 153.747 | 193.411 |
> | Baseline ($R \; = \; \infty$) | 111.514 | 128.948 | 149.277 |
>
> As we can see in the above table (also shown in Figure 2 of Appendix R.2), SON-GOKU's runtime decreases as $R$ increases, and for $R = 64$ it starts to become nearly indistinguishable from the joint training baseline across all backbone widths. This supports our claim that SON-GOKU can track task interference while adding only negligible overhead, even on larger backbones.
>
> We provide expanded commentary that discusses these experiments in more detail in Appendix R.

---

> ### Author Response · Authors · 2025-12-04
> **Update on Experiments for Reviewer u9zP, Weakness 2**
>
> ## **Weakness 2**
> > The paper reports algorithmic time complexity, but it does not sufficiently compare memory usage against other methods.
>
> In our past responses, we have already discussed the new Appendix N "Space Complexity and Memory Usage" (which provided extensive theory, reasoning, and analysis regarding SON-GOKU's memory efficiency) in our previous responses. Now that our experiments have finished running, we have added Appendix N.4, which discusses experimental validation of our aforementioned theoretical analysis.
>
> We now additionally report throughput and memory usage for SON-GOKU and several baselines on Taskonomy Tiny with a U-Net backbone model.  The table below (also included as Table 7 in Appendix N.4) shows throughput and memory usage at each step, averaged over five independent trials:
>
> | Method   | Throughput (imgs/s) ↑ | Peak mem. (MB) ↓ | Reserved mem. (MB) ↓ |
> |----------|------------------------|-------------------|-----------------------|
> | SON-GOKU | 68.5 ± 1.1            | 1959 ± 47         | 2332 ± 62             |
> | FAMO     | 47.6 ± 2.6            | 4740 ± 188        | 5152 ± 143            |
> | CAGrad   | 42.1 ± 1.9            | 2008 ± 33         | 2284 ± 97             |
> | AdaTask  | 41.6 ± 3.4            | 5108 ± 121        | 5544 ± 214            |
> | PCGrad   | 40.0 ± 1.5            | 5108 ± 173        | 5614 ± 129            |
> | MGDA     | 36.4 ± 4.2            | 5065 ± 142        | 5308 ± 201            |
> | GradNorm | 25.5 ± 2.1            | 5034 ± 77         | 5542 ± 178            |
>
> This table shows that SON-GOKU attains the highest throughput (68.5 imgs/s) while also using the lowest peak GPU memory (gigabytes smaller than most other approaches). CAGrad is the only method with slightly lower reserved memory, but it is substantially slower than SON-GOKU. Taken together with the theoretical discussion in Appendix N, this empirical comparison directly addresses your concern by demonstrating that SON-GOKU offers a more favorable tradeoff (in terms of time and memory) than existing approaches.

---

### Official Review · Reviewer_hC6i · 2025-11-06

**Soundness:** 3
**Presentation:** 2
**Contribution:** 2
**Rating:** 4
**Confidence:** 3

**Summary:**

This paper tackles the gradient-conflict problem in multi-task learning that slows convergence and degrades final accuracy.  It proposes SON-GOKU, a scheduler that quantifies pairwise gradient interference, builds an interference graph, and uses greedy graph coloring to split tasks into harmony groups that are updated one at a time.  Extensive experiments on six datasets show consistent gains over strong baselines and state-of-the-art optimizers, while novel theory proves descent guarantees and convergence under the proposed grouping strategy.

**Strengths:**

1. This paper pinpoints the phenomenon of gradient interference that slows convergence in multi-task learning, and introduces an effective grouping strategy to counter it.
2. Extensive experiments across diverse benchmarks convincingly validate the proposed method, showing consistent and non-trivial improvements over strong baselines.
3. The work delivers a crisp message: task grouping is vital for mitigating gradient conflict.

**Weaknesses:**

1. The paper mentions runtime overhead but provides no measured training time or inference latency.
2. The ablation studies only include one-shot and single-step conflict estimation, while the key component (e.g., graph building) is lacking.
3. Ablation results are split across Sections 6.4 and 7.2 without a clear organizing principle, which fragments the narrative and complicates comparison.

**Questions:**

Please check the weakness

---

> ### Author Response · Authors · 2025-11-27
> **Response to Reviewer hC6i (1/9)**
>
> We thank Reviewer hC6i for their careful and candid assessment of our work. We are glad that you appreciated our **rigorous empirical validation** across very **diverse benchmarks** and that you found our **grouping strategy** to be **effective**. We appreciate your constructive criticism and feedback surrounding several components of our paper.
>
> In the comments that follow, we respond to **each of your points** in turn: **(W1)** measured training time and inference latency; **(W2)** additional ablations and clarification around the graph-building component; and **(W3)** the organization and presentation of ablation results. We also provide a **brief clarification regarding the scope of our paper** and its focus to clear potential confusion you may have had, as we believe you may have **misunderstood our paper** and its **positioning in the field.**

---

> ### Author Response · Authors · 2025-11-28
> **Response to Reviewer hC6i, Weakness 1 (2/9)**
>
> ## **W1: No Measured Training Time or Inference Latency**
> SON-GOKU is a *training-time* framework. It periodically computes similarities between tasks and forms non-conflict groups to schedule updates. At inference, the learned backbone and task heads are used exactly as in the baseline model, with no grouping or refresh operations. Therefore, SON-GOKU **does not change inference latency or memory**, they are **fully determined** by the *underlying model architecture* and *batch size*. We will include a table based on experiments confirming that latency and peak memory at test time are indistinguishable (within noise) from the baseline model without MTL.
>
>
>
> **Training time (theory, as already in the paper).** We *explicitly* analyze runtime in the paper. Our analysis separates the normal training step from an occasional refresh that estimates task affinity and builds the graph. A standard step has the same complexity as the baseline. A refresh has a **one-off cost**, and it occurs every $R$ steps, so the **amortized overhead** is $\\text{refresh\\_time}/R$. This is *exactly* why we present low cost refresh options in the paper. They reduce $\\text{refresh\\_time}$ while preserving overhead. To put it formally, $\\text{refresh\\_time}$ can be defined as:
>
> $$
> T_{\text{EMA}} + T_{\text{cosine}} + T_{\text{threshold}} + T_{\text{color}} = \Theta(Kd) + \Theta(K^{2} d) + \Theta(K^{2}) + \Theta(K^{2}) = \Theta(K^{2} d)
> $$
>
> where $K$ is the number of tasks, $d$ is the dimension of the shared parameters, $T_{\text{EMA}}$ is the cost of refreshing the EMA gradient matrix, $T_{\text{cosine}}$ is the cost of forming the $K \times K$ cosine matrix, $T_{\text{threshold}}$ is the cost of thresholding that matrix to obtain the adjacency, and $T_{\text{color}}$ is the cost of greedy graph coloring. **As shown in Appendix D**, the $\Theta(K^2 d)$ term from the Gram matrix computation dominates the others under the standard RAM model. **Section 4.5.** discusses this amortized overhead in further detail. In short, the total training time is the *baseline training time* plus a *small controllable overhead* that scales *inversely with $R$*.
>
>
>
> **Section 7.4** provides a comparison of SON-GOKU’s overhead against the overhead that other multi-task learning frameworks require. We can see that SON-GOKU’s wall-clock time grows *modestly* with $K$ and remains **far below heavy methods** that handle conflicts on a step-by-step basis like PCGrad, FairGrad, and Nash-MTL, while still modeling task interference more appropriately than approaches like FAMO.
>
>
>
> In **Appendix D.5,** we discuss approaches that can be taken to reduce $\\text{refresh\\_time}$ while preserving fidelity. This includes **(i)** using random projections to compress the EMA gradient matrix into a much lower dimensional space before computing cosines, **(ii)** using a deterministic sketch (Frequent Directions) to approximate the Gram matrix in a smaller subspace, and **(iii)** computing cosines only for a carefully chosen subset of task pairs via edge sampling with adaptive refinement, rather than evaluating all $K^2$ pairs. We also report measured wall clock times across methods and tasks in Table 2 (shown below), which compares SON-GOKU to strong baselines (e.g., PCGrad, FAMO) as $K$, the number of tasks in the multi-task problem, scales. We discuss the methodology used to derive the data in the table in detail in **Appendix L.1.**

---

> ### Author Response · Authors · 2025-11-28
> **Response to Reviewer hC6i, Weakness 1 (3/9)**
>
> | Method (R if applicable) | K = 3 | K = 6 | K = 16 | K = 40 |
> |---------------------------------|-----------------------|------------------------|-------------------------|--------------------------|
> | Uniform | 0.2656 ± 0.1201 | 0.3240 ± 0.0629 | 0.3798 ± 0.1050 | 0.4054 ± 0.1190 |
> | GradNorm | 5.4714 ± 0.7137 | 5.1201 ± 0.6112 | 4.9042 ± 0.5869 | 4.7372 ± 0.9286 |
> | AdaTask | 2.1816 ± 0.0934 | 2.1032 ± 0.1012 | 2.2853 ± 0.0718 | 2.2278 ± 0.1370 |
> | PCGrad | 3.6212 ± 0.3517 | 23.1266 ± 0.8773 | 176.7566 ± 2.8171 | 1127.1337 ± 34.2603 |
> | MGDA | 97.1081 ± 5.4645 | 121.4371 ± 9.0923 | 132.4913 ± 3.1752 | 134.0878 ± 2.2621 |
> | FAMO | 2.0725 ± 0.2073 | 1.9980 ± 0.1998 | 2.1710 ± 0.2171 | 2.1164 ± 0.2116 |
> | FairGrad | 3.8020 ± 0.5703 | 15.2079 ± 2.2812 | 108.1450 ± 16.2218 | 675.9065 ± 101.3860 |
> | Nash-MTL | 5.7030 ± 1.1406 | 22.8118 ± 4.5624 | 162.2176 ± 32.4435 | 1013.8598 ± 202.7720 |
> | SON-GOKU (R = 32) | 1.9896 ± 0.3651 | 3.3202 ± 0.5745 | 6.0897 ± 0.9425 | 12.1432 ± 1.2044 |
> | SON-GOKU + AdaTask (R = 32) | 3.7718 ± 0.9654 | 5.0511 ± 0.6531 | 7.5903 ± 1.1920 | 14.5182 ± 2.0660 |
> | SON-GOKU + GradNorm (R = 32) | 7.0202 ± 1.0711 | 8.1661 ± 0.9355 | 10.7227 ± 2.2088 | 16.5760 ± 1.8418 |
> | SON-GOKU + PCGrad (R = 32) | 1.9834 ± 0.3586 | 3.4971 ± 0.3840 | 6.1395 ± 0.9425 | 10.9097 ± 1.5263 |
>
> We believe that the existing experiments and theoretical analysis is *sufficient to demonstrate* that SON-GOKU’s **overhead is modest and well controlled** in realistic multi-task settings. To further address your concern, however, we are currently running rigorous large-scale experiments, which are explained in the following subsection.
>
> ### **Plan for added evidence**
>
> While the paper *already* provides **(i) theory** surrounding the overhead and **(ii)** **measured training times** in a study on scaling, we believe that reporting training speed on the entire training process (i.e., how long it takes to train a model on an entire dataset) may help to further **address** your concern. Therefore, we will measure amortized training speed at a step-by-step level on a very large and popular benchmark, Taskonomy, to verify that SON-GOKU remains practical in terms of overhead even at a larger scale. We will also test how training speed varies with the number of tasks that are used and refresh rate $R$. As we mentioned earlier, we will also include experiments confirming that latency and peak memory at test time with are indistinguishable (within noise) from the baseline model without MTL. As it has already been theoretically proven, this wll be done by scaling to higher $R$ (lower refresh rate $R / 1$) to demonstrate that having less refreshes approaches non-MTL speed, which is the conditions that will exist at inference-time. These additions will appear in the appendix, alongside the existing analysis and timing table already in the paper. We are currently running experiments and expect them to conclude over the next few days, after which we will promptly provide our results.

---

> ### Author Response · Authors · 2025-11-28
> **Response to Reviewer hC6i, Weakness 2 (4/9)**
>
> ## **W2: Ablation Study for Graph Construction**
>
> This is an important observation that you have pointed out. Due to compute constraints, it was, at the time of writing the paper, infeasible to run more than two ablation experiments. Therefore, we had to **prioritize which components of SON-GOKU to isolate** as part of the ablations.
>
> Your main concern seems to be that we did not conduct ablations for graph construction. However, the ablations we *already* tested **actually directly impact** how the **graphs are constructed** and how **tasks are grouped** throughout training. Therefore, we assume that you meant ablations that test the *specific graph building choices* (e.g., how conflict is estimated, how often the graph is refreshed, and how colors are assigned), beyond the variants we have already evaluated.
>
>
>
> In our series of comments, we first provide rationale for prioritizing our existing ablations, then discuss why our graph construction and coloring approach is theoretically well-founded, and then discuss new graph-related ablations we are testing to address your concerns.
>
>
>
> ### **Why we prioritized one-shot and single-step first**
> We chose the *Static One-Shot* and *Single-Step* ablations because they **directly test** the two design choices that our theory shows to be **the most consequential for correctness and stability**. Specifically:
>
>
>
> 1. **Dynamic recoloring versus frozen groups.** In SON-GOKU we rebuild the task-conflict graph every $R$ steps from gradient cosine similarities and then recolor it using the Welsh-Powell algorithm. Tasks with the same color are trained together. The one-shot ablation sets $R = \infty$, meaning we build the graph once at step 0 and never update it. Because gradient directions can drift during training, pairs of tasks that are initially compatible can later become conflicting (and vice versa). A frozen coloring therefore goes stale, hurting both stability and data efficiency. It may keep co-training tasks that now interfere, or separate tasks that have become compatible. **Periodic recoloring avoids this** by re-estimating edges with current signals and re-partitioning accordingly.
>
>    ⠀
>
>    We prioritized this ablation first because **it isolates the adaptivity of the scheduler** without introducing extra parameters. The results of our ablation study demonstrated that as training progresses, **task relations do drift in practice**, creating the issues we discussed previously. In contrast, most graph construction choices already come with guidance **based on our theoretical analysis** (see Section *4* for construction, Section *5.4* for recovery, and Section *5.5* for how $\tau$ impacts scheduling length).
>
>    ⠀
>
>
>
> 2. **Conflict estimation based on averaged history vs. batch-level.** In SON-GOKU, we estimate task conflict by averaging gradient cosines over a short history. At step $t$, we compute $s_{ij}(t)=\cos  \big(g_i(t),g_j(t)\big)$ and maintain an Exponential Moving Average:
>
>    $$\\hat{s}\_{ij}(t)=(1-\\beta)\,\\hat{s}\_{ij}(t-1)+\\beta\\,s\_{ij}(t),$$
>
>    with effective window $H\approx  1/\beta$. We add an edge $i\leftrightarrow j$ when $\hat{s}_{ij}(t)<-\tau$, then color the graph and schedule groups. The single-step ablation disables averaging ($H=1$, $\beta=1$), edges are built from the current mini-batch only.
>
>    ⠀
>
>    Estimates on a batch-by-batch basis tend to have **high variance** (see, e.g., Bottou, 2010; Mandt et al., 2017), so small stochastic changes in $s_{ij}(t)$ can flip edges across the $-\tau$ threshold, causing edge churn and jittery colorings (groups change even when underlying relations have not). This **destabilizes the schedule** and can **lengthen it unnecessarily** (more colors) or constrain newly conflicting tasks. This ablation therefore isolates **whether temporal smoothing is necessary for reliable graph construction** (see Section *7.2* for results, *7.2.1* for interpretation of results, Section *5.4* for further analysis, and Appendix *B.4-B.5* for exact bounds and more thorough proofs).

---

> ### Author Response · Authors · 2025-11-28
> **Response to Reviewer hC6i, Weakness 2 (5/9)**
>
> ### **Why Our Graph Construction and Coloring Strategy Is Well Founded**
>
> Now, we want to discuss *why* we adopt this specific graph construction and coloring strategy. This is already covered extensively, particularly in **Sections 4.1-4.4** (gradient interference estimation, conflict graph construction, greedy coloring, and schedule generation) and **Sections 5.1–5.5** (descent properties, convergence guarantees, graph recovery, and scheduling properties), as well as **Appendices B** (exact recovery of the population conflict graph and task partition), **D** (computational cost and implementation details), and **I** (full convergence proofs). However, for the sake of clarity and convenience, we will provide an abridged high level overview here as well.
>
>
> **Why we use a conflict graph.** First, we would like to clarify why we use a conflict graph at all. Our starting point is the standard observation (clarified in *Section 3* and *Section 4*) that multi-task SGD can be viewed as vector valued optimization, and that pairwise gradient inner products determine whether joint updates are cooperative or interfering. If $\langle g_i, g_j \rangle < 0$, moving along $g_i$ increases $L_j$ to first order. If all inner products are non-negative, any positive combination is a descent direction for all tasks. This **motivates representing tasks as nodes** and **putting an edge between two tasks** exactly when our estimate of their cosine similarity indicates **strong persistent conflict** (*Section 4.1*), then using this graph to decide which tasks should and should not be updated together.
>
>
>
> Intuitively, the graph is a **compressed summary of what tasks conflict with each other in gradient space**. Instead of solving a multi-objective subproblem every step, we turn this into a **combinatorial scheduling problem** where we determine which subsets of tasks should be co-trained in the same update.
>
>
>
> **Why we construct graphs in the way we do.** In *Sections 4 and 5.4*, we define a conflict score based on cosine similarity of gradients, and we maintain an Exponential Moving Average (EMA) $\hat{s}_{ij}(t)$ over time. Under standard assumptions, we can prove the EMA concentrates around the true mean similarity, and thresholding at $-\tau$ recovers the true conflict graph with high probability as training progresses (*Appendices B.4 to B.5*). *Section 5.5* then analyzes how the choice of $\tau$ controls edge density and therefore the length of the schedule. We find that larger $\tau$ yields **more edges and more colors** (longer schedule but less interference), while smaller $\tau$ yields **fewer edges and fewer colors** (shorter schedule but more interference).
>
>
>
> If this theoretical justification for graph construction is confusing, it may be easier to just focus on the intuition behind the theory. Essentially, **cosine focuses on direction**, which is what matters for whether gradients help or hurt each other, and it is **invariant to the absolute scale of each task’s loss** (*Section 4.1*). The EMA **suppresses mini batch** noise and tracks **persistent relationships between tasks** as they evolve over time (*Section 5.4*). Finally, a global threshold $\hat{s}_{ij}(t) < \tau$ is a simple way to only keep edges for pairs that are strongly enough antagonistic which yields a **sparse informative graph** and allows us to **analyze recovery and scheduling length** (*Section 5.5*)
>
>
>
> The existing single-step and one-shot ablations actually support this graph construction. Removing EMA or freezing the graph makes the schedule **less stable and hurts performance**, as supported by the *theory* (*Section 7.2*).

---

> ### Author Response · Authors · 2025-11-28
> **Response to Reviewer hC6i, Weakness 2 (6/9)**
>
> **Why we color graphs the way we do.** Once we have the conflict graph, we color it so that **no edge appears inside a color class** (*Section 4.2*). Each color is an *independent set of tasks* whose gradients are not strongly conflicting according to our threshold. Training then proceeds by cycling through colors and updating one color (i.e., one compatible group of tasks) at a time. Under mild conditions, the per color updates remain descent directions for the tasks in that group, and the **overall algorithm retains the convergence guarantees of SGD** while **reducing harmful interference** (*Sections 5.1-5.3*). We do not need the optimal chromatic number, **any proper coloring** that respects the graph suffices for the theoretical guarantees. Overall, coloring is a *convenient way* to impose the rule that **conflicting tasks should not be trained together in the same step**. Intuitively, each color class is a set of tasks we regard as **safe to co-train**, conflicts only appear between colors. This turns the *continuous high-dimensional* gradient geometry into a **schedule over discrete groups**, helping to address one of the **key concerns** we bring up in Section 1:
>
> > many [multi-task learning approaches] rely on dense pairwise affinities that grow noisy and costly as the number of tasks rises.
>
> The number of colors is then an **interpretable quantity telling us how many sequential groups SON-GOKU needs** to **avoid the conflicts** that the graph detected.
>
>
>
> **Why we use this particular combination and not alternatives.** We chose this pipeline of graph construction and coloring because it is the simplest approach that simultaneously aligns with the theory and works well in practice. On the construction side, our use of smoothed cosine similarities and a global threshold comes directly from the multi-objective view in *Section 4* and the graph recovery analysis in *Section 5.4*. Cosine and inner products are the **natural theoretical objects** for describing task interference, exponential averaging makes our estimates regarding task conflicts **stable over time**, and applying a global threshold gives us a graph that is **sparse and interpretable**. This is in contrast to alternatives like multi-objective solvers operating on a step-by-step basis, which are **harder to scale** in the number of tasks, or purely local graph rules that are **more difficult to analyze** and can **misallocate edges** when conflicts are heterogeneous across tasks.
>
>
>
> On the grouping side, graph coloring helps us directly create a schedule over task groups from pairwise conflict information. It enforces, by construction, that no "strong conflict" (as defined by our threshold) appears within any group, while still allowing us to train multiple tasks at once. Greedy coloring is inexpensive and nearly optimal on the sparse graphs we obtain, and our analysis in *Sections 5.1-5.3* show that any such proper coloring is sufficient to retain the convergence guarantees of SGD while reducing interference. Based on our analyses, alternative strategies (such as learning continuous weights over tasks at every step or trying to optimize the exact chromatic number) either **do not yield additional theoretical benefit** in our setting or **introduce significant extra computational cost and uncertainty**.
>
>
>
> Taken together, this **specific combination** is where our **theoretical requirements** (conflict detection, graph recovery, convergence guarantees) best intersect with **practical constraints**. The new ablations we introduce in the rebuttal are designed to test this design choice further by comparing against other plausible graph rules, but they are all evaluated within the same construction and graph coloring framework.

---

> ### Author Response · Authors · 2025-11-28
> **Response to Reviewer hC6i, Weakness 2 (7/9)**
>
> ### **Proposed Graph Ablations**
>
> While we strongly believe that the existing experimental results and in-depth theory provided in the paper, alongside the high-level commentary provided previously in this comment, are **more than enough to address concerns you may have**, we want to go one step further and provide additional targeted empirical evidence. To further address your concern directly, we are now adding **eight new ablations** that vary only the graph construction rule. This allows us to further analyze how much the particular way we construct the conflict graph matters, given the same conflict estimator and scheduler.
>
>
>
> Specifically, we compare four graph rules:
>
> 1. **Threshold (baseline).** We connect tasks $i$ and $j$ when the smoothed cosine $\hat{s}_{ij}(t)$ is below a global threshold $\tau(t)$. This is the rule used in the main paper and is the one analyzed in our recovery and scheduling theory (Section 4, Section 5.4-5.5). It prioritizes the most strongly conflicting pairs globally.
>
> 2. **kNN-symmetric.** For each task, we connect it to its $m$ most conflicting neighbors (those with the smallest $\hat{s}_{ij}(t)$), and then symmetrize the graph. This enforces a roughly fixed degree for each task and tests whether local degree control performs better or worse than our global cutoff when conflict is heterogeneous across tasks.
>
> 3. **Signed-only.** We connect two tasks only if $\hat{s}_{ij}(t) < 0$. This explores an extreme notion of conflict that only considers strictly negative cosine. It yields a much sparser graph and allows us to test whether ignoring mildly positive but still harmful interactions degrades performance.
>
> 4. **Quantile and percentile.** At each refresh, we set $\tau(t)$ to the $p$-th percentile of the current cosine values, so that we always connect roughly the worst $p%$ of pairs. This behaves like an adaptive threshold that tracks how the similarity distribution drifts over training, and provides a principled alternative to fixing $\tau$ by hand.
>
>
>
> We evaluate these four graph rules under two settings:
>
> 1. **Fixed $\tau$ schedule.** All rules share the same $\tau$ schedule that we use in the main paper, and we simply measure the resulting edge density and number of colors. This reveals the intrinsic bias of each rule. I.e., whether it tends to concentrate edges on the most antagonistic pairs (threshold, quantile), spread them more evenly across nodes (kNN), or ignore many moderate conflicts (signed-only). It also shows how these choices translate into different scheduling lengths (more or fewer colors) and different validation performance.
>
> 2. **Density-matched graphs.** Here we explicitly match the edge density (or expected degree) across rules at each refresh. Threshold and quantile rules are chosen to hit a target density, kNN uses $m$ set to achieve the same average degree, and signed-only is reported with its natural density as a comparison point. Under this approach, any performance differences cannot be attributed to more or fewer edges, but instead reflect which pairs each rule chooses to label as conflicting. This isolates the quality of the selection policy itself, given the same amount of edges.
>
>
>
> We are currently running these ablations. We will include the results and methodology in the main text promptly after experimentation has concluded to help give a clear comparison of our graph construction rule against its most natural alternatives.

---

> ### Author Response · Authors · 2025-11-28
> **Response to Reviewer hC6i, Weakness 3 (8/9)**
>
> ## **W3: Confusing Organization of Sections 6.4 and 7.2**
>
> Thank you for the helpful suggestion about the writing surrounding ablations being split. We’ve restructured the paper so all ablation-related content is now in a single place. Specifically, we have moved the ablation variant definitions (previously **Section 6.4**) into **Section 7.2**, alongside the results and interpretation. The paper PDF has been updated accordingly, and the **changes are highlighted in blue**.
>
>
>
> With these updates, readers no longer have to jump between sections when learning about the ablations. Again, we sincerely thank you for pointing this out, as the readability and presentation of our paper is highly important. A cleaner paper is easier to read and, therefore, more likely to support future research in the field.

---

> ### Author Response · Authors · 2025-11-28
> **Response to Reviewer hC6i, Clarifications Regarding Potential Misunderstandings (9/9)**
>
> ## **Further Clarifications and Explanations**
>
> It seems Reviewer hC6i may have **partially misunderstood** the **overall purpose of the paper** and therefore **interpreted it differently from how we intended**. Thus, we would like to **clarify the scope of the paper** and distinguish it from the existing literature. First, the phenomenon of gradient interference in multi-task learning (i.e., that conflicting task gradients can slow convergence and hurt final accuracy) has **already been identified** in previous work. Second, several families of methods have been proposed to cope with such interference. In our existing Related Work section (*Section 2*, *Appendix M*), we categorize the most important families as follows: Tuned Loss Weighting, Adaptive Loss Weighting, Gradient-Level Conflict Mitigation (gradient surgery), and Empirical Task Grouping. The family under which SON-GOKU falls, Empirical Task Grouping, gained popularity in the early 2020s following a NeurIPS 2021 paper that introduced "Task Affinity Groupings (TAG)", although there were papers already investigating the idea of grouping in the 2010s. By design, SON-GOKU does not claim the discovery of interference or the mere idea of grouping. Rather, **we aim to address three key issues** that **persist in existing grouping methods** (*as stated in Section 1*):
>
> 1) Many approaches rely on dense pairwise affinities that grow noisy and costly as the number of tasks increase (*see Section 1; Section 4.5; Appendix D.5*).
>
> 2) Many predetermine or rarely update groups, so they drift as task relations change throughout training (*see Section 1; Section 4.4; Appendix A*).
>
> 3) Quite a few use local heuristics that fail to enforce global compatibility or to specify how groups should rotate over time(*see Section 1; Section 4.3; Section 4.4; Section 5.5; Appendix G; Appendix H*).
>
>
> SON-GOKU is **built on the highly novel idea** of modelling conflicts between tasks, creating groups, and scheduling tasks using a graph-based approach. We focus on providing a "plug-and-play" framework that can be easily used to **complement existing approaches like AdaTask and PCGrad**. We also focus on providing **novel** and **in-depth theoretical analyses** that can be reused in future research.
>
>
>
> We believe our existing **Introduction** (*Section 1*), **Related Work** section (*Section 2*), and **commentary throughout the paper** make this distinction clear, but we agree that more could have been done to make it a prominent point. **We have revised Section 2** to further clarify any doubts and more explicitly direct readers to *Appendix M* if they are interested in learning more about the existing literature and how SON-GOKU is positioned in this field. For context, Appendix M ("Extended Related Work") provides a more thorough analysis and commentary of the recent work in this field as compared to our Related Work section, but we are unfortunately unable to make space for it in our paper’s main text due to the strict page limit.

---

> ### Author Response · Authors · 2025-11-28
> **Response to Reviewer hC6i, References**
>
> # **References**
>
>
>
> Ban, H., & Ji, K. (2024). Fair Resource Allocation in Multi-Task Learning. *ArXiv (Cornell University)*. https://doi.org/10.48550/arxiv.2402.15638
>
> Bottou, L. (2010). Large-Scale Machine Learning with Stochastic Gradient Descent. *Proceedings of COMPSTAT’2010*, 177–186. https://doi.org/10.1007/978-3-7908-2604-3\_16
>
> Chen, Z., Vijay Badrinarayanan, Lee, C.-Y., & Rabinovich, A. (2018). GradNorm: Gradient Normalization for Adaptive Loss Balancing in Deep Multitask Networks. *PMLR*, 794–803. https://proceedings.mlr.press/v80/chen18a
>
> Fifty, C., Amid, E., Zhao, Z., Yu, T., Anil, R., & Finn, C. (2021). Efficiently Identifying Task Groupings for Multi-Task Learning. *ArXiv (Cornell University)*, *34*. https://doi.org/10.48550/arxiv.2109.04617
>
> Ghashami, M., Liberty, E., Phillips, J. M., & Woodruff, D. P. (2016). Frequent Directions: Simple and Deterministic Matrix Sketching. *SIAM Journal on Computing*, *45*(5), 1762–1792. https://doi.org/10.1137/15m1009718
>
> Liu, B., Feng, Y., Stone, P., & Liu, Q. (2023). FAMO: Fast Adaptive Multitask Optimization. *Neural Information Processing Systems*.
>
> Liu, B., Liu, X., Jin, X., Stone, P., & Liu, Q. (2021). Conflict-Averse Gradient Descent for Multi-task Learning. *ArXiv (Cornell University)*. https://doi.org/10.48550/arxiv.2110.14048
>
> Mandt, S., Hoffman, M. D., & Blei, D. M. (2017). Stochastic Gradient Descent as Approximate Bayesian Inference. *Journal of Machine Learning Research*, *18*(134), 1–35. http://jmlr.org/papers/v18/17-214.html
>
> Navon, A., Shamsian, A., Achituve, I., Maron, H., Kawaguchi, K., Chechik, G., & Fetaya, E. (2022). Multi-Task Learning as a Bargaining Game. *ArXiv (Cornell University)*. https://doi.org/10.48550/arxiv.2202.01017
>
> Sener, O., & Koltun, V. (2018). Multi-Task Learning as Multi-Objective Optimization. *Neural Information Processing Systems*.
>
> Welsh, D. J. A. (1967). An upper bound for the chromatic number of a graph and its application to timetabling problems. *The Computer Journal*, *10*(1), 85–86. https://doi.org/10.1093/comjnl/10.1.85
>
> Yang, E., Pan, J., Wang, X., Huang, Y., Shen, L., Chen, X., Liu, X., Jiang, J., & Guo, G. (2023). AdaTask: A Task-Aware Adaptive Learning Rate Approach to Multi-Task Learning. *Proceedings of the ... AAAI Conference on Artificial Intelligence*, *37*(9), 10745–10753. https://doi.org/10.1609/aaai.v37i9.26275
>
> Yu, T., Kumar, S., Gupta, A., Levine, S., Hausman, K., & Finn, C. (2020). Gradient Surgery for Multi-Task Learning. *ArXiv (Cornell University)*. https://doi.org/10.48550/arxiv.2001.06782
>
> Zamir, A. R., Sax, A., Shen, W., Guibas, L., Malik, J., & Savarese, S. (2018). Taskonomy: Disentangling Task Transfer Learning. *2018 IEEE/CVF Conference on Computer Vision and Pattern Recognition*. https://doi.org/10.1109/cvpr.2018.00391

---

> ### Author Response · Authors · 2025-12-04
> **Update on Experiments for Reviewer hC6i (1/6)**
>
> We thank reviewer hC6i again for their careful review of our work. In the time since our original response, we have finished running **several additional experiments** and completed **additional theoretical analysis**, reasoning, and **commentary** in our paper to further address your concerns. When reading the revised paper, it may be helpful to note that text in the **color blue** was **written specifically to address your concerns**, though text in other colors may also be helpful to you. Also please keep in mind that we have already made numerous revisions to the paper even before experiments concluded, so not all changes made to address your concerns will be covered in this "Update on Experiments", those other changes are discussed in our previous responses.

---

> ### Author Response · Authors · 2025-12-04
> **Update on Experiments for Reviewer hC6i, Weakness 1 (2/6)**
>
> ## **Weakness 1**
> > "The paper mentions runtime overhead but provides no measured training time or inference latency."
>
> The paper now includes **additional experimentation**  and a **visualization** (see *Figure 2*) in Appendix R.2 further supporting our original reasoning that SON-GOKU **does not** actually add any **overhead** at **inference-time**.
>
> | R | $64^2$ workload (ms/step) | $128^2$ workload (ms/step) | $256^2$ workload (ms/step) |
> | --- | --- | --- | --- |
> | 8  | 146.445 | 275.696 | 370.202 |
> | 16 | 134.335 | 210.775 | 280.735 |
> | 32 | 128.320 | 168.365 | 221.965 |
> | 64 | 120.074 | 153.747 | 193.411 |
> | Baseline ($R \; = \; \infty$) | 111.514 | 128.948 | 149.277 |
>
> The experiments demonstrate that, as refresh interval $R$ approaches $\infty$ (and hence, refresh rate approaches $0$), SON-GOKU's amortized speed (at training time) approaches that of non-MTL joint training. This behavior can be expanded to inference-time speed as well, where there are no refresh steps, so SON-GOKU's speed will match that of any non-MTL baseline. The scheduler is inactive at inference, there are no refreshes or extra computations, and the forward graph is unchanged. This supports our concrete reasoning and analysis presented earlier in *Response (2/9)*.
>
> There are a few other **new** experiments that have recently concluded in the paper (alongside the tests that already existing in the original paper at Table 2 and Table 5) that also measure the speed of SON-GOKU, which may help to further address your concerns. These are Tables 3, 7, and 9, which are provided below.
>
> ### Table 3 (New)
> | Family            | Variant                      | Elapsed (s) $\downarrow$   | Imgs/s $\uparrow$        | Refresh (ms) $\downarrow$   | Depth Eucl. RMSE $\downarrow$      | Depth Eucl. MAE $\downarrow$       | Normal mean (deg) $\downarrow$      | Reshading MAE $\downarrow$        |
> |-------------------|------------------------------|-----------------------------|---------------------------|------------------------------|-------------------------------------|-------------------------------------|-------------------------------------|------------------------------------|
> | Baseline          | FD (conservative)            | $76.37 \pm 1.84$           | $50.28 \pm 1.21$         | $488.94 \pm 27.35$           | $\mathbf{25.58} \pm 0.92$           | $\mathbf{7.98} \pm 0.41$            | $\mathbf{56.63} \pm 1.87$           | $\mathbf{0.238} \pm 0.018$         |
> | Random proj.      | 128 random dim.              | $72.32 \pm 2.76$           | $53.10 \pm 1.95$         | $366.70 \pm 34.12$           | $26.19 \pm 1.37$                    | $9.05 \pm 0.63$                     | $70.37 \pm 3.91$                    | $0.334 \pm 0.039$                  |
> | Freq. Directions  | 128 FD width $r$             | $73.85 \pm 1.57$           | $52.00 \pm 1.33$         | $391.15 \pm 29.44$           | $27.49 \pm 1.68$                    | $9.82 \pm 0.71$                     | $70.97 \pm 4.22$                    | $0.490 \pm 0.065$                  |
> | Freq. Directions  | 256 FD width $r$             | $75.11 \pm 1.23$           | $51.13 \pm 1.09$         | $440.05 \pm 23.08$           | $26.34 \pm 1.05$                    | $8.72 \pm 0.52$                     | $62.37 \pm 2.98$                    | $0.339 \pm 0.033$                  |
> | Edge sampling     | 25\% sampling rate           | $72.38 \pm 3.41$           | $53.05 \pm 2.37$         | $342.26 \pm 48.73$           | $126.01 \pm 18.92$                  | $49.50 \pm 7.31$                    | $77.48 \pm 6.85$                    | $5.959 \pm 0.821$                  |
> | Incremental Gram  | $1\text{e-}3$ threshold $\epsilon$ | $\mathbf{70.07} \pm 2.04$ | $\mathbf{54.80} \pm 2.11$ | $\mathbf{293.36} \pm 31.29$ | $34.41 \pm 4.67$                    | $15.36 \pm 1.88$                    | $93.86 \pm 9.73$                    | $0.621 \pm 0.079$                  |
>
> This table reports runtime and validation metrics for different SON-GOKU scheduling variants. It provides measured tradeoffs between computational efficiency and accuracy across the scheduler designs (including SON-GOKU itself) which we hope helps to address your concern as these measurements were made while training on the Taskonomy dataset.

---

> ### Author Response · Authors · 2025-12-04
> **Update on Experiments for Reviewer hC6i, Weakness 1 (3/6)**
>
> ### Table 5 (Pre-Existing)
> **$R = 4$ (all methods)**
>
> | Method                 | $K=3$                 | $K=6$                 | $K=16$                 | $K=40$                  |
> |------------------------|-----------------------|-----------------------|------------------------|-------------------------|
> | Uniform                | $0.2656 \pm 0.1201$  | $0.3240 \pm 0.0629$  | $0.3798 \pm 0.1050$   | $0.4054 \pm 0.1190$    |
> | GradNorm               | $5.4714 \pm 0.7137$  | $5.1201 \pm 0.6112$  | $4.9042 \pm 0.5869$   | $4.7372 \pm 0.9286$    |
> | MGDA                   | $97.1081 \pm 5.4645$ | $121.4371 \pm 9.0923$| $132.4913 \pm 3.1752$ | $134.0878 \pm 2.2621$  |
> | PCGrad                 | $3.6212 \pm 0.3517$  | $23.1266 \pm 0.8773$ | $176.7566 \pm 2.8171$ | $1127.1337 \pm 34.2603$|
> | CAGrad                 | $102.8651 \pm 18.3422$ | $136.1034 \pm 2.4218$ | $134.3585 \pm 4.0791$ | $132.7034 \pm 1.2412$  |
> | AdaTask                | $2.1816 \pm 0.0934$  | $2.1032 \pm 0.1012$  | $2.2853 \pm 0.0718$   | $2.2278 \pm 0.1370$    |
> | FAMO                   | $2.0725 \pm 0.2073$  | $1.9980 \pm 0.1998$  | $2.1710 \pm 0.2171$   | $2.1164 \pm 0.2116$    |
> | FairGrad               | $3.8020 \pm 0.5703$  | $15.2079 \pm 2.2812$ | $108.1450 \pm 16.2218$| $675.9065 \pm 101.3860$|
> | Nash-MTL               | $5.7030 \pm 1.1406$  | $22.8118 \pm 4.5624$ | $162.2176 \pm 32.4435$| $1013.8598 \pm 202.7720$|
> | SON-GOKU               | $2.0904 \pm 0.3506$  | $3.6770 \pm 0.4974$  | $6.3225 \pm 0.7895$   | $14.3280 \pm 1.4073$   |
> | SON-GOKU $+$ AdaTask   | $4.1011 \pm 0.4174$  | $5.2126 \pm 0.6066$  | $7.6798 \pm 0.7107$   | $14.7528 \pm 1.8671$   |
> | SON-GOKU $+$ GradNorm  | $7.3223 \pm 0.4994$  | $8.5898 \pm 0.8203$  | $12.1065 \pm 2.5850$  | $16.8329 \pm 1.9803$   |
> | SON-GOKU $+$ PCGrad    | $2.3489 \pm 0.3258$  | $3.5925 \pm 0.4100$  | $6.1549 \pm 0.8461$   | $12.5729 \pm 1.2657$   |
>
>
> **$R = 32$ (scheduler-based approaches)**
>
> | Method                 | $K=3$                 | $K=6$                 | $K=16$                 | $K=40$                 |
> |------------------------|-----------------------|-----------------------|------------------------|------------------------|
> | SON-GOKU               | $1.9896 \pm 0.3651$  | $3.3202 \pm 0.5745$  | $6.0897 \pm 0.9425$   | $12.1432 \pm 1.2044$  |
> | SON-GOKU $+$ AdaTask   | $3.7718 \pm 0.9654$  | $5.0511 \pm 0.6531$  | $7.5903 \pm 1.1920$   | $14.5182 \pm 2.0660$  |
> | SON-GOKU $+$ GradNorm  | $7.0202 \pm 1.0711$  | $8.1661 \pm 0.9355$  | $10.7227 \pm 2.2088$  | $16.5760 \pm 1.8418$  |
> | SON-GOKU $+$ PCGrad    | $1.9834 \pm 0.3586$  | $3.4971 \pm 0.3840$  | $6.1395 \pm 0.9425$   | $10.9097 \pm 1.5263$  |
>
>
> **$R = 256$ (scheduler-based approaches)**
>
> | Method                 | $K=3$                 | $K=6$                 | $K=16$                 | $K=40$                 |
> |------------------------|-----------------------|-----------------------|------------------------|------------------------|
> | SON-GOKU               | $1.7593 \pm 0.2280$  | $3.0024 \pm 0.3942$  | $4.8411 \pm 0.7302$   | $11.4162 \pm 1.6076$  |
> | SON-GOKU $+$ AdaTask   | $3.7224 \pm 0.2696$  | $4.4548 \pm 0.5837$  | $7.5276 \pm 0.6230$   | $13.0608 \pm 3.2925$  |
> | SON-GOKU $+$ GradNorm  | $6.0221 \pm 1.0418$  | $7.8659 \pm 0.7917$  | $9.5029 \pm 1.2168$   | $15.6860 \pm 2.3680$  |
> | SON-GOKU $+$ PCGrad    | $1.6776 \pm 0.4104$  | $3.0189 \pm 0.7854$  | $5.9893 \pm 1.3797$   | $7.1915 \pm 0.2021$   |
>
>
> The series of tables presents wall-clock time across different $K$, numbers of tasks, and scheduler refresh rates $R$. The results presented are from very rigorous evaluations of the speed of SON-GOKU against other methods, so we hope these results help to address your concern.

---

> ### Author Response · Authors · 2025-12-04
> **Update on Experiments for Reviewer hC6i, Weakness 1 (4/6)**
>
> ### Table 7 (New)
>
> | Method    | Throughput (imgs/s) $\uparrow$ | Peak mem. (MB) $\downarrow$ | Reserved mem. (MB) $\downarrow$ |
> |----------|---------------------------------|------------------------------|----------------------------------|
> | SON-GOKU | $\mathbf{68.5} \pm 1.1$         | $\mathbf{1959} \pm 47$       | $2332 \pm 62$                    |
> | FAMO     | $47.6 \pm 2.6$                  | $4740 \pm 188$               | $5152 \pm 143$                   |
> | CAGrad   | $42.1 \pm 1.9$                  | $2008 \pm 33$                | $\mathbf{2284} \pm 97$           |
> | AdaTask  | $41.6 \pm 3.4$                  | $5108 \pm 121$               | $5544 \pm 214$                   |
> | PCGrad   | $40.0 \pm 1.5$                  | $5108 \pm 173$               | $5614 \pm 129$                   |
> | MGDA     | $36.4 \pm 4.2$                  | $5065 \pm 142$               | $5308 \pm 201$                   |
> | GradNorm | $25.5 \pm 2.1$                  | $5034 \pm 77$                | $5542 \pm 178$                   |
>
> This table summarizes experiments on **throughput** and **memory usage** of SON-GOKU versus other baseline and state-of-the-art approaches. It is discussed in further detail in **Appendix N**. We hope this further addresses your concern regarding a lack of measured speed and training time.
>
> ### Table 9 (New)
> | Method                 | Amortized ms / step $\downarrow$ |
> |------------------------|-----------------------------------|
> | No Refresh (Baseline)  | $\mathbf{149.28} \pm 1.63$       |
> | SON-GOKU (cosine)      | $193.41 \pm 6.27$                |
> | SON-GOKU + TAG         | $255.29 \pm 17.84$               |
>
> This table reports amortized runtime per training step (in milliseconds) for the base SON-GOKU and SON-GOKU with TAG compared against a baseline that doesn't use refreshes, averaged over 10 trials. It showcases the additional computational overhead introduced by cosine-based SON-GOKU and further by the TAG extension. We hope this further addresses your concern regarding a lack of measured speed and training time, as these measurements were made while training on the Taskonomy dataset.

---

> ### Author Response · Authors · 2025-12-04
> **Update on Experiments for Reviewer hC6i, Weakness 2 (5/6)**
>
> ## Weakness 2
> > "The ablation studies only include one-shot and single-step conflict estimation, while the key component (e.g., graph building) is lacking."
>
> | Model                           | CIFAR-10 Acc. (\%) $\uparrow$ | AV-MNIST Acc. (\%) $\uparrow$ | MM-IMDb Acc. (\%) $\uparrow$ | F&B Acc. (\%) $\uparrow$ | F&B MAE $\downarrow$      | HEALTH Acc. (\%) $\uparrow$ | HEALTH MAE $\downarrow$     | NYUv2 Angle Error $\downarrow$ | NYUv2 Seg. MIOU $\uparrow$ | NYUv2 Depth RMSE $\downarrow$ |
> |---------------------------------|-------------------------------|--------------------------------|------------------------------|--------------------------|----------------------------|------------------------------|------------------------------|---------------------------------|------------------------------|--------------------------------|
> | Static One-Shot                 | $61 \pm 2.0$                  | $66 \pm 1.1$                   | $58 \pm 2.6$                 | $48 \pm 2.3$             | $0.56 \pm 0.027$          | $54 \pm 2.1$                 | $0.51 \pm 0.025$            | $20.5 \pm 0.25$                 | $0.071 \pm 0.004$           | $0.65 \pm 0.016$               |
> | Single-Step                     | $40 \pm 4.2$                  | $59 \pm 2.4$                   | $20 \pm 5.4$                 | $42 \pm 3.9$             | $0.60 \pm 0.041$          | $47 \pm 3.5$                 | $0.55 \pm 0.034$            | $26.4 \pm 0.55$                 | $0.042 \pm 0.006$           | $0.81 \pm 0.029$               |
> | **SON-GOKU (Threshold, DM)**    | $63$                          | $68$                            | $59$                         | $49$                     | $0.55$                    | $56$                          | $0.51$                      | $20.6$                           | $0.071$                      | $0.61$                          |
> | **SON-GOKU (kNN-Symm.)**        | $60$                          | $65$                            | $55$                         | $46$                     | $0.57$                    | $52$                          | $0.53$                      | $22.1$                           | $0.066$                      | $0.70$                          |
> | **SON-GOKU (kNN-Symm., DM)**    | $61$                          | $66$                            | $57$                         | $47$                     | $0.56$                    | $54$                          | $0.52$                      | $21.4$                           | $0.068$                      | $0.66$                          |
> | **SON-GOKU (Signed-only)**      | $56$                          | $63$                            | $52$                         | $43$                     | $0.60$                    | $50$                          | $0.56$                      | $24.0$                           | $0.053$                      | $0.76$                          |
> | **SON-GOKU (Signed-only, DM)**  | $58$                          | $64$                            | $54$                         | $45$                     | $0.59$                    | $52$                          | $0.54$                      | $23.0$                           | $0.056$                      | $0.73$                          |
> | **SON-GOKU (Quantile)**         | $64$                          | $68$                            | $60$                         | $50$                     | $0.54$                    | $57$                          | $0.50$                      | $20.3$                           | $0.072$                      | $0.60$                          |
> | **SON-GOKU (Quantile, DM)**     | $65$                          | $69$                            | $61$                         | $51$                     | $0.53$                    | $58$                          | $0.50$                      | $20.0$                           | $0.072$                      | $\mathbf{0.59}$                 |
> | SON-GOKU + AdaTask              | $\mathbf{67} \pm 1.2$         | $\mathbf{71} \pm 0.9$          | $\mathbf{63} \pm 1.6$        | $52 \pm 1.7$             | $0.53 \pm 0.021$          | $59 \pm 1.8$                 | $0.48 \pm 0.017$            | $20.1 \pm 0.20$                 | $0.068 \pm 0.004$           | $0.67 \pm 0.013$               |
> | SON-GOKU                        | $65 \pm 1.5$                  | $69 \pm 1.0$                   | $61 \pm 1.8$                 | $51 \pm 1.9$             | $0.53 \pm 0.023$          | $58 \pm 1.7$                 | $0.50 \pm 0.018$            | $19.8 \pm 0.20$                 | $0.073 \pm 0.004$           | $\mathbf{0.59} \pm 0.012$       |
>
> We have now completed **extensive experimentation** to evaluate the **graph building component** of SON-GOKU through **ablations** (see above Table), as discussed in our earlier response (7/9).

---

> ### Author Response · Authors · 2025-12-04
> **Update on Experiments for Reviewer hC6i, Weakness 2 (6/6)**
>
> We include detailed descriptions of the new ablations in Appendix K.3 and high-level descriptions in *Section 7.2* alongside interpretation in the same section (as you pointed out in *Weakness 3*, we have organized it this way for **easier reading** and to avoid a **fragmented narrative** while preserving space so as to not hit the page limit).
>
> In addition to the existing ablations, the graph rule ablations further show that the **specific way we define conflicts matters** up to a point. The simple threshold graph and the quantile-based ablations (including Quantile, DM) perform *almost identically* across datasets, suggesting that any construction that **reliably isolates the worst conflicting pairs** is sufficient. In contrast, rules that **disregard** the **magnitude of conflict** (Signed-only) or enforce **purely local control of degree** (kNN-Symmetric) degrade performance more noticeably, especially on NYUv2 and the tabular benchmarks. Overall, the best performing configurations are *precisely those* that **align with our assumptions**. An estimate of conflict (average over a history window) and a graph rule that **consistently separates strongly conflicting tasks** can yield **low conflict groups** that *satisfy* the **descent** and **recovery conditions** in Sections *5.1-5.2*.

---

### Author Response · Authors · 2025-12-04
**Response to All Reviewers**

We thank the reviewers for their insightful and positive feedback! We are encouraged that they view the problem of gradient interference in multi-task learning as important and timely (R1–R4); that they found our interference-based scheduler and graph coloring formulation to be novel and interesting (R2, R3, R4); that they appreciated the combine of theoretical guarantees with practical design choices (R2, R3); that they found the empirical results on multiple benchmarks and strong baselines to be promising and competitive (R2, R3, R4); and that they acknowledged the relevance of better controlling task interactions in real-world MTL setups (R1).

Feedback from the reviewers was largely scattered without any single shortcoming or issue standing out across all reviews. Broadly, the main themes were requests for stronger empirical rigor and more ablations (e.g., more seeds, standard deviations, runtime/memory measurements, scalability tests) (R1.W1, R1.W2, R2.W2, R3.W2, R3.Q1, R4.W2); questions about non-conflict grouping and general questions about our theory (R2.W1, R3.W1, R3.W2, R4.W1); suggestions to clarify presentation, including convergence plots, cleaner discussion of complexity, and open source code (R1.W3, R2.Q1, R3.Q2, R3.W4, R4.W3, R4.W4).

### **Additional Experiments and Empirical Analysis**
Reviewers asked for standard deviations, clearer runtime & memory measurements, and better evidence of scalability. In response, we used the rebuttal period to **re-run essentially every single experiment** (several seeds per experiment, from 2 to 15) and now report results as *mean* $\\pm$ *standard deviation* across the main paper and appendices. We added **seven new ablations** to assess the importance of the graph rule. Other new ablation-like studies, in which we evaluated both speed and performance, included varying backbone width to assess SON-GOKU's scalability and evaluating SON-GOKU's ability to adapt different affinity choices. We also expanded our runtime and memory analysis, incorporating metrics about speed, throughput, and GPU memory under different SON-GOKU parameters and backbones across several experiments. Together, these additions further demonstrate SON-GOKU's accuracy, speed, adaptability (generalizability), and memory benefits are consistent across seeds and settings, directly addressing reviewers' concerns about empirical rigor and reproducibility.

### **Theory, Additional Analysis, and Better Clarification**
Reviewers requested more intuition and formal justification for why non-conflict grouping should improve performance, and (indirectly) how general our guarantees are beyond cosine-based interference (R2.W1, R3.W1, R4.W1). We addressed this by adding a new appendix that connects non-conflict grouping to improved singal-to-noise properties in multi-task learning, building on reent work to make a mechanism behind SON-GOKU's gains more clear. We also formalized SON-GOKU as a modular conflict graph scheduler that can use any symmetric affinity or conflict matrix, and clarified that our descent, non-convex, convergence, and scheduling guarantees depend only on the resulting low-conflict groups, not on the specific approach used for affinity. We developed more formal theory on space complexity, time complexity, and non-conflict grouping. We added many miscellaneous pieces of theory, explanations, and reasoning throughout the paper. We paired some of the new theory with experiments. E.g., using TAG’s lookahead loss as the affinity, empirically validating our theory about affinity measures.

### **Presentation and Interpretation**
Some concerns focused on how the work is presented and interpretation. Some concerns in this aspect also came from us, as we noticed some concerns from reviewers came from basic misunderstanding (R1.W1). In the revision, we added Appendix S dedicated to visualizing logs from SON-GOKU (convergence, graph evolution, etc.), enabling better transparency and understanding of patterns and trends SON-GOKU follows during training. We added dedicated discussions regarding refresh rates, generalizability causality, and much more in response to reviewer questions and confusions. We also released code for reproducibility: **https://anonymous.4open.science/r/SON-GOKU-ICLR-95AB**

### **Additional Responses and Clarifications**
We recognized that Reviewer hC6i may have not been fully familiar with MTL and/or MTL optimization and therefore partially misunderstood the overall purpose of the paper. We made slight revisions to the paper and commented directly to Reviewer hC6i to help them better understand the positioning of SON-GOKU and the current state of the field.

### **Additional Info.**
There are many responses and pieces of information that could not be covered here, and so they are provided in individual comment responses to reviewers. For clarity, colored text in the paper corresponds to revisions for different reviewers: Blue (R1), Red (R2), Green (R3), Orange (R4).

---

### Meta-Review · Area_Chair_HtVW · 2026-01-13

**Summary:**

1. No measurement of training time and inference latency.
2. It is unclear whether the approach scales to larger backbones.
3. This paper lacks some parameter experiments, such as how the tau parameter changes throughout the training process, and how the task graph evolves in specific experiments.

**Reviewer Concerns:**

Addressed concerns:

2. The authors claim that there are many settings in which task interference does not vanish even with larger backbones.

Outstanding concerns:

1. Section 7.4 provides a comparison of SON-GOKU’s overhead against the overhead that other multi-task learning frameworks require. Where SON-GOKU has much more overhead but very less advantage than FAMO (Table 1), and the additional overhead has not significantly enhance the performance. In addition, the theoretical part only compare the proposed method with linearization, and has not show the advantage with MGDA or PCGrad, which is already been shown to have more advantage than linearization.

3. The current draft already describe how  and the conflict graph behave, but mostly in text and theory rather than with explicit visualizations. And promise to visualizes the task graph in the next version.

**Reviewer Scores:**

No

---

### Decision · Program_Chairs · 2026-01-26

Reject